# Microbial iron oxide respiration coupled to sulfide oxidation

Song-Can Chen[1,2,3✉], Xiao-Min Li[4,5], Nicola Battisti[3], Guoqing Guan[3,6], Maria A. Montoya[3], Jay Osvatic[7,8], Petra Pjevac[3,7], Shaul Pollak[3], Andreas Richter[4], Arno Schintlmeister[3], Wolfgang Wanek[4], Marc Mussmann[3✉] & Alexander Loy[3,7✉]

Microorganisms have driven Earth's sulfur cycle since the emergence of life[1–6], yet the sulfur-cycling capacities of microorganisms and their integration with other element cycles remain incompletely understood. One such uncharacterized metabolism is the coupling of sulfide oxidation with iron(III) oxide reduction, a ubiquitous environmental process hitherto considered to be strictly abiotic[7,8]. Here we present a comprehensive genomic analysis of sulfur metabolism across prokaryotes, and reveal bacteria that are capable of oxidizing sulfide using extracellular solid phase iron(III). Based on a phylogenetic framework of over hundred genes involved in dissimilatory transformation of sulfur compounds, we recorded sulfur-cycling capacity in most bacterial and archaeal phyla. Metabolic reconstructions predicted co-occurrence of sulfur compound oxidation and iron(III) oxide respiration in diverse members of 37 prokaryotic phyla. Physiological and transcriptomic evidence demonstrated that a cultivated representative, *Desulfurivibrio alkaliphilus*, grows autotrophically by oxidizing dissolved sulfide or iron monosulfide (FeS) to sulfate with ferrihydrite as an extracellular iron(III) electron acceptor. The biological process outpaced the abiotic process at environmentally relevant sulfide concentrations. These findings expand the known diversity of sulfur-cycling microorganisms and unveil a biological mechanism that links sulfur and iron cycling in anoxic environments, thus highlighting the fundamental role of microorganisms in global element cycles.

The global biogeochemical sulfur cycle is largely driven by bacteria and archaea that have evolved an array of enzymatic mechanisms for diverse dissimilatory sulfur redox transformations (hereafter called sulfur microorganisms). Sulfur transformations are linked to redox reactions of other elements, such as carbon, oxygen, iron and nitrogen[1–3]. The intertwined electron transfer reactions mediated by sulfur microorganisms in diverse ecosystems represent a metabolic network on the global scale that profoundly impacts biogeochemistry, climate and redox state on the Earth's surface over geological timescales[4–6]. Functional gene and genome-centric surveys have expanded the known diversity of specific guilds of sulfur microorganisms beyond cultivated representatives, and predicted previously unrecognized roles of uncultivated sulfur microorganisms in biogeochemical cycles across ecosystems[9–11]. However, the microbial genomic signatures for the full spectrum of dissimilatory sulfur metabolisms, including oxidation of reduced sulfur compounds (hereafter called sulfur oxidation), has not been explored. Moreover, most genome-based predictions of sulfur-related metabolisms lack support from experimental evidence.

Among the various transformation processes catalysed by sulfur microorganisms, the re-oxidation of sulfide, which occurs mainly by dissimilatory reduction of oxidized sulfur compounds such as sulfate, is a major biogeochemical process. Sulfur oxidation replenishes sulfate and contributes to the sulfur cycle in diverse ecosystems[12–14]. Microorganisms oxidize reduced sulfur compounds by utilizing light or chemical oxidants, including oxygen, nitrate and manganese oxides[15]. Ferric oxyhydroxides and oxides (hereafter called iron(III) oxides) represent one of the largest pools of oxidants on the Earth's surface[16]. The interaction with iron(III) oxides shapes the concentration of free sulfide and the cycling of sulfur in various anoxic environments, such as marine sediments, wetlands and aquifers[7,12,17–19]. However, current biogeochemical models consider the reaction of sulfide with iron(III) oxides as purely abiotic, producing mainly elemental sulfur (S(0)) and poorly crystalline FeS[7,8]. Although geochemical studies have hinted at microbial oxidation of sulfide to sulfate with iron(III) oxides[20–22], microorganisms and reactions that catalyse this process thus far remained uncharacterized.

Here we report the wide distribution of dissimilatory sulfur-transforming potential across the bacterial and archaeal tree of life, including many previously unsuspected microbial taxa. By genome-based reconstruction of the metabolism of potential sulfur-oxidizing

[1]State Key Laboratory of Soil Pollution Control and Safety, Zhejiang University, Hangzhou, China. [2]MOE Key Laboratory of Environment Remediation and Ecological Health, College of Environmental and Resource Sciences, Zhejiang University, Hangzhou, China. [3]Division of Microbial Ecology, Centre for Microbiology and Environmental Systems, University of Vienna, Vienna, Austria. [4]Division of Terrestrial Ecosystem Research, Centre for Microbiology and Environmental Systems Science, University of Vienna, Vienna, Austria. [5]Key Laboratory of Urban Environment and Health, Ningbo Observation and Research Station, Institute of Urban Environment, Chinese Academy of Sciences, Xiamen, China. [6]Doctoral School in Microbiology and Environmental Science, University of Vienna, Vienna, Austria. [7]Joint Microbiome Facility of the Medical University of Vienna and the University of Vienna, Vienna, Austria. [8]Division of Clinical Microbiology, Department of Laboratory Medicine, Medical University of Vienna, Vienna, Austria. ✉e-mail: songcan.chen@zju.edu.cn; marc.mussmann@univie.ac.at; alexander.loy@univie.ac.at

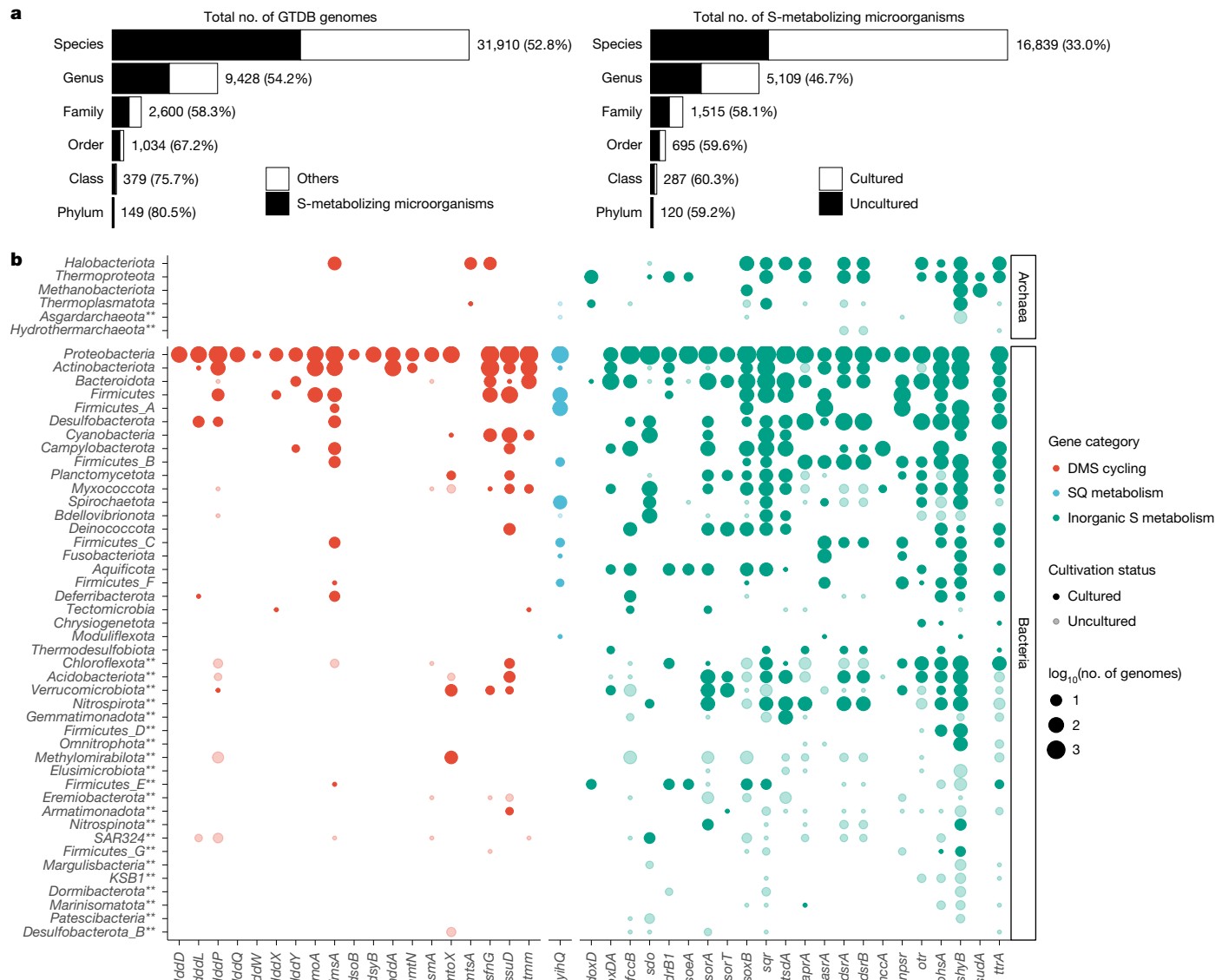

**Fig. 1 | Distribution of sulfur-cycling potential across bacterial and archaeal phyla. a**, Left, number of GTDB taxa carrying at least one of 42 sulfur-cycling marker genes. Right, among sulfur-metabolizing microorganisms, a substantial fraction is exclusively represented by uncultured taxa. **b**, Distribution of 42 sulfur-cycling marker genes across archaeal and bacterial phyla. The top 50 of 120 phyla with the highest number of sulfur-cycling genes are shown. The size of points indicates the number of genomes encoding a particular sulfur-cycling gene (with log transformation). Sulfur-cycling guilds lacking any cultivated representatives in a phylum are depicted in less opaque colour. Double asterisks indicate microbial phyla with no or few cultivated representatives. Genes associated with broad sulfur-cycling functional categories are shown with different colours. DMS, dimethylsulfide; SQ, sulfoquinovose.

microorganisms, we further revealed sulfur-based electron transfer pathways that could facilitate anaerobic sulfur oxidation coupled to the reduction of extracellular iron(III) oxide in diverse bacterial and archaeal taxa. Physiological and transcriptomic evidence of the predicted process in a cultivated representative, *D. alkaliphilus*, contributes to the evolving view on the biological coupling of the biogeochemical sulfur and iron cycles.

## Sulfur-metabolizing bacteria and archaea

We first established a computational framework to more accurately predict dissimilatory sulfur metabolism from bacterial and archaeal genomes (Supplementary Fig. 1). To this end, we performed phylogenetic analyses of 116 proteins that are involved in diverse sulfur redox transformations, and developed hidden Markov models (HMMs) for all monophyletic clades that corresponded to functional homologues of experimentally validated proteins (Supplementary Text).

The resulting phylogenies and clade-specific HMMs enabled us to effectively and accurately retrieve homologues of sulfur-cycling proteins from genomes and curate their function within a phylogenetic context (Supplementary Fig. 2).

Using this resource, we systematically queried a subset of 42 sulfur-cycling enzymes against representative genomes of all bacterial and archaeal species from the Genome Taxonomy Database (GTDB)[23]. The selected enzymes are key markers for specific, mostly dissimilatory sulfur metabolisms (Supplementary Table 1). More than half of the species, representing 120 (80.5%) of 149 known bacterial and archaeal phyla, encoded at least one sulfur-cycling marker protein (Fig. 1). Most genomes that encode a marker protein (for example, DsrA) also encode additional proteins (for example, DsrB, DsrC, DsrE and DsrF) associated with the respective marker protein for coordinated enzymatic function (Supplementary Fig. 3). The occurrence of sulfur-cycling capability in most bacterial and archaeal phyla reflects the deep integration of sulfur redox processes into microbial metabolism since the origin

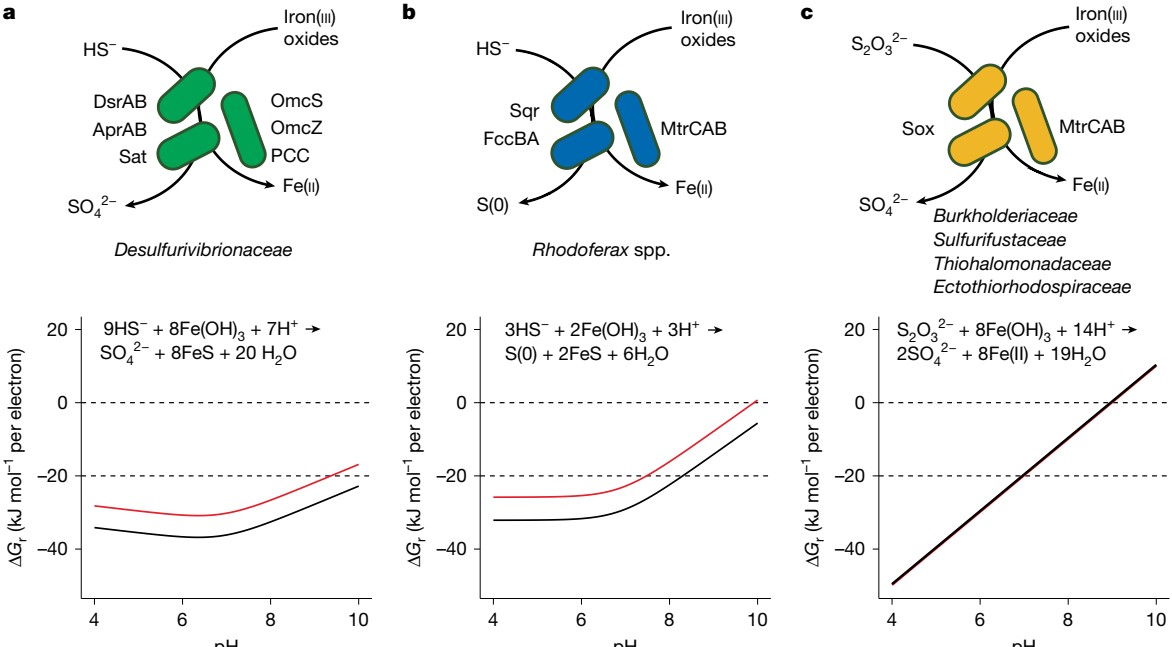

**Fig. 2 | Three potential metabolic pathways for coupling sulfur oxidation with reduction of iron(III) oxides in a single bacterium.** Top, genomic reconstruction predicts that dissimilatory reduction of iron(III) oxides is coupled with oxidation of sulfide to sulfate (reaction 1; **a**), oxidation of sulfide to elemental sulfur (reaction 2; **b**) and oxidation of thiosulfate to sulfate (reaction 3; **c**). Key enzymes from *Desulfobacterota* and *Gammaproteobacteria* taxa are indicated as examples. Bottom, Gibbs free energy ($\Delta G_r$) of the iron-dependent sulfur oxidation reactions across a range of pH was calculated, assuming two different environmental settings. Marine ecosystem conditions (red line): ionic strength ($I$) = 0.7 M, $T$ = 25 °C, [total dissolved sulfide] = 100 μM, [$SO_4^{2-}$] = 28 mM, [$S_2O_3^{2-}$] = 1 μM, [$Fe^{2+}$] = 10 μM; freshwater ecosystem conditions (black line): $I$ = 0.001 M, $T$ = 25 °C, [total dissolved sulfide] = 100 μM, [$SO_4^{2-}$] = 100 μM, [$S_2O_3^{2-}$] = 1 μM, [$Fe^{2+}$] =10 μM. The red and black lines overlap in **c**. Sox, thiosulfate-oxidizing system; OmcS and OmcZ, outer membrane multi-haem c-type cytochromes; PCC, porin–cytochrome complex.

of life[2,5,6]. In this context, 5,561 species with sulfur-cycling potential, affiliated to 71 phyla, were represented exclusively by genomes from so far uncultured microorganisms (Fig. 1). This highlights the broad taxonomic distribution of sulfur-cycling potential and underscores that many of the microorganisms that we identified here as capable of sulfur-cycling remain uncultured and poorly characterized. Our overview of putative sulfur microorganisms provides a foundation for more detailed metabolic predictions and experimental validation.

## Microbial S oxidation with iron(III) oxide

We next screened the genomes of the identified sulfur microorganisms for potential electron transfer metabolisms that facilitate anaerobic sulfur oxidation by reducing iron(III) oxides, a microbial physiology of potentially broad biogeochemical importance in marine and terrestrial ecosystems[24]. In contrast to coupling of elemental, zero-valent sulfur oxidation to dissolved iron(III) reduction by acidophiles in highly acidic environments[25,26] (pH < 3), no microorganism has been shown to gain energy from coupled reduction of solid phase extracellular iron(III) oxides and oxidation of reduced sulfur compounds. We identified co-occurring genetic features for dissimilatory sulfur oxidation and extracellular iron(III) reduction metabolisms across 37 bacterial and archaeal phyla (Supplementary Text and Supplementary Figs. 4 and 5). Reconstruction of sulfur and iron energy conservation pathways revealed three metabolic options (Fig. 2). The first option couples iron(III) reduction to the oxidation of sulfide to sulfate (reaction 1). The underlying metabolic pathway was found in, for example, members of *Desulfurivibrionaceae*, whose genomes encode a full suite of enzymes (sulfate adenylyltransferase (Sat), adenosine-5′-phosphosulfate reductase (AprAB) and dissimilatory sulfite reductase (DsrAB)) for the canonical sulfate reduction pathway (Supplementary Fig. 5). The DsrAB sequences of *Desulfurivibrionaceae* form a well-supported

monophyletic clade, including *D. alkaliphilus*, which operates DsrAB in reverse during sulfide oxidation to sulfate[27] (Supplementary Fig. 6). The same genomes also contain multiple homologues of *Geobacter*-type cytochromes, which are necessary for dissimilatory iron(III) reduction (Supplementary Fig. 5), including extracellular OmcS and porin–cytochrome complex[28–30] (OmaB–OmbB–OmcB). The interactions of these enzymes would establish a conduit channeling sulfide-derived electrons to insoluble extracellular substrates such as iron(III) oxides. The second option involves a sulfide:quinone oxidoreductase (Sqr) and FccBA for sulfide oxidation, and a multi-haem protein complex (MtrCAB) for extracellular respiration of iron(III) oxides (Supplementary Fig. 7). Homologues of Sqr, FccBA and MtrCAB were detected in two uncultured *Rhodoferax* species. The Sqr is affiliated to type I Sqr that has a physiological role in sulfide-based energy transduction, as described in *Aquifex aeolicus*[31] and *Rhodobacter capsulatus*[32]. The MtrCAB homologue encoded in the *Rhodoferax* genome is closely related to MtrCAB from the known iron(III) reducer *Rhodoferax ferrireducens*[33,34]. These genetic systems could support an energy metabolism that couples iron reduction with dissimilatory oxidation of sulfide to elemental sulfur (reaction 2). Additionally, MtrCAB was detected in multiple known thiosulfate oxidizers (Supplementary Fig. 5), including cultivated members within *Burkholderiaceae*, *Sulfurifustaceae*, *Thiohalomonadaceae* and *Ectothiorhodospiraceae*. This suggests a third option of extracellular iron(III)-dependent thiosulfate oxidation (reaction 3). Beyond *Desulfobacterota* and *Proteobacteria*, members from 35 other microbial phyla also have the potential to catalyse reactions 1–3 via different gene combinations (Supplementary Text and Supplementary Figs. 4 and 5).

Calculation of the Gibbs free energy showed that all three reactions could provide energy to microorganisms, even in neutral to moderately alkaline conditions (Fig. 2). For example, iron(III)-dependent sulfide oxidation under natural settings (that is, freshwater and marine

sediment) yields −20 to −40 kJ per mole electron, which is considered sufficient to support microbial growth[35]. Overall, these findings suggest wider occurrence of sulfur oxidation-dependent energy metabolisms across microbial taxa than previously recognized and provide testable hypotheses regarding the physiology of uncultured and cultured microorganisms.

## Physiology and transcriptome

We aimed to experimentally validate the genome-predicted potential to oxidize sulfide with iron(III) oxide in *D. alkaliphilus*, which encodes the genomic repertoire for reaction 1. We prioritized validatation of this predicted sulfide metabolism given the pervasive co-occurrence of sulfide and iron(III) oxides across a wide range of ecosystems[7,18,36]. *D. alkaliphilus* has a versatile sulfur metabolism. It grows by disproportionation of elemental sulfur or by coupling sulfide oxidation with nitrate reduction[27,37] but was not known to reduce iron(III) oxides. Here we expand the known physiology of *D. alkaliphilus* by showing its capability to reduce iron(III) oxides (ferrihydrite) with formate, poorly crystalline FeS or dissolved sulfide as electron donor. Initial incubation of *D. alkaliphilus* with ferrihydrite and formate led to simultaneous formate consumption and Fe(II) production, with the following stoichiometry: $HCOO^- + 2Fe(III) \rightarrow CO_2 + 2Fe(II) + H^+$. This verified its capacity to reduce extracellular solid iron(III) oxides (Fig. 3a).

We then fed *D. alkaliphilus* FeS as electron donor and ferrihydrite as electron acceptor (Fig. 3b). Over a five-day incubation period, we observed the concurrent production of sulfate and Fe(II). By contrast, Fe(II) production was not detected in autoclaved controls, indicating that FeS was not chemically oxidized by ferrihydrite to form more oxidized sulfur species[38], such as S(0). This observation excludes the possibility of sulfate formation by microbial disproportionation of S(0). Sulfate formation by FeS oxidation with residual nitrate from the inoculum was also excluded, as addition of FeS alone (that is, without ferrihydrite addition) did not result in sulfate accumulation. The slight increase of sulfate in ferrihydrite-only controls is probably owing to residual sulfide from the inoculum. After accounting for sulfur transformations observed in control incubations, the calculated ratio of sulfate to Fe(II) formation was close to the predicted stoichiometry of 1:8 for reaction 1 (Fig. 3b), indicating that *D. alkaliphilus* oxidizes FeS-sulfide to sulfate by reducing ferrihydrite. The slight deviation from the expected stoichiometry is likely to result from reverse electron flow for carbon fixation (Supplementary Text and Supplementary Figs. 8 and 9).

Finally, to examine whether *D. alkaliphilus* oxidizes free sulfide with iron(III) as electron acceptor, we incubated the culture with ferrihydrite and supplied 1 mM dissolved sulfide daily (Fig. 3c). This led to progressive accumulation of sulfate, S(0), and Fe(II). In sterile controls, S(0) and Fe(II) accumulated, but not sulfate. Similarly, sulfate was not formed in biotic controls lacking either ferrihydrite or sulfide (Fig. 3c, Supplementary Fig. 10 and Supplementary Text). These results showed that sulfate was formed in a biological process requiring ferrihydrite. We reasoned that biological oxidation of dissolved sulfide with ferrihydrite contributed to sulfate formation, as evidenced by: (1) exclusion of sulfur disproportionation as the only source of sulfate; (2) a biphasic production of sulfate following the supply of dissolved sulfide; and (3) accelerated sulfide consumption with cells compared to abiotic controls. We consider these lines of physiological evidence separately in the following three paragraphs.

A plausible source of sulfate in our incubation is microbial disproportionation of S(0), derived from the chemical oxidation of sulfide by ferrihydrite[12,39]. Under this canonical biogeochemical scenario, consumption of dissolved sulfide is controlled by the availability of reactive sites on the surface of ferrihydrite[7], and the rate should decrease over time with periodic supply of sulfide due to surface saturation and production of sulfide by S(0) disproportionation. However, tracing the kinetics of dissolved sulfide removal revealed accelerated consumption

upon successive supply of 1 mM sulfide, whereas the sterile control followed the predicted trend (Fig. 3d,e). The faster rate indicates that additional processes consumed the dissolved sulfide. We suggest dissimilatory iron(III) reduction fuelled by reduced sulfur species oxidation[40] as an additional process that rapidly precipitates dissolved sulfide by producing excessive Fe(II) (Fig. 3f). Depending on the sulfur speciation in our incubation, the sulfur compounds supporting biological iron(III) reduction include dissolved sulfide, FeS and S(0). Their oxidation, together with S(0) disproportionation by *D. alkaliphilus* contributes to the observed sulfate formation.

To probe for direct oxidation of dissolved sulfide to sulfate by *D. alkaliphilus*, we measured sulfate formation upon sulfide addition (1 mM) to ferrihydrite at higher temporal resolution. We observed biphasic sulfate formation dynamics within 24 h (Fig. 3g). Within the first 70 min, when dissolved sulfide was available (phase I), sulfate was formed at an average rate of 0.69 fmol cell$^{-1}$ h$^{-1}$. However, the rate decreased by approximately tenfold (0.061 fmol cell$^{-1}$ h$^{-1}$) after the depletion of dissolved sulfide (phase II). To measure the effect of poorly soluble sulfur species S(0) and FeS on sulfate accumulation rates, we first allowed sulfide and ferrihydrite to react for 70 min, after which dissolved sulfide has been completely transformed into FeS and S(0), and only then inoculated cells. Here we also observed a low sulfate formation rate (0.047 fmol cell$^{-1}$ h$^{-1}$) comparable to the one in the sulfide-free phase II (Fig. 3g). These results demonstrated that the presence of poorly soluble sulfur species alone (such as S(0) and FeS) was insufficient to sustain the sulfate formation rate observed at phase I. We thus attribute the sulfide-dependent, high sulfate formation rate of *D. alkaliphilus* in phase I to direct biological sulfide oxidation with ferrihydrite.

To further show that *D. alkaliphilus* directly oxidizes dissolved sulfide with iron(III) oxide, we tracked sulfide kinetics in the culture incubated with a low concentration of dissolved sulfide (approximately 50 μM) and ferrihydrite. We hypothesized that the low sulfide concentration reduces its chemical reaction rate with ferrihydrite, enabling a clear readout of the biological sulfide transformation process. Indeed, sulfide was readily consumed by the ferrihydrite-amended culture down to 10 μM within 25 min, whereas 20–30 μM of sulfide remained in controls lacking cells. Repetitive addition of sulfide showed the same recurring pattern. The culture rapidly consumed sulfide and its estimated rate constant of removal was significantly higher than of abiotic controls (P < 0.01, Student's *t*-test; Extended Data Fig. 1). This pattern was observed reproducibly in independent incubations with different cell densities (Extended Data Fig. 1). By ruling out alternative biogeochemical processes that support accelerated sulfide consumption (Supplementary Text), we concluded that *D. alkaliphilus* participates in sulfide transformation using ferrihydrite. Repeated supply (*n* = 8) of sulfide showed the culture transformed more than 90% of spiked sulfide to sulfate with negligible S(0) formation, while producing an excessive amount of Fe(II) (Extended Data Fig. 2). In the abiotic control, however, S(0)−but not sulfate−was the main oxidized sulfur compound and Fe(II) accumulated to much lower levels (Extended Data Fig. 2). These results indicate that respiration of ferrihydrite with sulfide by *D. alkaliphilus* outpaces the chemical reaction rate between sulfide and ferrihydrite at environmentally relevant low sulfide concentrations.

To show that coupled sulfide and iron(III) oxides metabolism supports growth, we monitored the cell density of *D. alkaliphilus* during 4- to 13-day incubation with ferrihydrite and sulfide. We observed twofold to threefold increases in cell numbers for ferrihydrite-incubated cultures periodically amended with either 1 mM dissolved sulfide, approximately 50 μM dissolved sulfide or FeS (Fig. 4a–c). In comparison, incubation of *D. alkaliphilus* with dissolved sulfide and nitrate led to a 5−6 fold increase of cell number over 3 days (Fig. 4d and Supplementary Text). The calculated specific growth rate was higher in ferrihydrite cultures with dissolved sulfide (0.288 ± 0.015 day$^{-1}$ at 1 mM and 0.436 ± 0.074 day$^{-1}$ at 50 μM) compared to those with solid phase FeS (0.089 ± 0.005 day$^{-1}$) (Supplementary Fig. 11), probably owing to

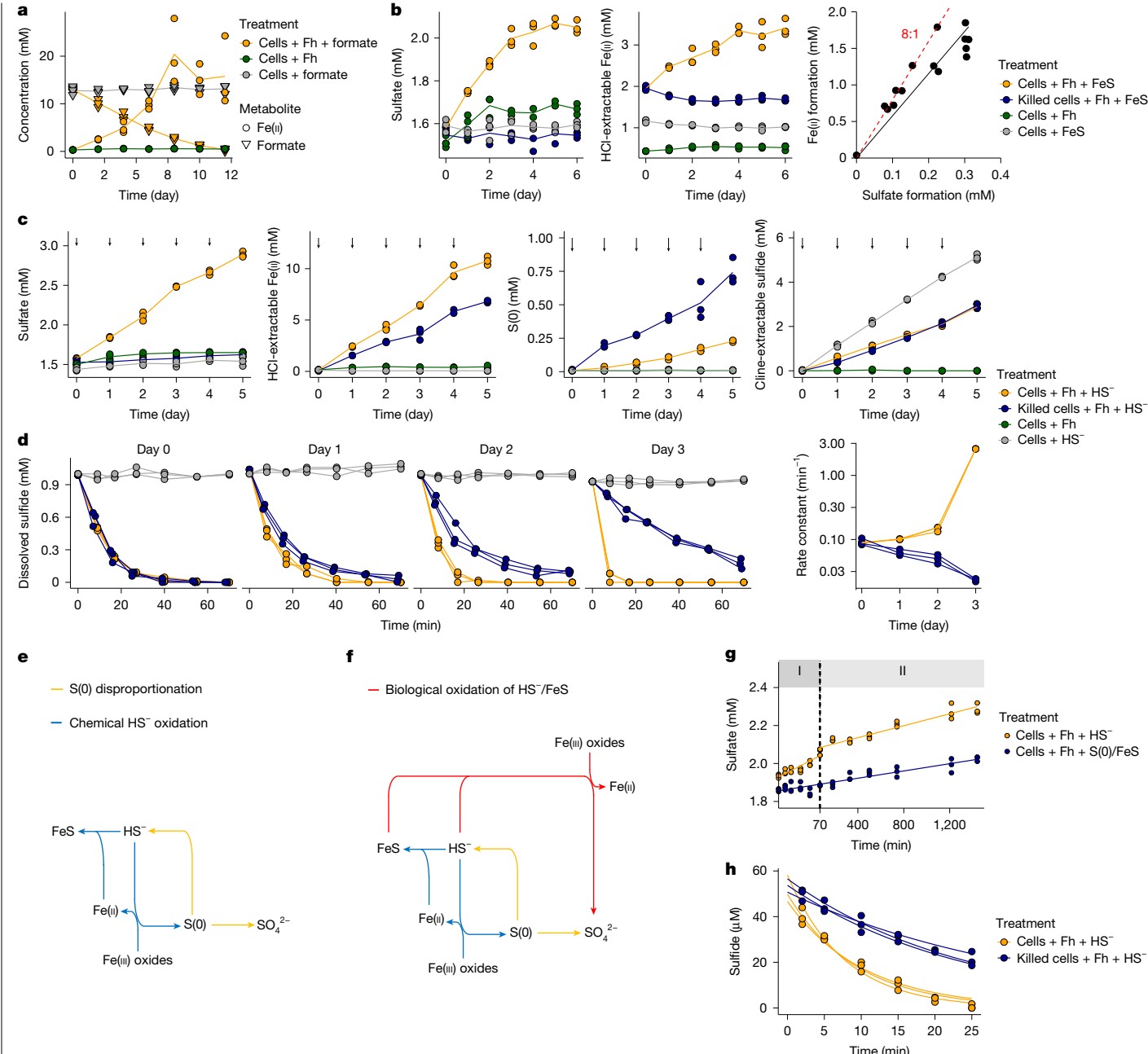

**Fig. 3 | *D. alkaliphilus* is capable of reducing ferrihydrite and oxidizing sulfide or FeS. a**, *D. alkaliphilus* oxidizes formate (triangles) while reducing ferrihydrite (Fh) to Fe(II) (circles). No Fe(II) or formate consumption occurred in the absence of Fh or formate. **b**, Left and middle, *D. alkaliphilus* couples oxidation of poorly crystalline FeS to sulfate with reduction of Fh. Sulfate and Fe(II) formation were not detected in abiotic controls or in cultures with FeS or Fh alone. Right, the ratio of sulfate to Fe(II) formation (solid line) is close to predicted 1:8 stoichiometry (dashed line). **c**, Sulfate, HCl-extractable Fe(II), elemental sulfur and Cline-extractable sulfide during the incubation of *D. alkaliphilus* cultures with Fh and daily spike of sulfide (1 mM). Parallel incubations with killed cells, or living cells without Fh or sulfide served as controls. Arrows indicate sulfide spike. **d**, Kinetics of dissolved sulfide and first-order rate constant in the incubations described in **c**. **e**, The canonical biogeochemical scenario of sulfate formation considering chemical sulfide oxidation to elemental sulfur (S(0)), followed by bacterial disproportionation. **f**, Updated scenario considering the contribution of sulfate formation from microbial oxidation of sulfide or FeS with iron oxides. **g**, Biphasic sulfate formation by *D. alkaliphilus* during 24 h incubation with ferrihydrite and sulfide. Phases I and II differ by sulfide availability. Phase I is shown with a larger scale on the *x* axis for visualization. A control for phase II was conducted by inoculating cells after the chemical reaction between sulfide and ferrihydrite (cells + Fh + S(0)/FeS). **h**, Faster consumption of a low concentration (approximately 50 μM) of sulfide by *D. alkaliphilus* compared with chemical controls lacking cells. Three spikes of sulfide were supplied at 1.5 h intervals. Results with different cell density are shown in Extended Data Fig. 1. In **a**–**d**, **g**, **h**, triplicate cultures (*n* = 3) were used for each incubation condition.

the limited solubility of FeS. By contrast, no or only minor increases in cell density were detected in the dissolved sulfide-only, FeS-only and ferrihydrite-only controls (Fig. 4a,b). These results demonstrate that *D. alkaliphilus* is capable of utilizing sulfide or FeS together with ferrihydrite for growth. We attribute the overall restricted growth

with ferrihydrite to the low energy field and higher maintenance energy requirements at alkaline pH (Supplementary Text). To further explore the association between growth and carbon fixation, we supplemented parallel cultures with [13]C-bicarbonate (10 atom%), ferrihydrite, and sulfide or FeS. Carbon isotope composition analysis

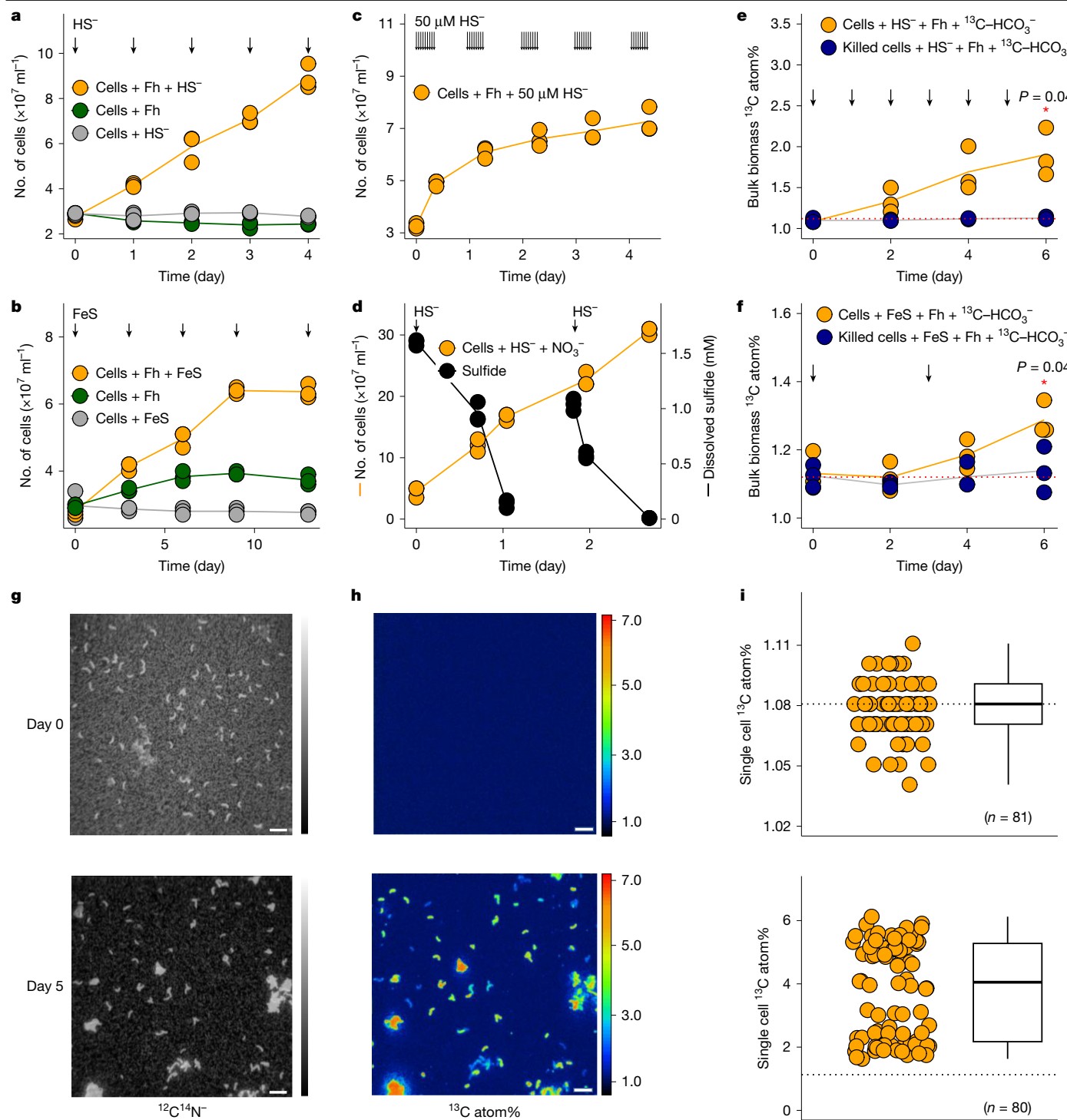

**Fig. 4 | *D. alkaliphilus* grows autotrophically with ferrihydrite and sulfide or FeS. a**, Mean cell density ($n = 3$) increased significantly ($P = 2.18 \times 10^{-8}$; ANOVA) over 4 days of incubation with ferrihydrite and daily addition of sulfide (1 mM). **b**, Mean cell density ($n = 3$) increased significantly ($P = 3.79 \times 10^{-10}$; ANOVA) over 13 days of incubation with ferrihydrite and periodic spikes of FeS (approximately 1 mM). **c**, Mean cell density ($n = 3$) increased significantly ($P = 1.84 \times 10^{-8}$; ANOVA) over 5 days of incubation with ferrihydrite and periodic addition of approximately 50 µM sulfide. **d**, Mean cell density ($n = 3$) increases significantly ($P = 5.53 \times 10^{-11}$; ANOVA) alongside sulfide consumption over 3 days of incubation with sulfide and nitrate (4 mM). **e**, Bulk $^{13}$C abundance (atom%) increased ($P = 2 \times 10^{-4}$; ANOVA) over 6 days in living cultures ($n = 3$; orange circles) incubated with 10% $^{13}$C-bicarbonate, ferrihydrite, and daily addition of sulfide (1 mM). **f**, Bulk $^{13}$C abundance increased ($P = 0.003$; ANOVA) over 6 days in living cultures ($n = 3$) incubated with 10% $^{13}$C-bicarbonate, ferrihydrite, and two spikes of FeS (approximately 1 mM). **e,f**, Asterisks denote significant differences ($P < 0.05$; two-sided Student's $t$-test) on day 6. Dashed lines indicate natural $^{13}$C levels. **a–f**, Arrows indicate sulfide or FeS addition. **g,h**, Nano-scale secondary ion mass spectrometry (NanoSIMS) images showing cellular biomass (**g**; $^{12}$C$^{14}$N$^{-}$) and $^{13}$C content (**h**) of cells at day 0 and 5. Colour scale refers to $^{13}$C atom%. Scale bars correspond to 5 µm. **i**, Dot plots displaying the $^{13}$C content of individual cells analysed in a representative NanoSIMS field view at day 0 ($n = 81$ cells) and 5 ($n = 80$ cells). Box plots show the median, the 25th and 75th percentiles of $^{13}$C content of cells; whiskers extend to 1.5 times the interquartile range from the first and third quartiles.

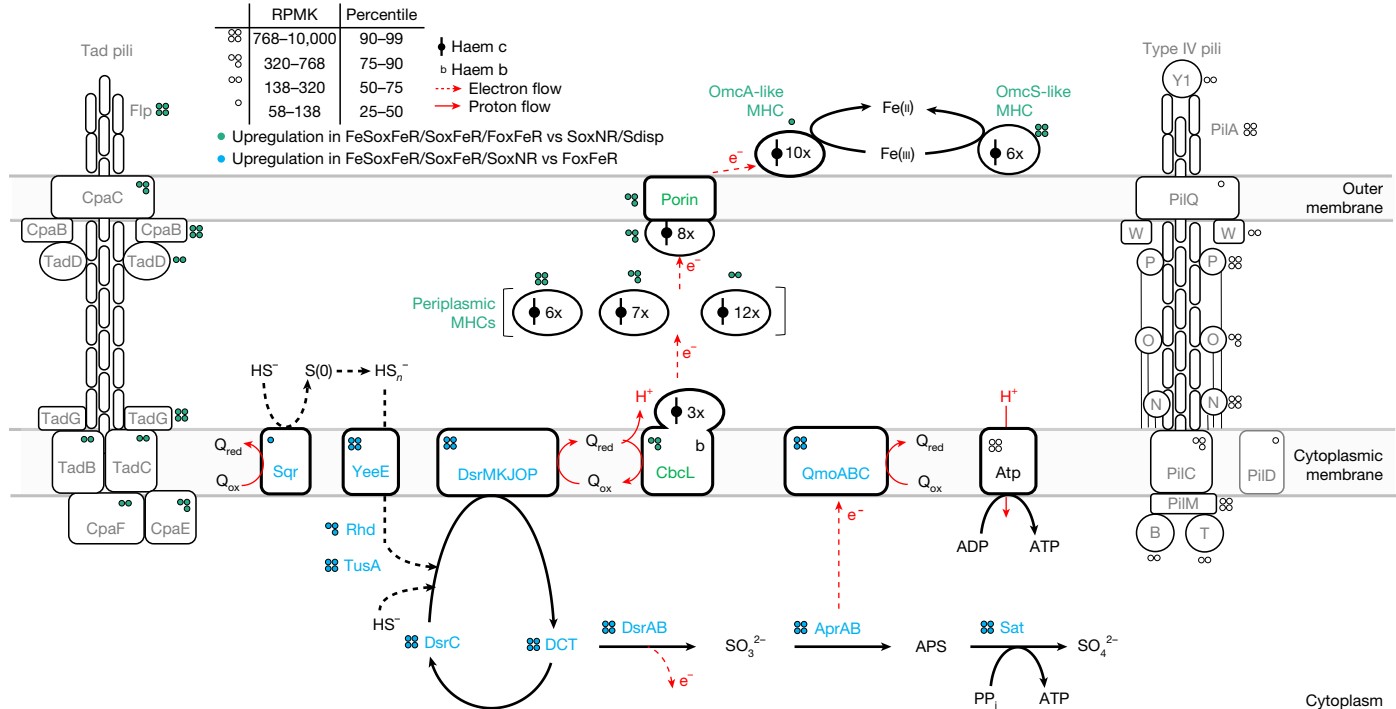

**Fig. 5 | Schematic metabolic model of *D. alkaliphilus* for sulfide oxidation with iron(III) oxides based on comparative transcriptomics.** *D. alkaliphilus* uses a reversed canonical dissimilatory sulfate reduction (Dsr) pathway to oxidize sulfide to sulfate (blue protein labels). Sulfide reacts with DsrC to form DsrC trisulfide (DCT), which is oxidized stepwise to sulfate via DsrAB, AprAB and Sat. The reducing equivalents are transferred to iron(III) oxides via EET mechanisms (green protein labels) facilitated by MHCs. *D. alkaliphilus* also has genes encoding type IV pili (grey protein labels). The tight adhesion (Tad) pilus may drive the cell adherence to the surface of solid iron oxides and/or FeS. The transcription level of each gene during MISO is indicated by mean reads per kilobase of transcript per million mapped reads (RPKM) of four replicates corresponding to the colour in the legend. Blue and green dots show significantly increased gene transcription during sulfide/FeS oxidation and ferrihydrite

reduction, respectively. The identifier, full name, RPKM, statistical significance of differential transcription and genomic arrangement for each gene are provided in Extended Data Fig. 3 and Supplementary Tables 2–5. Electron flow and proton translocation are indicated by red arrows. Black dotted arrows indicate putative sulfur redox reactions with unclear enzymology. APS, adenosine-5′-phosphosulfate, $Q_{ox}$, oxidized menaquinone; $Q_{red}$, reduced menaquinone; FoxFeR, incubation of *D. alkaliphilus* under formate-oxidizing and ferrihydrite-reducing conditions; FeSoxFeR, incubation of *D. alkaliphilus* under FeS-oxidizing and ferrihydrite-reducing conditions; Sdisp, incubation of *D. alkaliphilus* under S(0) disproportionation conditions; SoxFeR, incubation of *D. alkaliphilus* under sulfide-oxidizing and ferrihydrite-reducing conditions; SoxNR, incubation of *D. alkaliphilus* under sulfide-oxidizing and nitrate-reducing conditions.

of bulk biomass and single cells revealed enrichment of $^{13}$C in living cells after five-day incubation (Fig. 4e–i), reflecting a chemoautotrophic lifestyle of *D. alkaliphilus* when growing on ferrihydrite and dissolved sulfide or FeS.

After providing physiological evidence for microbial reduction of extracellular iron(III) being coupled to sulfide oxidation (MISO) in *D. alkaliphilus*, we revealed the differential activity of candidate genes of this metabolism by comparative transcriptomics under various ferrihydrite-amended and ferrihydrite-free growth conditions. Sulfide oxidation is most probably achieved by the reversal of the canonical dissimilatory sulfate reduction pathway (Fig. 5 and Extended Data Fig. 3), as proposed for *D. alkaliphilus* growing under nitrate-reducing conditions[27] and its close relatives, *Electrothrix* cable bacteria, which grow by oxygen-dependent sulfide oxidation[41]. All genes required for this pathway had substantial transcription in *D. alkaliphilus* during MISO, and many showed significant upregulation (adjusted *P* value ($P_{adj}$ < 0.05) compared with cultures grown with formate as electron donor. Additionally, *D. alkaliphilus* may oxidize sulfide in a two-step process in which sulfide is initially converted into zero-valent sulfur in the periplasm, which is then either disproportionated or transported into the cytoplasm to enter the reverse Dsr pathway[27]. Multiple genes proposed for this pathway were transcribed and/or upregulated during MISO growth, including those encoding Sqr, the sulfur-transferring membrane protein YeeE, the rhodanese Rhd and the sulfur transferase TusA.

After the transfer of sulfide-derived electrons to the membrane quinone pool via the proposed sulfide oxidation pathways, their flow to extracellular ferrihydrite is central to the activities and energy conservation mode of MISO. We propose that this extracellular electron transfer (EET) is facilitated by a network of multi-haem cytochromes (MHCs) at the inner membrane, periplasmic space and outer membrane, similar to mechanisms shown for the model iron(III)-respiring bacteria *Geobacter sulfurreducens* and *Shewanella oneidensis*[42–44]. During MISO growth, *D. alkaliphilus* showed upregulated transcription of 13 out of 46 predicted MHC genes compared with nitrate-reducing and S(0)-disproportionation growth conditions (Extended Data Fig. 4). The most prominent MHC, upregulated by more than 100-fold (Supplementary Fig. 12), was predicted to be extracellular, and to share structural homology and haem arrangement with OmcS from *Geobacter* (Extended Data Fig. 5). This OmcS homologue may facilitate EET to iron oxides (Fig. 5, Supplementary Text and Supplementary Fig. 13), as observed in *G. sulfurreducens*[28,45]. Another differentially transcribed MHC, also predicted to be extracellular, is homologous to OmcA, which is involved in anaerobic iron reduction in *Shewanella*[46]. The remaining MHCs are predicted to be located in the periplasm or associated with the cytoplasmic membrane. One of the latter includes a homologue of a cytochrome *bc* complex CbcL that is essential for the growth of *G. sulfurreducens* on iron(III) oxides[47]. As proposed for *Geobacter*, the periplasmic MHCs may facilitate EET by channelling electrons to the extracellular MHCs[30]; the CbcL homologue may pass

electrons from the quinone pool to periplasmic MHCs, while generating a proton motive force via a scalar mechanism[47]. The involvement of these MHCs in the reduction of iron(III) oxides was further supported by their increased transcription in *D. alkaliphilus* grown with formate and ferrihydrite (Extended Data Fig. 4). *D. alkaliphilus* also upregulated transcription of genes for tight adherence (Tad) type IV pilus assembly proteins during iron(III) reduction. Transcription of *pilA*, encoding the main component of another type IV pilus, was high across all growth conditions but not increased during iron(III) reduction. These proteins were suggested to have a role in EET by forming electrically conductive pili[48] or facilitating the surface attachment and/or the secretion of extracellular MHCs[49–52]. Under MISO growth conditions, genes involved in Wood Ljungdahl pathways ranked among the top 30% highly transcribed genes (Supplementary Fig. 14), suggesting a role in autotrophic carbon fixation.

## Ecological and biogeochemical impacts

Experimental validation of MISO in *D. alkaliphilus* expands the known physiologies of bacteria, and broadens the diversity of microbial sulfide oxidation pathways beyond what was proved possible in the absence of oxygen. Unlike oxidation of S(0) with dissolved iron(III) by acidophiles[26], MISO relies on solid phase iron(III) oxides, which represent the most prevalent form of iron in natural habitats with mildly acidic to mildly alkaline pH. Our data suggested that MISO utilizes an elaborate multi-haem apparatus to conduct EET to iron(III) oxides, similar to iron-reducing model organisms. The proposed sulfide-fuelled EET model provides a mechanistic basis for several underexplored sulfide oxidation observations, including those driven by manganese oxides[15], coupled with electricity generation in microbial fuel cells[53–55], and supporting direct interspecies electron transfer to methanogens[56,57]. The genomic potential to perform MISO appears as a conserved trait within the *Desulfurivibrionaceae* family (Extended Data Fig. 6). *Desulfurivibrionaceae* members with the MISO gene set occupy a wide range of habitats in marine (for example, sediments and hydrothermal vents), freshwater (for example, aquifers and lake sediments), terrestrial (for example, wetlands and soils) and engineered (for example, microbial fuel cells) environments. These observations suggest that MISO-performing *Desulfurivibrionaceae* members are biogeochemically relevant in diverse ecological contexts. Given the presence of genomic potential for MISO in microbial lineages beyond *Desulfurivibrionaceae* (Fig. 2, Supplementary Fig. 4 and Supplementary Text), we envisage an even broader environmental distribution of MISO capacity.

This study adds a new dimension to the understanding of the interplay of sulfur and iron biogeochemistry in anoxic environments. Although re-oxidation of sulfide with iron(III) oxides has previously been regarded to be a strictly chemical reaction, we demonstrate here that MISO can not only outperform chemical oxidation of sulfide at environmentally relevant concentrations, but also acts on solid phase FeS-sulfide, which known to be chemically inert towards iron(III) oxides[38]. In this context, FeS, which is widespread in anoxic environments, offers a distinct ecological niche for MISO bacteria, largely devoid of competition from chemical oxidation. Thermodynamic modelling indicates that MISO is exergonic over a range of ecologically relevant pH values and substrate conditions. This contrasts with canonical reduction processes of iron(III) oxides, which become thermodynamically unfavourable at alkaline pH[19]. MISO is likely to overcome such energy limitation by exploiting highly reactive sulfide that maintains its reducing power over a wide pH range. The free energy yielded from MISO is sufficient to support microbial growth and power sulfate formation at a rate resembling that of sulfate reduction in environments such as marine sediments (Supplementary Text). These unique features suggest that MISO bacteria may directly facilitate environmental sulfate formation from sulfide or FeS with iron(III) oxides, bypassing the need for S(0) disproportionation. Including MISO in the existing biogeochemical model could help explain the observed widespread geochemical pattern of sulfide removal and sulfate formation in anoxic, iron-rich marine, freshwater and terrestrial environments[14,20,22,58–60] (Supplementary Text). A notable example of such phenomena is the occurrence of high oxidation rates of sulfide, produced from dissimilatory sulfate reduction, back to sulfate in anoxic layers of marine sediments, where typical oxidants for sulfide oxidation, such as oxygen and nitrate, have been depleted, leaving iron(III) oxides as the primary oxidants[20]. On the basis of a first estimate, MISO could account for 1–7% of total sulfide oxidation to sulfate in marine sediments on a global scale, owing to the large flux of reactive iron from terrestrial runoff to the sea (Supplementary Text). The MISO metabolism may therefore be globally important in modulating the oxidation and reduction of sulfur, with broader implications for Earth's biogeochemical cycles and climate.

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

## Methods

### Phylogenetic framework and HMMs of sulfur-cycling proteins

To discern sulfur-cycling genes or proteins from their functionally divergent homologues, phylogenetic analysis was conducted for 116 sulfur-cycling proteins (Supplementary Table 1). For each protein family, sequences of enzymes with biochemically validated functions, including related sequences of enzymes with divergent functions (outgroups), were identified by literature surveys and recovered from SwissProt[61]. Additional homologues of experimentally validated proteins in KEGG prokaryotic genomes were retrieved using KEGG BLAST Search (https://www.genome.jp/tools/blast/; E value: $10^{-4}$). Distant homologues that did not align properly with biochemically characterized proteins (alignment length covered <50% of query and target length) were removed. The resulting homologues were de-replicated using CD-HIT v4.8.1 (ref. 62), with longest sequences retained as representatives. For computational efficiency, different clustering identity thresholds (75–95%) were chosen for de-replication to ensure the total number of representative sequences for phylogenetic analysis did not exceed 500. The genome context of the representative KEGG homologues was analysed by retrieving genes located in a distance of fewer than $n$ genes ($n = 7$–15), followed by annotation using biochemically characterized gene clusters based on BLAST analysis[63]. All representative KEGG homologues were further aligned with biochemically validated proteins and outgroups using Muscle v3.8.1551 (ref. 64). Poorly aligned regions were excised using TrimAl v1.4.rev15 (ref. 65). Protein phylogeny was inferred from the trimmed alignment using FastTree v2.1.7 (ref. 66) with -wag and -gamma options. Statistical support for each branch of the tree was estimated by nonparametric bootstrap ($n = 100$).

Information on reference sequences from biochemically verified proteins (for example, ingroup/outgroup, conserved residues or motif) and genomic contexts of all homologues were mapped on the tree. To identify monophyletic, orthologous clades within each tree, interior nodes of the annotated tree were scrutinized using the following criteria: (1) bootstrap support over 70%; (2) presence of at least one biochemically verified ingroup protein and absence of outgroup proteins; and (3) consistent gene neighbouring patterns and biochemical traits (that is, catalytic residues and PFAM domain composition) among its members. All descendants of the identified clade were regarded as functional orthologs of the biochemically verified protein. If possible, existing definitions of orthologous clades from previous phylogenetic analysis of sulfur-cycling proteins was preserved, including the well-recognized clades in the phylogeny of DsrAB[10] and Sqr[67]. For proteins for which the biochemically validated ingroup proteins formed polyphyletic groups, multiple monophyletic clades were proposed to fulfill our criteria.

To leverage our phylogenetic framework for large-scale homology searches, sequences from the defined monophyletic clades of sulfur-cycling proteins were used to build HMMs. A cut-off that optimizes the sensitivity and specificity of homology search was calculated for each HMM using receiver operating curve (ROC)[68]. This cut-off was embedded in the HMMER profile HMM file as the gathering threshold of the model (HMMER User's Guide, p. 108; ref. 69). The performance of the newly developed HMMs was compared with that of six published sets of HMMs for sulfur metabolism genes, including those from KoFam[70], TIGRFAM[71], PFAM[72], metabolicHMM[73], DiSCo[74], Teng et al.[75] and HMS-S-S[76]. This was accomplished by querying each HMM against the phylogeny-curated protein dataset using hmmsearch in HMMER v3.2.1 with a predefined cut-off (http://hmmer.org/). The performance of the various HMM sets in detecting sulfur-cycling genes and proteins was assessed in terms of specificity, sensitivity, and $F$ score (Supplementary Text). $F$ score balancing both precision and recall of the homology detection was calculated using $F$ score = 2 × (precision × recall) / (precision + recall).

### Sulfur-cycling genes in bacterial and archaeal genomes

To provide a comprehensive overview of sulfur metabolism across bacteria and archaea, the phylogeny-derived HMMs were searched against all genomes in GTDB release 95 (ref. 23) using hmmsearch with the --cut_ga option. Each retrieved homologue was then searched against the full set of phylogeny-derived HMMs using hmmscan with --cut_ga, and annotated as the HMM showing the highest score. For initial screening, a subset of genes ($n = 42$) was selected as markers for specific sulfur metabolisms if the gene: (1) has been widely recognized as a marker for a specific sulfur metabolism, (2) encodes a catalytic subunit essential for the activity of enzymatic complex; or (3) on its own confers a specific sulfur redox transformation (see justification for each of selected genes in Supplementary Table 1). The retrieved homologues were further curated using our reference phylogeny of sulfur proteins. Specifically, the GTDB homologues were aligned with sequences contained in our reference phylogeny using Muscle. A maximum-likelihood tree was reconstructed from the alignment trimmed by TrimAl. The tree was overlaid with biochemical information and data on the genomic context of sulfur genes, and visualized using ggtree[77]. The physiological role of the GTDB homologues was interpreted on the basis of their evolutionary relationship with biochemically validated proteins and genome context. To predict the dissimilatory iron(III) reduction potential, GTDB genomes were screened for marker genes involved in EET on the basis of homology search and/or motif analysis. Homologues of iron(III) reduction genes with established HMMs in FeGenie database (https://github.com/Arkadiy-Garber/FeGenie/tree/master/hmms/iron/iron_reduction) were retrieved using hmmsearch from HMMER v3.2.1, with cut-off recommended by FeGenie (https://github.com/Arkadiy-Garber/FeGenie/blob/master/hmms/iron/HMM-bitcutoffs.txt). Additionally, homologues of MmcA gene, which is involved in dissimilatory iron(III) reduction in *Methanosarcina acetivorans*[78], were extracted using BLASTP on the basis of an e-value of $10^{-4}$. The outer membrane MHCs responsible for EET with metal oxides in anaerobic methanotrophs[79] and putative electroactive bacteria[80] were recognized on the basis of the following: (1) the presence of four or more haem-binding motifs (CXXCH); and (2) their predicted outer membrane or extracellular localization, as determined by DeepProLoc v1.0 (ref. 81).

### Annotation and metabolic reconstruction of specific sulfur-cycling microbial lineages

The genomes of microbial lineages of interest were downloaded from the GTDB database. The protein-coding genes were predicted from the genome using Prodigal v2.6.3 with default setting. The predicted genes were annotated using KoFam[70], PFAM[72], and the EggNOG[82] database. Additional metabolic pathways were predicted using HMMs (Supplementary Table 6) downloaded from dbCAN[83], metabolicHMM[73], CANT-HYD[84], MicRhoDE[85] and FeGenie[86]. For HMM-based annotation, the HMMs were used as queries to search against microbial genomes using hmmsearch from HMMER v3.2.1, with cut-off recommended by each database (-T, -domT or -cut_ga options). The cellular localization of the protein was predicted using Signalp v6.0 (ref. 87). The completeness of the KEGG metabolic pathway was calculated on the basis of the definition of each module. The KEGG module is defined with a logic expression of K numbers that records the composition of enzymes in the pathway. A particular metabolic module was considered to be present in the genome when: (1) the diagnostic/marker genes of the module were detected; and (2) the overall completeness of the pathway module was >70%. The environmental distribution of the GTDB species was retrieved by searching their GTDB species name in the Sandpiper database[88]. The occurrence of the GTDB species across biomes was downloaded as CSV from Sandpiper (https://sandpiper.qut.edu.au/) and further visualized with R v4.1.0.

## Thermodynamic modelling

The Gibbs free energy associated with iron(III)-dependent sulfur oxidation at environmentally relevant conditions was estimated by following the guidelines described previously[89]. In brief, the actual Gibbs free energy of reaction ($\Delta G_r$) was calculated using:

$$\Delta G_r = \Delta G_r^0 + RT\ln Q_r$$

where $\Delta G_r^0$ refers to the standard Gibbs free energy of reaction, given in kJ mol$^{-1}$; $R$ and $T$ are the universal gas constant (8.314 J K$^{-1}$mol$^{-1}$) and the temperature in Kelvin, respectively; and $Q_r$ is the reaction quotient. $\Delta G_r^0$ values were calculated from the values of the standard Gibbs free energy of formation ($\Delta G_f^0$) of reactants and products (Supplementary Table 7). Values of $Q_r$ were determined from the activity ($a_i$) and the stoichiometric coefficient ($\nu_i$) of the $i^{\text{th}}$ chemical species involved in the reaction using:

$$Q_r = \prod a_i^{\nu_i}$$

The activity of the solvent (that is, pure water) and the solids (that is, ferrihydrite and FeS) were taken to be 1. The activity of dissolved ions was related to the concentration ($C_i$) using:

$$a_i = \gamma_i \times C_i / C_i^0$$

where $\gamma_i$ denotes the activity coefficient; $C_i^0$ represents the standard state concentration (usually 1 M). $\gamma_i$ for cations (that is, Fe$^{2+}$) and anions (that is, HS$^-$, S$_2$O$_3^{2-}$ and SO$_4^{2-}$) in solutions of different ionic strength were retrieved from Amend et al.[89]. Sulfide speciation in aqueous phase across a range of pH was determined from the pH, and p$K_{a1}$ (7.04) and p$K_{a2}$ (11.96) of hydrogen sulfide.

## Synthesis of ferrihydrite and poorly crystalline FeS

Synthetic ferrihydrite was prepared by titrating 1 M NaOH (Sigma Aldrich) into 0.1 M aqueous solution of FeCl$_3$ 6H$_2$O (Carl Roth) under vigorous stirring until pH 7.5 was reached, as described[90]. The suspension was centrifuged (Centrifuge 5804 R, Eppendorf) at 4 °C, 12,857$g$ and the ferrihydrite nanoparticles were washed thoroughly with deionized water to remove traces of chloride. The pellets were then freeze-dried (Alpha 1-4 LSCbasic, Christ) and stored at −20 °C for no longer than 3 weeks before use. The mineralogy was determined by LabRAM HR800 Raman microscope (Horiba Jobin-Yvon) equipped with a 532-nm neodymium-yttrium aluminium garnet laser and either 300 or 600 grooves/mm diffraction grating. Iron monosulfide (FeS; 30 mM) was prepared by mixing equal volume of 60 mM aqueous solution of Na$_2$S.9H$_2$O (Acros Organics) with 60 mM aqueous solution of FeCl$_2$.4H$_2$O (Sigma Aldrich) in an anaerobic chamber (Coy Lab) with 95% N$_2$ and 5% H$_2$ (O$_2$ < 1 ppm) atmosphere. The initially precipitated FeS is often designated as 'amorphous FeS' or 'poorly crystalline FeS'[91]. The dissolved sulfide in the FeS stock is less than 50 μM. The FeS solution was freshly prepared and used on the same day.

## Cultivation of *D. alkaliphilus* DSM 19089

*D. alkaliphilus* (DSM 19089, ATH2) was purchased from the German Collection of Microorganisms and Cell Cultures GmbH (DSMZ). The bacterium was cultivated at room temperature in an alkaline mineral medium (pH 9.3) containing 3 g NaCl (Carl Roth), 0.25 g K$_2$HPO$_4$ (Merck), 6.5 g Na$_2$CO$_3$ (Carl Roth), and 15 g NaHCO$_3$ (Sigma Aldrich) per liter of medium. After autoclaving, the medium was cooled down under N$_2$ atmosphere and supplemented aseptically with 1 ml liter$^{-1}$ of following components (all stored under anoxic conditions): 4 M NH$_4$Cl (Sigma Aldrich), 1 M MgCl$_2$ (Sigma Aldrich), trace element solution, Se-W solution, and four different vitamin solutions (DSMZ medium 1104). The culture was routinely grown under nitrate-reducing, sulfide-oxidizing conditions in 500 ml Schott bottles[27], with 2 mM Na$_2$S 9H$_2$O and 1.2 mM KNO$_3$ (Sigma Aldrich). This yielded a culture with an optical density at 600 nm (OD$_{600}$) of ~0.040, corresponding to a cell density of ~1.3 × 10$^8$ cells per ml. To test alternative growth modes, five incubation experiments were conducted, each supplemented with different electron donors and acceptors (details provided below). For all experiments, regularly maintained cultures (30 ml) that have been depleted in sulfide (<100 μM) and nitrate (<10 μM) were used as inoculum. Incubations were set up in 60 ml serum bottles and sealed with butyl rubber stoppers in the anaerobic chamber (N$_2$:H$_2$ = 95:5). Each culture was then flushed with pure N$_2$ to remove H$_2$ in the headspace, and incubated in the dark at room temperature. All incubations, abiotic and biotic controls from each experiment were set up in triplicates.

(1) *Incubations with sulfide and nitrate.* The incubations were set up by supplying 2 mM sulfide and 2 mM nitrate to 30 ml pre-growns cells in 60 ml serum bottles. Sulfide and nitrate was spiked using syringes flushed with pure N$_2$. The growth was monitored by phase-contrast microscopy and by the measurement of sulfide and sulfate over 3 days.

(2) *Incubations with elemental sulfur.* The incubations were initiated by adding 0.1 g elemental sulfur in 3 ml MilliQ water (Sigma Aldrich) to each of the serum bottles, followed by autoclaving at 110 °C for 60 min. After sterilization, 30 ml pre-grown cells were inoculated into the S(0) suspension (approximately 94 mM) and incubated under an N$_2$ atmosphere for 15 days. Microbial activity was monitored by measuring sulfide and/or sulfate.

(3) *Incubations with ferrihydrite and formate.* Synthetic ferrihydrite (0.2 g) was ground into fine particles with an agate mortar and pestle before being added to the culture. Assuming ferrihydrite has a composition[92] Fe(OH)$_3$, the final concentration of Fe(III) was approximately 62 mM. Formate was spiked anoxically using a syringe to a final concentration of 10 mM. To test the coupling of ferrihydrite reduction and formate oxidation, parallel cultures were set up with either ferrihydrite or formate alone. The culture activity was monitored by measuring total Fe(II) and formate concentrations over a 15 day incubation.

(4) *Incubations with ferrihydrite and poorly crystalline FeS.* Synthetic ferrihydrite (0.2 g) was supplied to the cultures as described in the incubation (3). To amend poorly crystalline FeS, 1 ml stock solution of freshly prepared FeS (30 mM) was anoxically spiked to the cultures using syringes, resulting in a final FeS concentration of 1 mM. Abiotic controls were prepared using 30 ml autoclaved cells as inoculum to test for chemical reactions between ferrihydrite and poorly crystalline FeS. Biotic controls amended with either ferrihydrite or FeS were set up to assess the impacts of residual sulfide and/or nitrate on culture activity. Cultures were sampled daily over 5 days for sulfate and total Fe(II) measurements.

(5) *Incubations with ferrihydrite and dissolved sulfide.* The cultures were prepared similarly as incubation (4), but with dissolved sulfide replacing FeS. Due to the rapid chemical reaction between dissolved sulfide and ferrihydrite, dissolved sulfide was anoxically spiked daily at a concentration of 1 mM using N$_2$-flushed syringes. Abiotic controls and the sulfide-only biotic controls received dissolved sulfide at the same concentration and frequency. To trace the transformation of S and Fe over 5 days, subsamples were taken daily for measurement of S(0), total Fe(II), sulfate, and Cline-extractable sulfide before the addition of sulfide. The consumption of dissolved sulfide in the cultures was monitored by sampling at 2, 10, 20, 35, 50 and 70 min after the spike of sulfide. The kinetics of sulfide consumption were modelled as a first-order reaction. The rate constant was estimated using the exponential decay model in the drm() function from the drc R package[93]. To compare sulfate formation patterns with and without sulfide, cultures incubated with ferrihydrite and sulfide were sampled for sulfate measurement following two phases after the 1st sulfide spike. During phase I, detectable sulfide was present

in the culture, and the samples were collected at 0, 11, 21, 37, 53 and 70 min of the incubation. Phase II, spanning the next 23 h, began once sulfide was depleted, with samples taken at 3, 5.5, 8.33, 12.33, 20.25 and 24 h. As a control for phase II, cells were incubated with chemically sulfidized ferrihydrite. Specifically, 1 mM sulfide was firstly added to 0.2 g ferrihydrite (approximately 62 mM) with 30 ml autoclaved cultures for chemical reaction. After 70 min, the reaction mixture was centrifuged (12,857$g$; room temperature) under anoxic conditions, and 30 ml of active cells were inoculated to resuspend the solid phase compounds (for example, FeS and S(0)) produced by chemical reaction between sulfide and ferrihydrite. Samples were collected from cultures for sulfate measurement at the same time intervals as those in phases I and II.

To test whether the microbial process can outperform the chemical process in transforming sulfide with ferrihydrite, the incubation (5) was repeated using ca. 50 μM sulfide instead of 1 mM. In this experiment, a small amount of sulfide was spiked three times at 1.5-h intervals into ferrihydrite-amended cultures, abiotic controls, and sulfide-only biotic controls. After each spike, subcultures (~ 0.3 ml) were collected at 2, 5, 10, 15, 20, and 25 min for dissolved sulfide measurements. Two biological replicates were performed for each treatment. To verify the reproducibility of the observed sulfide consumption pattern, incubations were conducted using inocula at different cell densities (OD$_{600}$ of 0.042, 0.075, and 0.086). To quantify the transformation of spiked sulfide during the incubation, independent cultures were set up using an inoculum with an OD$_{600}$ of 0.072 and supplied with eight spikes of sulfide. Ferrihydrite-amended cultures, abiotic controls, and sulfide-only controls received ca. 50 μM sulfide at 1.5-h intervals, whereas ferrihydrite-only biotic controls were spiked with anoxic water. Three replicate incubations were performed for each treatment. Subsamples were taken every three hours for concentration measurement of S(0), total Fe(II), sulfate and Cline-extractable sulfide.

## Chemical analysis of metabolites

To monitor the dynamics of metabolites in the incubation experiments, subsamples of the culture were taken periodically with sterile syringes flushed with pure N$_2$ as described above. HCl-extractable Fe(II) was determined by adding 0.1 ml sample aliquots to 0.2 ml 0.75 N HCl. The sample was immediately centrifuged for 15 min at 12,044$g$. Fe(II) in the resulting 0.5 N HCl was measured using the ferrozine assay. Previous studies have shown the 0.5 N HCl treatment allowed quantitative extraction of the solid phase Fe(II) associated with the surface of iron oxides, Fe(II) from FeS, and the dissolved Fe(II) in the Fe/S system[7,94]. Therefore, we referred to HCl-extractable Fe(II) as total Fe(II).

Aqueous and total sulfide were determined using spectrophotometric methods. To measure dissolved sulfide, approximately 0.3 ml subculture was filtered through a 0.2 μm membrane (CHROMAFIL). The dissolved sulfide in the filtrate (0.1 ml) was fixed by 0.25 ml 3% w/v zinc acetate dihydrate (Sigma Aldrich), followed by quantification using the Cline method[95]. The filtered sample from the incubation with ferrihydrite and 1 mM dissolved sulfide showed black colour, indicating the formation of FeS particles smaller than 0.2 μm. The sulfide associated with this FeS fractionation was approximated as HCl-extractable Fe(II), assuming a 1:1 stoichiometry. The total sulfide was determined as Cline-extractable sulfide. The Cline reagent contains 6 N HCl that dissolves some solid sulfides (for example, freshly formed FeS), and thus the Cline-extractable sulfide comprises dissolved sulfide and HCl-reactive solid phase sulfide. Total sulfide in the Fe/S system is typically determined as acid volatile sulfide. Acid volatile sulfide was not analysed here owing to the large uncertainties inherent to this methodology[91,96].

Sulfate and formate concentrations in the incubations were determined by capillary electrophoresis techniques. Sample preparation for sulfate measurement involved fixation of 100 μl subsample with 10 μl 3% w/v zinc acetate, dilution with 890 ul MilliQ water, filtration

through a 0.2 μm membrane, and addition of 1 mM chlorate as the internal standard. The standards were prepared by adding defined amounts of sulfate (Sigma Aldrich) to the alkaline medium, followed by the same treatment procedure as described for samples. The sulfate content in the prepared samples/standards was measured using an Agilent 7100 capillary electrophoresis system (Agilent Technologies), equipped with a capillary (72 cm × 72 μm internal diameter; Agilent Technologies) and a diode array UV-vis detector (DAD). Electrolytes for anion separation contains 2.25 mM pyromellitic acid (Sigma Aldrich), 1.6 mM triethanolamine (Sigma Aldrich), 0.75 mM hexamethonium hydroxide (Sigma Aldrich), and 6.5 mM NaOH[97] at pH 7.8 ± 0.1. Anion separation was implemented at a voltage of −30 kV. The data were acquired through indirect UV detection at a wavelength of 350 nm with a bandwidth of 60 nm, and a reference wavelength of 245 nm with a bandwidth of 10 nm. For the formate measurement, 900 μl MilliQ water was added to 100 μl samples/standards (Sigma Aldrich), which were then filtered through a 0.2 μm membrane. L-malate (Sigma Aldrich) was added to the filtrate as the internal standard. Organic Acids Buffer for capillary electrophoresis (pH 5.6; Agilent Technologies) was used as electrolytes, and the separation conditions, including DAD and capillary electrophoresis settings, were configured according to manufacturer's instructions. All electropherogram data were analysed with the Agilent ChemStation.

Elemental sulfur was measured using high performance liquid chromatography (HPLC). One-hundred microlitres of sample was fixed with 10 μl of 3% w/v zinc acetate. Then, 300 μl chloroform was added, and the mixture was shaken at 500 rpm for 1 h. The elemental sulfur in chloroform phase was then measured using a Dionex UltiMate 3000 UPLC system, equipped with an UltiMate 3000 pump (0.2 ml min$^{-1}$), a column Compartment (25 °C), a column Waters ACCQ-TAG ULTRA C18 1.7 μm × 2.1 × 100 mm, and an UltiMate 3000 Variable Wavelength Detector (UV) (wavelength 254 nm). The isocratic elution with 100% methanol was applied. With these adjustments, the peak appeared after 3.4 min. Data were analysed with Dionex Chromeleon software.

## Microscopy of *D. alkaliphilus* incubated with ferrihydrite and sulfide

For scanning electron microscopy (SEM), transmission electron microscopy (TEM) and fluorescence microscopy, cultures incubated with ferrihydrite and sulfide (daily spike of 1 mM) for 5 days were fixed in 2% glutaraldehyde or 2.3% formaldehyde, respectively. For SEM imaging, solid iron phase iron was allowed to settle without centrifugation, carefully washed with MilliQ water, and transferred to 100% ethanol. Samples were then dried using rapid chemical drying with hexamethyldisilazane and mounted on aluminium stubs with double-sided sticky carbon tape and sputtered with Gold (JEOL JFC-2300HR). The images were taken with a Scanning Electron Microscope (JEOL IT 300 LAB6EOL) with Secondary Electron Detector (SED) and Backscattered Electron Detector (BED-C) at 20 kV.

For TEM imaging, cultures were treated with a solution containing 50 g l$^{-1}$ sodium dithionite, 0.2 M sodium citrate and 0.35 M acetic acid (hereafter termed dithionite solution) as previously described[98]. After dissolution of solid iron phase, cells were pelleted at low speed (2,300$g$) to minimize shear forces and washed with MilliQ water before suspending cells in MilliQ water. For negative staining, 4 μl of sample was incubated for 1 min on a formvar-filmed and carbon-coated grid (200 mesh, Cu) and excess liquid was removed with a filter paper. A drop of stain (2.5% gadolinium acetate) was applied and immediately removed. Samples were examined in a TEM EM 900 N (Zeiss) at 80 kV.

For fluorescence microscopy, the formaldehyde-fixed cultures were resuspended and a subsample was filtered onto a 0.2 μm pore size polycarbonate membrane (Millipore). Cells on the filter were stained with a 1× SYBR Green solution, and images were acquired using a epifluorescence microscope (Zeiss Axio Imager M1 with an AxioCam MRm).

## Monitoring the growth of *D. alkaliphilus* during incubation experiments

Growth was monitored by cell counting for cultures incubated under 4 different conditions: (1) ferrihydrite (approximately 62 mM Fe equivalent) and periodic spike of approximately 50 µM dissolved sulfide (sulfide was spiked 40 times over 5 days, with one spike every hour and 8 times per day); (2) ferrihydrite (approximately 62 mM Fe equivalent) and daily spike of 1 mM sulfide; (3) ferrihydrite (approximately 62 mM Fe equivalent) and periodic spike of FeS (approximately 1 mM Fe equivalent); and (4) nitrate (4 mM) and 2 spikes of sulfide at concentration of 1–2 mM. The setup of the cultures and controls was the same as described in 'Cultivation of *alkaliphilus* DSM 19089' except that a lower starting cell density ($3–5 \times 10^7$ cells per ml) was used. During each of the incubation experiments, subcultures (450 µl) were sampled periodically and preserved in 2.3% formaldehyde (final concentration). Before counting, 500 µl dithionite solution was added to 50–100 µl of fixed cells to dissolve the FeS and ferrihydrite particles. After dissolution of solid iron phase (within 10–15 min), 100 µl of each sample was diluted in 900 µl of 1× phosphate-buffered saline (PBS). The suspension was then sonicated using a SONOPULS ultrasonic homogenizer (Bandelin, Berlin, Germany) at 25% power with a cycle setting of 2 for a total of 30 s. Cells were subsequently stained with SYBR Green 1× (ThermoFisher) and incubated for 10 min at room temperature in the dark. Flow cytometric analysis was performed using a CytoFLEX S flow cytometer (Beckman Coulter) equipped with a blue 488 nm laser. SYBR Green fluorescence was detected using a 525/40 nm bandpass filter. A fluorescence threshold was applied on the SYBR Green signal to exclude background events. For each sample, 80–100 µl was measured. Data were gated on SYBR Green–positive cells displaying fluorescence shifts relative to unstained controls to identify the target population (Supplementary Fig. 15). Data were acquired and analysed with the CytExpert 2.6 software (Beckman Coulter). The specific growth rate ($k$; day$^{-1}$) was estimated via linear regression analysis of $\ln(\text{Cell}_t/\text{Cell}_0)$ versus time (day) over an apparent exponential growth phase. Here, $\text{Cell}_t$ is the cell concentration (in cells per ml) at sampling time $t$ (day).

## $^{13}$C-bicarbonate labelling experiments and isotope analysis

To probe for autotrophic carbon fixation during MISO growth conditions, $^{13}$C-labelled bicarbonate (98 atom% $^{13}$C; Sigma Aldrich) was added to ferrihydrite-incubated cultures receiving dissolved sulfide (1 mM) or FeS (ca. 1 mM S equivalent), to reach a 10 atom% $^{13}$C in the inorganic carbon pool. The dissolved sulfide or solid phase FeS were spiked in the same frequency as for the growth experiment. Abiotic controls for each culture were set up using autoclaved inoculum. To detect $^{13}$C content in bulk biomass and in single cells, subcultures were sampled, fixed by formaldehyde (2.3% final concentration), and analysed using elemental analyser-isotope ratio mass spectrometry (EA-IRMS) and NanoSIMS. For EA-IRMS, 1.5 ml of fixed samples that included ferrihydrite and cells were pelleted by centrifugation and washed with MilliQ water, followed by overnight treatment by 0.1 M HCl to remove residual carbonates. The dried cells attached to ferrihydrite particles were weighed (4–6 mg) and transferred to tin cups. Bulk cell carbon isotope ratios ($^{13}$C:$^{12}$C) were measured by EA-IRMS (Delta V Advantage) coupled by a ConFlo IV interface to an elemental analyser (EA-Isolink, all Thermo Finnigan). Sample $^{13}$C contents were calculated as atomic percentage of $^{13}$C in total carbon, following $^{13}$C atom% = $^{13}$C/($^{13}$C + $^{12}$C) × 100%. The analytical precision of replicate analyses of isotopically homogeneous international standards was ±0.0001% for $^{13}$C atom% measurements.

For NanoSIMS analysis, 0.1 ml formaldehyde-fixed samples that included ferrihydrite and cells were mixed with dithionite solution as described above and incubated for 2 h. After complete dissolution of ferrihydrite, 400 µl of the suspension was transferred onto gold-coated polycarbonate filters (GTTP type, 0.2 µm pore size, Millipore). The filters were gold-coated by physical vapour deposition, utilizing an Agar B7340 sputter coater (Agar Scientific) equipped with an Agar B7348 film thickness monitor (Agar Scientific) for precise adjustment of the coating thickness (150 nm). The filters were incubated for 2 h in 0.1 M HCl to remove residual carbonates and then washed twice in MilliQ water and then air-dried. Filter sections were attached to antimony-doped silicon wafers (7.1 ×7.1 mm, Active Business Company) with a commercially available glue (SuperGlue Loctide).

NanoSIMS measurements were carried out on a NanoSIMS 50 L instrument (Cameca) at the Large-Instrument Facility for Environmental and Isotope Mass Spectrometry at the University of Vienna. Prior to data acquisition, analysis areas were pre-conditioned in situ by rastering a high-intensity, slightly defocused Cs$^+$ ion beam for removal of surface adsorbates and establishment of the steady state secondary ion signal intensity regime with minimum sample erosion. For this purpose, the following sequence of high and extreme low Cs$^+$ ion impact energy (EXLIE) was applied: high energy (16 keV) at 100 pA beam current to a fluence of $5 \times 10^{14}$ ions cm$^{-2}$; EXLIE (50 eV) at 400 pA beam current to a fluence of $5 \times 10^{16}$ ions cm$^{-2}$; high energy to an additional fluence of $2.5 \times 10^{14}$ ions cm$^{-2}$. Data were acquired as multilayer image stacks by scanning of a finely focused Cs$^+$ primary ion beam with 2 pA beam current at approximately 80 nm physical resolution (probe size) over areas between $60 \times 60$ and $62 \times 62$ µm$^2$ with $512 \times 512$ pixel and $1,024 \times 1,024$ pixel image resolution and a per-pixel dwell time of 5 ms and 1.5 ms, respectively. The detectors of the multicollection assembly were positioned for parallel detection of $^{12}$C$_2^-$, $^{12}$C$^{13}$C$^-$, $^{12}$C$^{14}$N$^-$, $^{31}$P$^-$ and $^{32}$S-secondary ions. Secondary electrons were detected simultaneously for gaining information about the sample morphology and topography. The mass spectrometer was tuned to achieve a mass resolving power ((MRP) = $M/\Delta M$) of >10,000 for detection of C$_2^-$ secondary ions.

Measurement data were processed using the WinImage software package provided by Cameca (WinImage V4.8) and the OpenMIMS plugin in the image processing package ImageJ (V1.54p). Prior to data evaluation, images were corrected for detector dead-time and positional variations emerging from primary ion beam and/or sample stage drift. Carbon isotope composition images displaying the $^{13}$C/($^{12}$C + $^{13}$C) isotope fraction, given in atom percent (atom%), were inferred from the C$_2^-$ secondary ion signal intensity distribution images via per-pixel calculation of $^{13}$C$^{12}$C$^-$/($2 \times ^{12}$C$_2^-$ + $^{12}$C$^{13}$C$^-$) intensity ratios. For numerical data evaluation, regions of interest, referring to individual cells, were manually defined on the basis of the $^{12}$C$^{14}$N$^-$ and $^{31}$P$^-$ secondary ion maps as indicators of biomass and verified by the topographical/morphological appearance in the secondary electron images. Biomass aggregates, in which an unambiguous identification of single cells was not feasible, were rejected.

Cells were assessed as being significantly enriched in $^{13}$C after incubation in the presence of $^{13}$C-bicarbonate if (1) the $^{13}$C isotope fraction value was higher than the mean plus 3 standard deviations ($\sigma$) of the values determined on the cells from the control (on day 0) and (2) the statistical counting error ($3\sigma$, Poisson) was smaller than the difference between the considered $^{13}$C enriched cell and the mean value measured on the cells from the control. The Poisson error was calculated from the secondary ion signal intensities (given in counts per region of interest) via

$$\sigma_{\text{Pois}} = 1/\left(2 \times {}^{12}\text{C}_2^- + {}^{12}\text{C}{}^{13}\text{C}^-\right)^2$$
$$\times \sqrt{\left(\left({}^{12}\text{C}_2^-\right)^2 \times {}^{12}\text{C}{}^{13}\text{C}^- + \left({}^{12}\text{C}{}^{13}\text{C}^-\right)^2 \times {}^{12}\text{C}_2^-\right)}$$

On the basis of these two criteria, all individual cells measured in the $^{13}$C incubated sample showed a significant enrichment in $^{13}$C.

### RNA-seq and transcriptomics

*D. alkaliphilus* cultures grown under five incubation conditions (as described in 'Cultivation of *D. alkaliphilus* DSM 19089'), each in four replicates, were used for comparative transcriptomic analysis. Cultures (30 ml) showing metabolic activity (for example, Fe(II) production, sulfide consumption or production) were collected in the middle to late stage of incubation experiments. Cells were collected by centrifuging (12,857$g$; room temperature) under anoxic condition using oak ridge tubes (Thermo Fisher Nalgene) with replacement O-rings for sealing cap (Thermo Fisher Nalgene). The cell pellets were resuspended with 1.5 ml supernatant, and distributed to three lysis matrix E tubes (MP Biomedicals), each with approximately 0.5 ml. The collected cells were immediately frozen with liquid $N_2$, and stored at −80 °C before subsequent analysis. The total nucleic acids were extracted following a phenol-chloroform protocol as described previously[99,100]. In brief, the sample was lysed for 30 s at a speed of 5.5 m s$^{-1}$, after mixing with hexadecyltrimethylammonium bromide extraction buffer and phenol-chloroform-isoamyl alcohol (25:24:1) (pH 8.0). The aqueous phase was extracted by centrifugation, and the phenol within was removed by mixing with chloroform-isoamyl alcohol (24:1). The total nucleic acids in the aqueous phase were then precipitated with polyethylene glycol 6000, followed by centrifugation. The pelleted nucleic acids were washed with ice-cold ethanol and dried before resuspension in diethyl pyrocarbonate-treated water. DNA from the total nucleic acids were removed using the TURBO DNA-free kit (Thermo Fisher Scientific).

RNA-sequencing was performed at the Joint Microbiome Facility of the Medical University of Vienna and the University of Vienna (JMF) under project IDs JMF-2311-14 and JMF-2405-05. Sequencing libraries were prepared from rRNA depleted (Ribo-Zero Plus rRNA Depletion Kit, Illumina) RNA samples (NEBNext Ultra II Directional RNA Library Prep Kit for Illumina, New England Biolabs) and sequenced in 2× 100 bp paired-end mode (NextSeq 6000, Illumina), yielding 74.2–303.7 million raw reads per sample. Individual read libraries were quality checked using fastQC v0.12.1 (http://www.bioinformatics.babraham.ac.uk/projects/fastqc/) and quality statistics were merged using multiQC v.1.21 (ref. 101). Adapters were trimmed and phiX contamination was removed using BBDuk (part of BBMap v39.06). Reads were k-trimmed from the right with a kmer of 21, minimum kmer of 11 and hamming distance of two along with the tpe and tbo options. Quality trimming was performed from the right with a $q$-score of 28 to a minimum of 50 bases in length (https://sourceforge.net/projects/bbmap/). The quality filtered reads were aligned to the reference genome of *D. alkaliphilus* (NC_014216.1) using BBMap with a mapping identity of 99% and with ambiguous reads assigned to the best location (that is, counted only once for duplicated genes). FeatureCounts (part of SubRead 2.0.6 (ref. 102)) with reverse-stranded and −countReadPairs were used to generate counts tables with the resulting alignments based on gene call locations by prodigal v2.6.3 (ref. 103). Counts tables were analysed using DESeq2 release 3.19 (ref. 104) to calculate the RPKM and to determine statistical significance of differential transcription between treatment groups. All $P$ values are adjusted for multiple comparisons using the Benjamini–Hochberg method[105].

Quantitative PCR with reverse transcription (RT–qPCR) was performed to verify the upregulated transcription for the MHC gene DA_402 under iron-reducing conditions. Primers DA_402_998F (5′-TTCCCAATCGGGGCGAATAC-3′) and DA_402_1081R (5′-TGGCCTCG GTATAGAGGGTC-3′) were used to target DA_402. Primers recA_79F (5′-TTCGGCAAAGGCTCCATCAT-3′) and recA_221R (5′-TCCGGCCCA TATACCTCGAT-3′) were used to quantify the transcription level of the house-keeping gene *recA* (DA_1926) encoding the DNA recombination protein. Primers for both genes were newly designed using Primer-Blast[106]. For RT–qPCR, DNA-free RNA was first reverse transcribed to cDNA using SuperScript III reverse transcriptase according to the manufacturer's instructions. The absolute abundance of transcripts from DA_402 and *recA* were quantified by quantitative PCR using cDNA as a template. Purified PCR products of gene DA_402 and *recA* amplified from genomic DNA of *D. alkaliphilus* were used as quantitative PCR standards. The PCR reactions were prepared in triplicates and run at 95 °C for 3 min, followed by 40 cycles of 95 °C for 30 s, 60 °C for 30 s, and 72 °C for 45 s, on the Thermal Cycler with CFX96 Real-Time System (Bio-Rad). The RT–qPCR data were acquired and analysed using CFX Maestro software (Bio-Rad). The transcription level of DA_402 was compared between treatments after normalization with that of *recA*. The statistical significance of differential transcription between treatments were determined via Student's *t*-test.

### Structure prediction and phylogenetic analysis of multi-haem c-type cytochromes in *D. alkaliphilus*

*D. alkaliphilus* proteins with more than one haem-binding motifs ($CX_nCH$; $n = 2$ to 5) were considered MHCs[107]. The haem-binding motifs in protein sequences were counted using regex expressions in the Python re package. The subcellular localization of all putative MHCs ($n = 46$) from *D. alkaliphilus* was predicted using PSORTb v3.0 (ref. 108) and DeepLocPro v1.0 (ref. 81). Prediction from DeepLocPro was used for the proteins for which PSORTb returned 'Unknown'. The transcription levels of MHCs were compared between different incubation experiments on the basis of RPKM values. The statistical significance of differential transcription was assessed as described in the 'RNA-Seq and transcriptomics' chapter. The most highly transcribed extracellular MHC (DA_402) during MISO was further selected for structure prediction and phylogenetic analysis. The structure of the DA_402 monomer and oligomer were predicted using AlphaFold2 v2.3.2 at the COSMIC$^2$ science gateway. The leading signal peptide, predicted using SignalP 5.0 (ref. 109), was cleaved from the protein sequence before structure modelling. For comparison, the cryo-EM structure of OmcS from *G. sulfurreducens* was retrieved from the Protein Data Bank (PDB) database (6EF8). The protein sequence of DA_402 was aligned to OmcS using the T_coffee alignment tool[110]. The structure-structure similarity between DA_402 and OmcS was calculated using an online TM-align tool and DaliLite.v5 (ref. 111–113). The haem-binding sites in DA_402 and OmcS were visualized using MacPyMOL v.1.7.4 (https://pymol.org). The haem was docked to the target haem-binding site in DA_402 using AutoDockTools v1.5.7 (ref. 114) and AutoDock Vina 1.1.2 (ref. 115). To conduct phylogenetic analysis of DA_402, homologues of DA_402 were retrieved from the KEGG database using Blastp with an E value of 0.01. The retrieved homologues were then de-replicated with CD-HIT at 70% identity, aligned with Muscle, trimmed with trimAl (--gt 0.1). The resulting sequence alignment was used to reconstruct the maximum-likelihood tree using RAxML v8.2.12. The clustering pattern and decoration of the tree were performed using iTOL v6 (ref. 116).

### Environmental distribution of *Desulfurivibrionaceae* with genomic potential of MISO

The metabolic potential of members belonging to the *Desulfurivibrionaceae* family was analysed using publicly available genomes recovered from different environments. Metagenome-assembled genomes (MAGs) classified as *Desulfurivibrionaceae* were obtained from GTDB r214 ($n = 121$), NCBI ($n = 9$), JGI IMG ($n = 68$) and GMGC ($n = 7$). The environmental origins of these genomes were retrieved from the metadata in the respective databases (Supplementary Table 8). The taxonomy of collected genomes was assigned using GTDB-tk version 2.3.2 with database release 214 (ref. 117). The phylogenomic tree of *Desulfurivibrionaceae* was reconstructed from a concatenated alignment of 120 single-copy genes with FastTree v2.1.10 (ref. 66). The protein-coding genes in the genomes were called using Prodigal v2.6.3 (ref. 103), and the resulting proteomes were screened for proteins involved in dissimilatory sulfide oxidation (DsrAB) and iron oxides reduction (that is, OmcS, OmcZ, porin–cytochrome complex and OmcE). DsrAB was

detected using HMMs and the phylogenetic framework established in this study, while proteins involved in dissimilatory iron reduction were identified with HMMs from FeGenie[86]. Additional proteins likely involved in dissimilatory reduction of iron oxides—that is, extracellular MHC DA_402 and PilA—were retrieved from *Desulfurivibrionaceae* genomes by hmmsearch or BLASTP. Homologues of PilA were extracted by searching TIGR02532 HMM model against the *Desulfurivibrionaceae* genomes using hmmsearch with --cut_ga option. For DA_402, homologues were collected from the *Desulfurivibrionaceae* genomes using BLASTP with an E value of 1e-10, followed by prediction of the subcellular localization and counting of haem-binding sites. The extracellular homologues containing multi-haem-binding sites ($n > 3$) were then placed into a reference tree created through phylogenetic analyses of DA_402 (see above) with the RAxML evolutionary placement algorithm (EPA). The alignment for EPA was generated using MAFFT v7.407 with --add option. The homologues that were placed with accumulated probability over 0.95 to the OmcS-like clade were considered as functional orthologs of DA_402. For visualization purposes, *Desulfurivibrionaceae* genomes ($n = 119$) encoding both dissimilatory iron and sulfur metabolism were de-replicated on the basis of relative evolutionary divergence (RED). RED was calculated for each internal node of the *Desulfurivibrionaceae* phylogenomic tree following the procedure described previously[118]. The tree was then collapsed at a RED value of 0.9 and one representative was chosen randomly from the collapsed clades, yielding 53 representative members that were visualized in the tree.

## Statistics and reproducibility

The physiological experiments showing the ability of ferrihydrite-incubated *D. alkaliphilus* to oxidize formate (Fig. 3a), FeS (Fig. 3b), 1 mM sulfide (Fig. 3c,d,g) or ~50 μM sulfide (Fig. 3h) were repeated independently at least three times, all yielding consistent results. The sulfide removal kinetic experiments at low sulfide concentration were replicated independently for two times, and all results are present in the Extended Data Fig. 1. The experiment showing the transformation of sulfide and ferrihydrite with periodic supply of ~50 μM sulfide was performed independently twice, and both yielded similar results. The growth experiments of ferrihydrite-incubated cells with periodic addition of 1 mM sulfide (Fig. 4a), FeS (Fig. 4b) or 50 μM sulfide (Fig. 4c), and the experiment with nitrate and sulfide (Fig. 4d) were conducted once with three biological replicates per treatment and control. The [13]C-bicarbonate labelling experiment and bulk [13]C analysis of cells incubated with ferrihydrite and either dissolved sulfide (Fig. 4e) or FeS (Fig. 4f) were independently repeated for two times, both yielding comparable outcomes. Samples for NanoSISM analysis were chosen randomly among biological replicates collected at day 0 and 5, and representative field views are present in Fig. 4g–i. Transcriptomic analysis of cells growing under five different conditions was conducted once, with four biological replicates per condition (Fig. 5 and Extended Data Fig. 4).

## Reporting summary

Further information on research design is available in the Nature Portfolio Reporting Summary linked to this article.

## Data availability

The GTDB r95 genomes were retrieved from the GTDB repository (https://data.ace.uq.edu.au/public/gtdb/data/releases/release95/95.0/genomic_files_reps/). The protein structure of OmcS from *G. sulfurreducens* was retrieved from the PDB accession 6EF8. Reference sequences and gene-specific HMMs mentioned in the Methods were acquired from SwissProt (https://www.uniprot.org/), KEGG (https://www.kegg.jp/), PFAM (http://pfam.xfam.org/), EggNOG v5.0 (http://eggnog5.embl.de/#/app/home), Sandpiper (https://sandpiper.qut.edu.au/), dbCAN (https://bcb.unl.edu/dbCAN2/download/), metabolisHMM

(https://github.com/elizabethmcd/metabolisHMM), CANT-HYD (https://github.com/dgittins/CANT-HYD-HydrocarbonBiodegradation), MicRhoDE (http://application.sb-roscoff.fr/micrhode/), FeGeneie (https://github.com/Arkadiy-Garber/FeGenie), NCBI (https://www.ncbi.nlm.nih.gov/) and JGI IMG (https://img.jgi.doe.gov/). The transcriptomic data of *D. alkaliphilus* cultured under five conditions (each with four replicates) have been deposited at the NCBI under BioProject PRJNA1165744 (NCBI Sequence Read Archive (SRA) accession: SRX26208148–SRX26208167). HMM models for the 116 sulfur-cycling genes have been deposited at Github (https://github.com/SongCanChen11/HMMs). Source data are provided with this paper.

## Code availability

A custom script to retrieve homologues of sulfur-cycling marker genes from GTDB database has been deposited at Github (https://github.com/SongCanChen11/HMMs/).

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

**Acknowledgements** The authors thank B. Hausmann, K. Wasmund and T. S. Tanabe for help with selection of sulfur-cycling target genes; M. Schmid for Raman analyses of ferrihydrite; J. Ramesmayer for preparation of cDNA libraries; L. Seidl for technical assistance with elemental sulfur analysis; M. Watzka for isotope ratio mass spectrometry analysis; N. Kumar for support in ferrihydrite synthesis; O. Coruh for help with protein structure modelling; K. Schmidt, S. Dürr and D. Gruber for SEM and TEM imaging; T. Rattei and his team for maintaining and providing access to the Life Science Compute Cluster (https://lisc.univie.ac.at/); the Bundesministerium für Bildung und Forschung (BMBF)-funded deNBI cloud within German Network for Bioinformatics Infrastructure (de.NBI) (no. 031A532B, 031A533A, 031A533B, 031A534A, 031A535A, 031A537A, 031A537B, 031A537C, 031A537D and 031A538A) for providing computational resources; and H. Daims for valuable discussion. The drawing of pili was inspired by figures in Cai et al. (2021)[119] and Melville & Craig (2013)[120]. This research was funded by the Austrian Science Fund (FWF) (grants https://doi.org/10.55776/P31996 and https://doi.org/10.55776/COE7), the EU MSCA postdoctoral fellowship (action number 101059607, DatingSuCy) to S.-C.C., the National Natural Science Foundation of China (grants 42307167 and 42021005), and a Chinese Scholarship Council fellowship to X.-M.L. For open access purposes, the author has applied a CC BY public copyright license to any author accepted manuscript version arising from this submission.

**Author contributions** S.-C.C., M.M. and A.L. conceived the study and planned experiments, with help from P.P. and S.P. S.-C.C. performed bioinformatic analyses. S.-C.C., X.-M.L., N.B., G.G. and M.A.M. designed and performed physiological experiments. S.-C.C., N.B., M.M. and M.A.M. quantified cell growth and metabolite concentrations. W.W. contributed to elemental sulfur and isotope ratio mass spectrometry analysis. A.S. performed NanoSIMS analysis. S.-C.C. and M.A.M. extracted RNA from cultures incubated under different conditions. S.-C.C., J.O. and P.P. performed comparative transcriptome analyses. W.W. and A.R. provided essential infrastructure for metabolite and isotope analyses. S.-C.C. analysed the data. S.-C.C., M.M. and A.L. interpreted the data. S.-C.C. and A.L. wrote the paper, with comments from all other authors.

**Funding** Open access funding provided by University of Vienna.

**Competing interests** The authors declare no competing interests.

**Additional information**
**Correspondence and requests for materials** should be addressed to Song-Can Chen, Marc Mussmann or Alexander Loy.

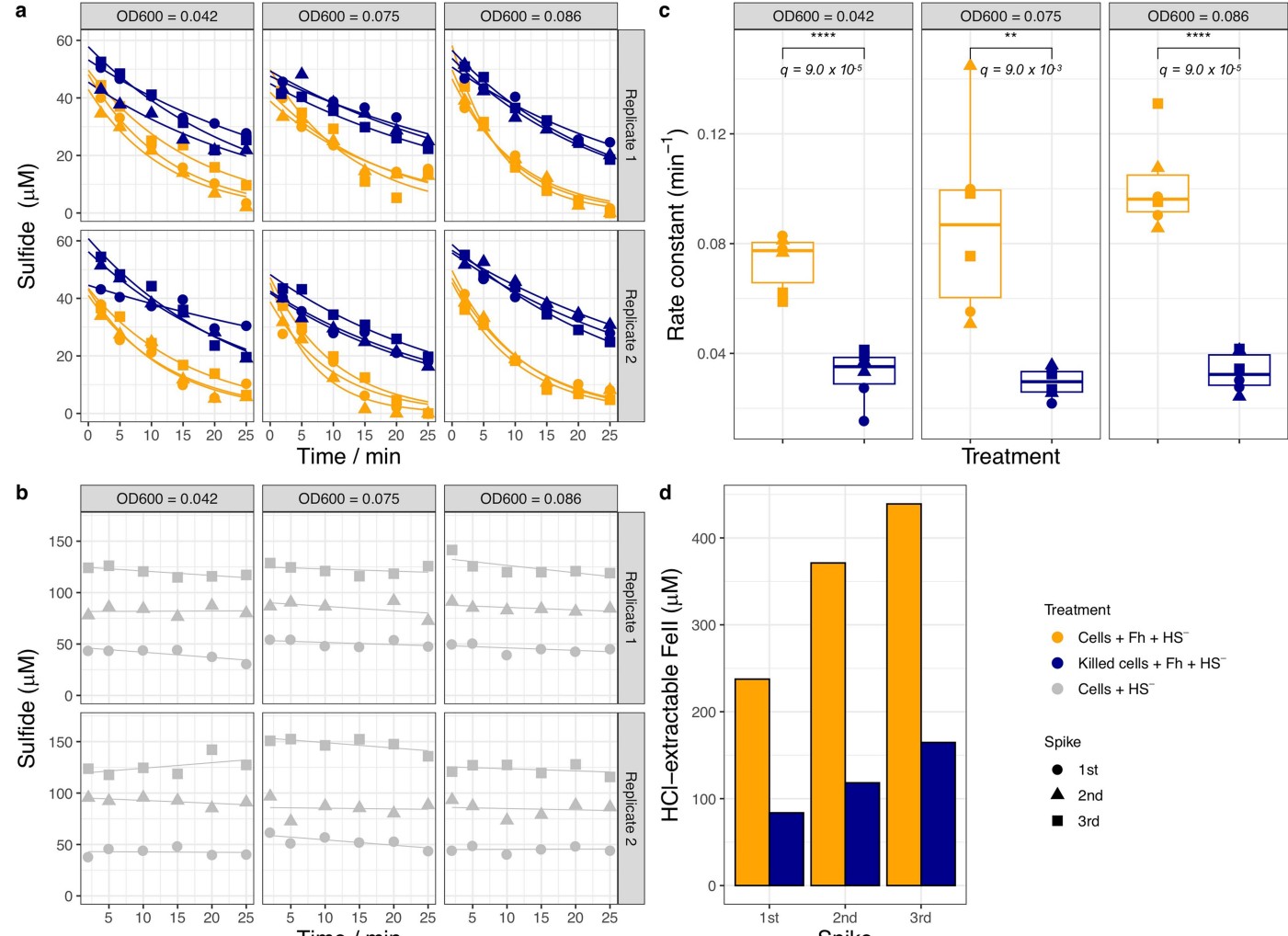

**Extended Data Fig. 1 | Sulfide removal rates at low concentration of sulfide (ca. 50 μM) are higher in ferrihydrite-amended *D. alkaliphilus* cultures than in abiotic controls. a**, Kinetics of dissolved sulfide during 25-min incubation with ferrihydrite (ca. 62 mM) in the presence or absence of *D. alkaliphilus*. The incubation experiment was conducted independently with three different cell densities, indicated by optical density at 600 nm ($OD_{600}$) of the inoculum. Each incubation received three spikes of sulfide at 1.5-h intervals. Two replicates were performed for each condition. $OD_{600}$ of 0.042 corresponds to an average cell density of $1.38 \times 10^8$ cells ml$^{-1}$. **b**, Sulfide concentrations in the sulfide-only control of the incubation experiments remained constant. The residual sulfide carried over from the inoculum was subtracted from the measured sulfide concentration. **c**, The rate constant of sulfide consumption was significantly higher (P < 0.01; two-sided Student's t-test) in *D. alkaliphilus* cultures than in abiotic controls. The rate was determined by fitting the first-order rate equation to the measured sulfide concentration changes (see panel **a**). The center lines and box limits of the boxplot denote the median, and the 25% and 75% percentile of the estimated rate constant, respectively. The whiskers extend 1.5 times the interquartile range from the 25th and 75th percentiles. The two-sided Student's t-test between biotic (n = 6) and abiotic (n = 6) treatment was conducted for each cell density. P values were adjusted for multiple comparisons (q-values) using the Benjamini and Hochberg method. **d**, Concentration of HCl-extractable Fe(II) before three consecutive spikes of sulfide to ferrihydrite with or without *D. alkaliphilus*. The values represent the mean of duplicate culture with $OD_{600}$ of 0.086. Fe(II) concentration in both treatments increased with repeated spikes of low conditions of sulfide. The treatment with cells produced an excessive amount of Fe(II) compared to the abiotic control. Fe(II) before the 1st sulfide spike results from biological or chemical oxidation of sulfide (ca. 50 μM) carried over from the inoculum.

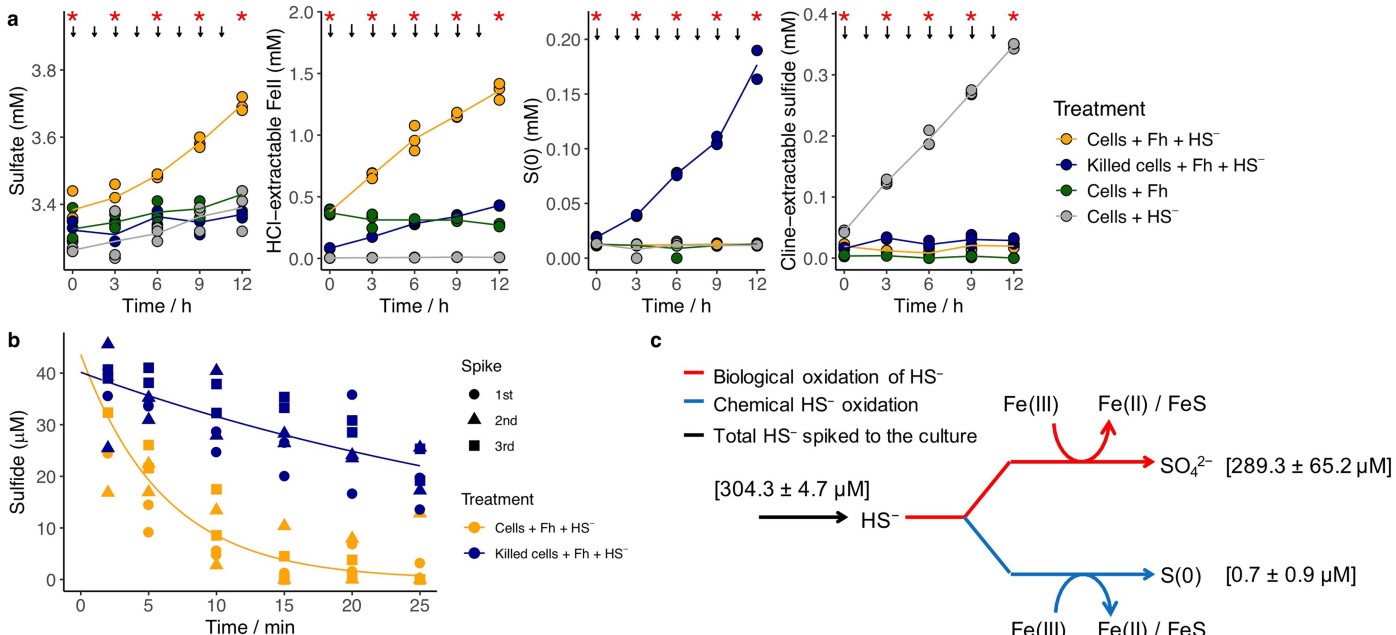

**Extended Data Fig. 2 | Transformation of sulfide during the incubation of *D. alkaliphilus* with ferrihydrite (ca. 62 mM) and periodic supply (*n* = 8) of a low concentration of sulfide (ca. 50 μM) at 1.5-h intervals. a**, Development of sulfate, HCl-extractable Fe(II), elemental sulfur (S(0)), and Cline-extractable sulfide during the incubation experiment (n = 3 replicate cultures). Parallel incubations with living cells without Fh or sulfide, and with killed cells served as controls. The arrows in each panel indicate the addition of sulfide. The red stars denote sampling from the culture for chemical analysis. The inoculum used for the incubation experiment had an $OD_{600}$ of 0.072. **b**, Kinetics of dissolved sulfide following the first three spikes to ferrihydrite-amended culture in a parallel incubation experiment with the same inoculum as in panel **a**. Sulfide measurements from two biological replicates are displayed. The data confirmed that the culture consumed sulfide more rapidly than the abiotic control. **c**, Schematic graph showing the flow of spiked sulfide during the 12-h incubation. Sulfur metabolite concentrations derive from data in panel **a**. All values are the means of triplicate cultures with standard deviation.

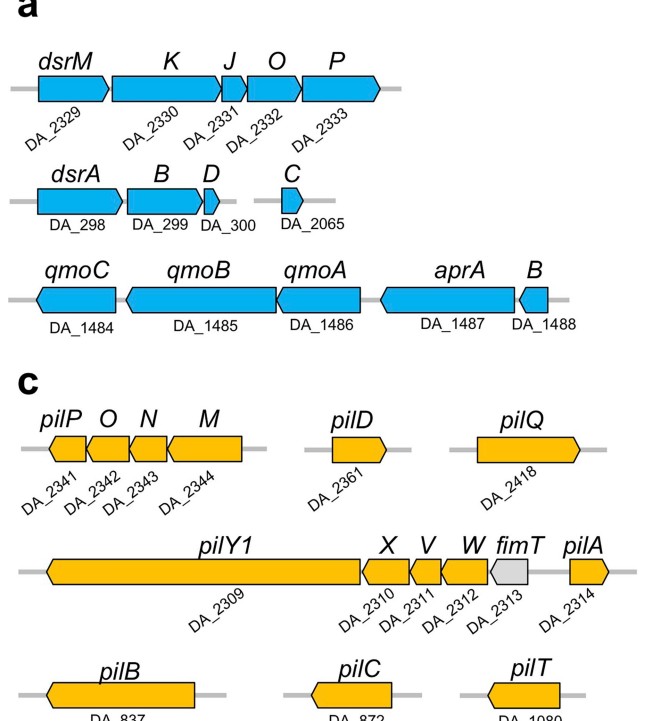

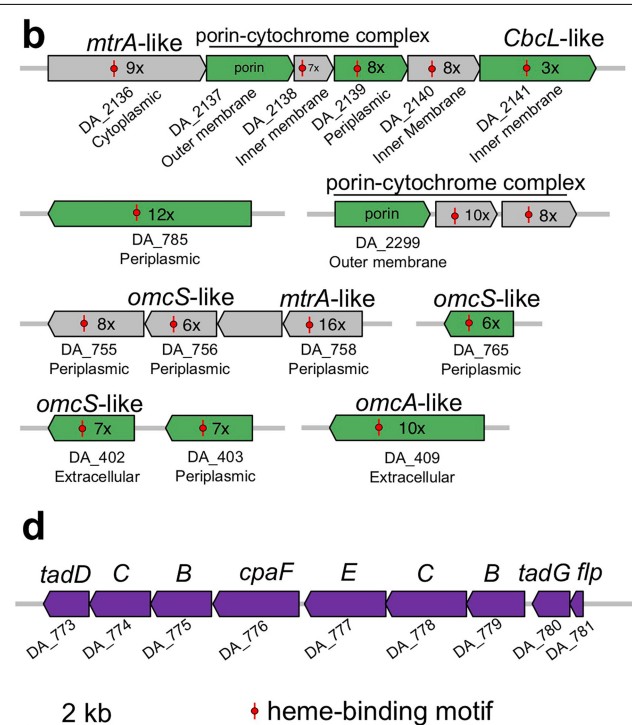

**Extended Data Fig. 3 | Genomic arrangement of genes putatively involved in MISO. a**, Genes for reversal of the canonical sulfate reduction pathway. **b**, Genes for the up-regulated multi-heme c-type cytochromes (MHCs) during MISO. **c**, Genes for the PilA-containing type IV pilus. **d**, Genes for the tight adhesion (Tad) type IV pilus. The gene name and locus tag are shown above and below the gene arrows, respectively. Count of heme-binding motifs (CX$_{2-5}$CH) is shown for each MHC in **b**. Co-localized genes not shown in Fig. 5 are colored in gray.

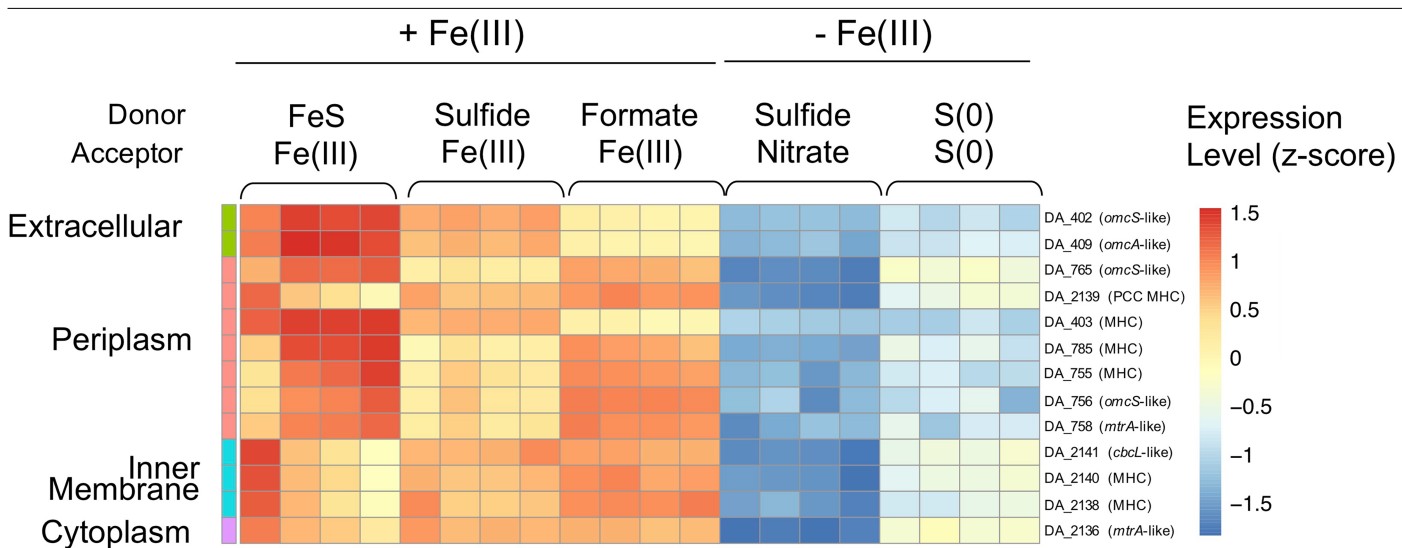

**Extended Data Fig. 4 | Up-regulated transcription of 13 multi-heme c-type cytochromes (MHCs) in *D. alkaliphilus* under ferrihydrite-amended conditions.** Comparative transcriptomics of three ferrihydrite-amended and two ferrihydrite-free growth conditions (n = 4 replicate cultures each). The heatmap is based on the z-score of reads per kilobase of transcript per million mapped reads (RPKM) after logarithm transformation. The statistical significance (P-value) of up-regulation was calculated using the DESeq2 R package, followed by adjustment of P for multiple comparison using the Benjamini and Hochberg method. The 13 MHCs showed significantly elevated relative transcription levels (adjusted P < 0.01) in all pairwise comparisons (n = 6) between ferrihydrite-amended incubations (n = 3) vs. incubations (n = 2) without ferrihydrite. Electron donors and acceptors for each of the five incubations are indicated above the heatmap. Four biological replicates of each incubation are displayed side-by-side. The subcellular localization of MHCs was predicted by Psortb v3.0 and DeepLocPro v1.0. PCC MHC, MHC from a gene cluster encoding a putative porin-cytochrome complex (PCC).

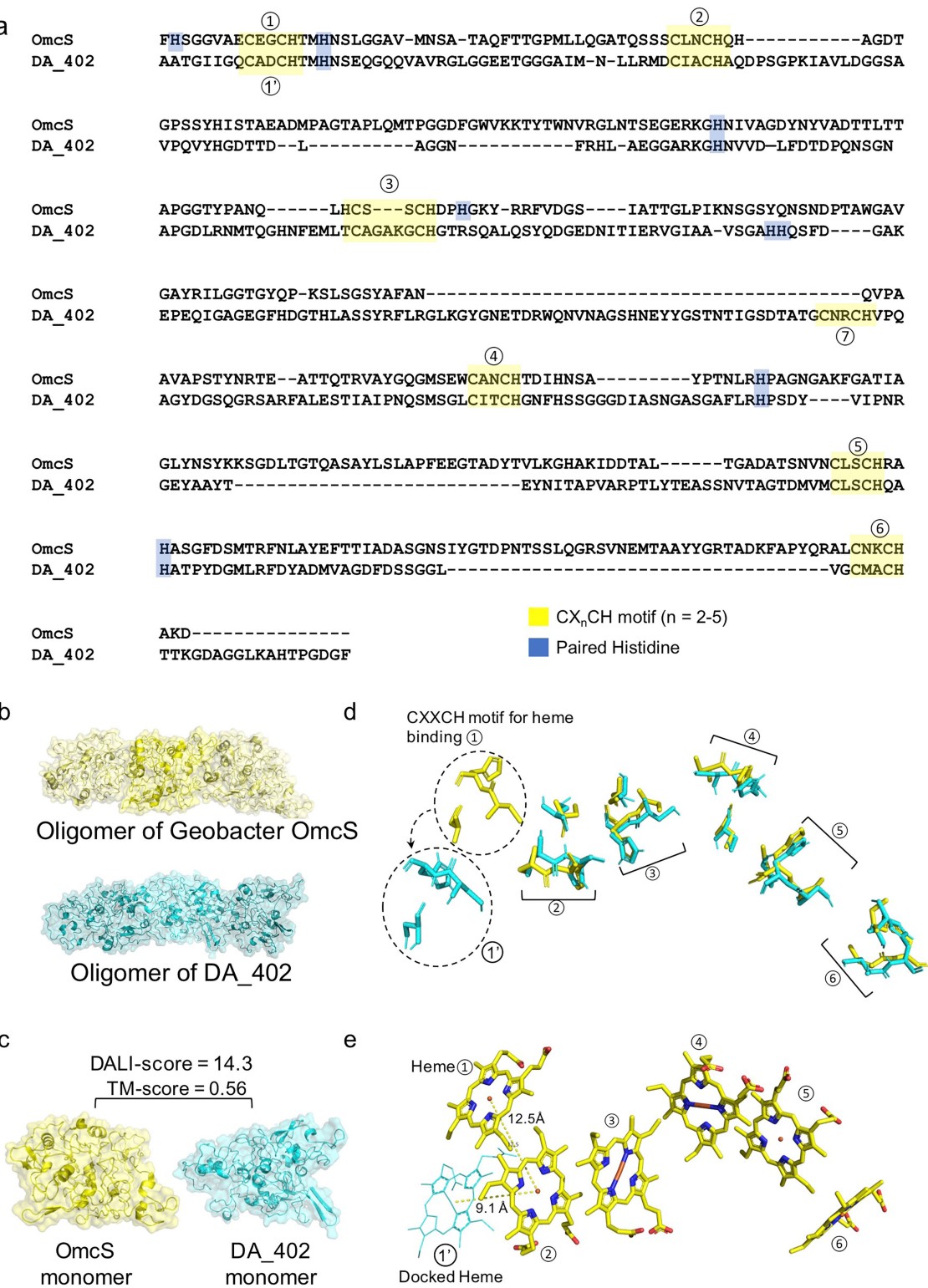

**Extended Data Fig. 5 | Sequence alignment and predicted structure of the most expressed multi-heme c-type cytochrome (DA_402) of *Desulfurivibrio alkaliphilus* during MISO. a**, Pairwise alignment between DA_402 and the OmcS from *G. sulfurreducens*. Six out of seven predicted heme-binding motifs ($CX_{2-5}CH$) are aligned with those in OmcS (labeled with numbers in circles). The seventh heme-binding motif lacks the paired histidine (blue color) for coordination of heme in the predicted structure of DA_402 and is therefore not considered in the following analysis. The leading signal peptide in DA_402 and OmcS is not shown. **b**, Predicted structure of DA_402 oligomer. DA_402 monomers are predicted to polymerize into a filament structure, similar to the cryo-EM structure of OmcS[28]. **c**, Structural comparison between DA_402 and OmcS. DA_402 shares significant topological similarity with OmcS, with a TM-score of 0.56 and DALI-score of 14.3. **d**, Spatial arrangement of six heme-binding sites in DA_402 (cyan). All heme-binding sites predicted in DA_402 align closely with those in OmcS (yellow), except the site 1' showing shifted position. **e**, Predicted heme position of the heme-binding site 1'. Molecular docking analysis placed the heme (1') in close proximity to the adjacent heme ②, with an iron-to-iron distance of 9.1 Å, comparable with the distance (12.5 Å) observed in OmcS.

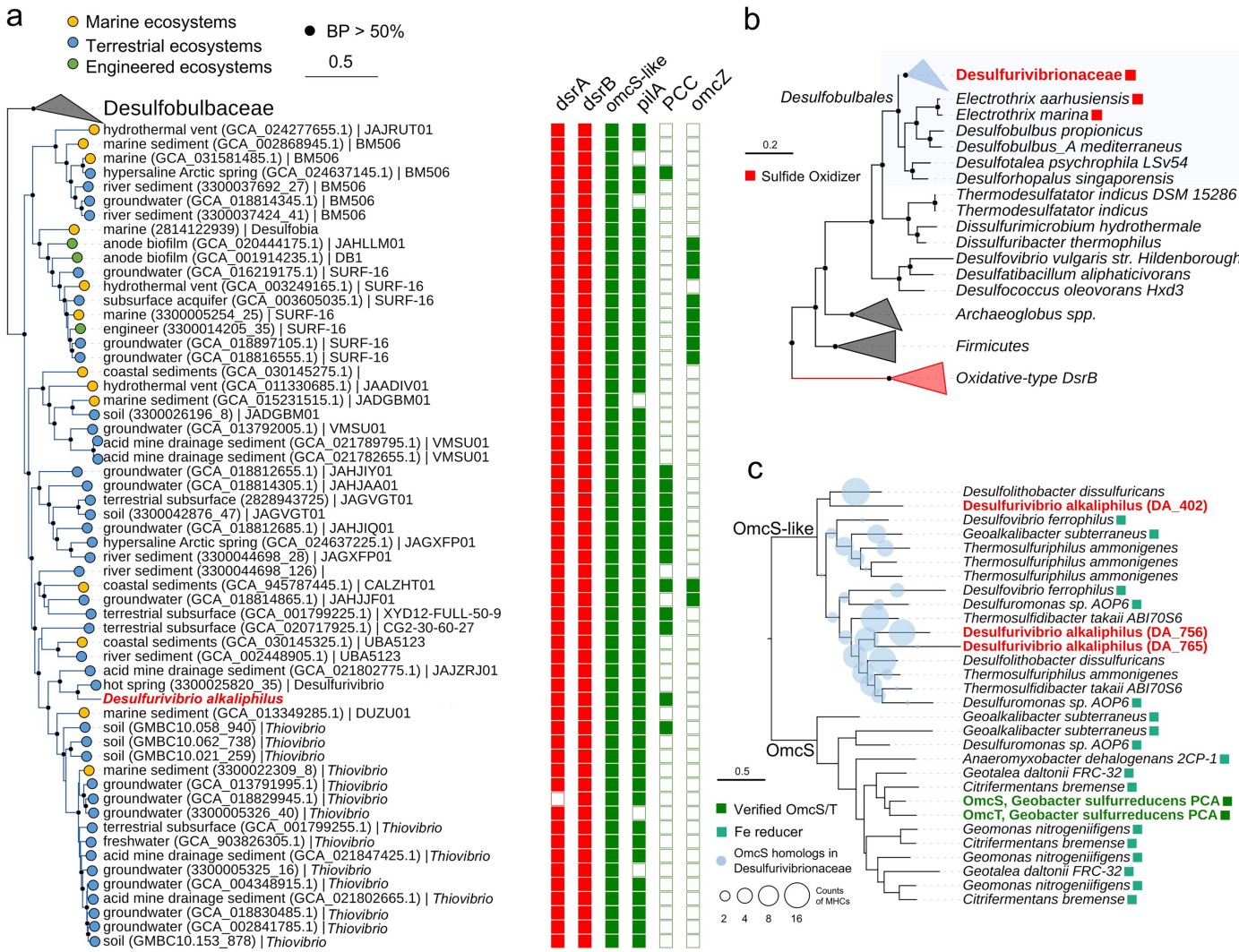

**Extended Data Fig. 6 | *Desulfurivibrionaceae* members with genomic potential for MISO are widespread across various ecosystems.**
**a**, Co-occurrence of marker proteins for dissimilatory sulfide oxidation and iron reduction in diverse *Desulfurivibrionaceae* members recovered from different ecosystems. Phylogenomic tree of *Desulfurivibrionaceae* is based on maximum-likelihood analysis of a concatenated alignment of 120 bacterial single-copy proteins. 119 of 205 *Desulfurivibrionaceae* genomes encode both dissimilatory sulfur and iron metabolisms. For visualization purposes, the phylogeny includes only 53 de-replicated genomes. *D. alkaliphilus* is shown in bold and red type. The source ecosystem of each genome is denoted by filled circles at the tips of the tree. Environmental origin, NCBI accession, and genus classification is shown at each branch tip. *Desulfobulbaceae* members were used as an outgroup. Presence and absence of marker proteins involved in sulfide oxidation (DsrAB, red squares) and extracellular respiration of iron oxides (green squares) is shown for each genome. Abbreviation: OmcS and OmcZ, two evolutionarily unrelated outer membrane cytochromes; PilA, major

component (pilin) of the type IV pilus; PCC, porin-cytochrome complex.
**b**, Phylogeny of DsrB from *Desulfurivibrionaceae* shown in **a**, which are classified as the reductive-type DsrB putatively involved in sulfide oxidation, as proposed in *D. alkaliphilus* and in cable bacteria. Known sulfide oxidizers (e.g., cable bacteria genus *Electrothrix*) encoding reductive-type DsrBs are indicated by red squares. **c**, Phylogeny of OmcS-like proteins from *Desulfurivibrionaceae* shown in **a**, which form a sister clade of OmcS. The sister clade of OmcS comprise extracellular MHCs from known iron reducers (dark green squares) and three MHCs (DA_402, DA_756, and DA_765) from *D. alkaliphilus* that are up-regulated under MISO conditions. All OmcS-like proteins shown in **a** are predicted to be extracellular multiheme c-type cytochromes (MHCs). Blue circles indicate the numbers of *Desulfurivibrionaceae* MHC genes that are placed to the tree branches using RAxML EPA algorithm. The scale bar in **a**, **b**, and **c** indicates the substitutions of amino acids per site. The branches with bootstrap values > 50 are shown with black dots in **a** and **b**.

| | |
|---|---|

# Reporting Summary

## Statistics

For all statistical analyses, confirm that the following items are present in the figure legend, table legend, main text, or Methods section.

| n/a | Confirmed | |
|---|---|---|
| ☐ | ☒ | The exact sample size (*n*) for each experimental group/condition, given as a discrete number and unit of measurement |
| ☐ | ☒ | A statement on whether measurements were taken from distinct samples or whether the same sample was measured repeatedly |
| ☐ | ☒ | The statistical test(s) used AND whether they are one- or two-sided *Only common tests should be described solely by name; describe more complex techniques in the Methods section.* |
| ☒ | ☐ | A description of all covariates tested |
| ☐ | ☒ | A description of any assumptions or corrections, such as tests of normality and adjustment for multiple comparisons |
| ☒ | ☐ | A full description of the statistical parameters including central tendency (e.g. means) or other basic estimates (e.g. regression coefficient) AND variation (e.g. standard deviation) or associated estimates of uncertainty (e.g. confidence intervals) |
| ☐ | ☒ | For null hypothesis testing, the test statistic (e.g. *F*, *t*, *r*) with confidence intervals, effect sizes, degrees of freedom and *P* value noted *Give P values as exact values whenever suitable.* |
| ☒ | ☐ | For Bayesian analysis, information on the choice of priors and Markov chain Monte Carlo settings |
| ☒ | ☐ | For hierarchical and complex designs, identification of the appropriate level for tests and full reporting of outcomes |
| ☒ | ☐ | Estimates of effect sizes (e.g. Cohen's *d*, Pearson's *r*), indicating how they were calculated |

*Our web collection on statistics for biologists contains articles on many of the points above.*

## Software and code

Policy information about availability of computer code

| Data collection | The genomic data was downloaded via FTP protocol. No specific software except Firefox Browser was used for data collection. |
|---|---|
| Data analysis | CD-HIT v4.8.1, BLAST 2.9.0+, Muscle v3.8.1551, TrimAl v1.4.rev15, FastTree v2.1.7, HMMER v3.2.1, ggtree v1.16.6 in R v3.6.3, SWISS-MODEL (https://swissmodel.expasy.org/), MacPyMOL v.1.7.4, R v4.1.0, Agilent ChemStation, Dionex Chromeleon, CFX Maestro Software for Bio-Rad CFX Real-Time PCR Systems, fastQC v0.12.1, multiQC v1.21, BBMap v39.06, SubRead 2.0.6, prodigal v2.6.3, DESeq2 release 3.19, Python v2.7 and v3.9, PSORTb v3.0, DeepLocPro v1.0, AlphaFold2 v2.3.2, SignalP 5.0, Signalp v6.0, T_coffee, AutoDockTools v1.5.7, AutoDock Vina 1.1.2, RAxML v8.2.12, iTOL v6, GTDB-tk v2.3.2, MAFFT v7.407, FeatureCounts (SubRead v2.0.6), WinImage v4.8, ImageJ v1.54p, CytExpert v2.6 (Beckman Coulter), DaliLite.v5, TM-align (https://zhanggroup.org/TM-align/). |

For manuscripts utilizing custom algorithms or software that are central to the research but not yet described in published literature, software must be made available to editors and reviewers. We strongly encourage code deposition in a community repository (e.g. GitHub). See the Nature Portfolio guidelines for submitting code & software for further information.

## Data

Policy information about availability of data

All manuscripts must include a data availability statement. This statement should provide the following information, where applicable:

- Accession codes, unique identifiers, or web links for publicly available datasets
- A description of any restrictions on data availability
- For clinical datasets or third party data, please ensure that the statement adheres to our policy

> The GTDB r95 genomes were retrieved from the GTDB repository (https://data.ace.uq.edu.au/public/gtdb/data/releases/release95/95.0/genomic_files_reps/). The protein structure of OmcS from Geobacter sulfurreducens in PDB format (6EF8) was retrieved from the PDB database (https://www.rcsb.org/structure/6EF8). Reference sequences and gene-specific HMMs mentioned in the method were acquired from following database SwissProt (https://www.uniprot.org/), KEGG (https://www.kegg.jp/), PFAM (http://pfam.xfam.org/), EggNOG v5.0 (http://eggnog5.embl.de/#/app/home), Sandpiper (http://eggnog5.embl.de/#/app/home), dbCAN (https://bcb.unl.edu/dbCAN2/download/), metabolicHMM (https://github.com/elizabethmcd/metabolisHMM), CANT-HYD (https://github.com/dgittins/CANT-HYD-HydrocarbonBiodegradation), MicRhoDE (http://application.sb-roscoff.fr/micrhode/), FeGeneie (https://github.com/Arkadiy-Garber/FeGenie), NCBI (https://www.ncbi.nlm.nih.gov/), JGI IMG (https://img.jgi.doe.gov/). The transcriptomic data of D. alkaliphilus cultured under five conditions (each with four replicates) have been deposited at the NCBI under Bioproject PRJNA1165744 (NCBI SRA accession: SRX26208148 - SRX26208167). HMM models for the 116 sulfur-cycling genes have been deposited at Github (https://github.com/SongCanChen11/HMMs).

## Research involving human participants, their data, or biological material

Policy information about studies with human participants or human data. See also policy information about sex, gender (identity/presentation), and sexual orientation and race, ethnicity and racism.

| Reporting on sex and gender | This study does not involve human participants. |
|---|---|
| Reporting on race, ethnicity, or other socially relevant groupings | This study does not involve human participants. |
| Population characteristics | This study does not involve human participants. |
| Recruitment | This study does not involve human participants. |
| Ethics oversight | Not applicable |

Note that full information on the approval of the study protocol must also be provided in the manuscript.

# Field-specific reporting

Please select the one below that is the best fit for your research. If you are not sure, read the appropriate sections before making your selection.

☒ Life sciences ☐ Behavioural & social sciences ☐ Ecological, evolutionary & environmental sciences

For a reference copy of the document with all sections, see nature.com/documents/nr-reporting-summary-flat.pdf

# Life sciences study design

All studies must disclose on these points even when the disclosure is negative.

| Sample size | No statistical methods were used to predetermine sample size. A sample size of three was chosen for most physiological experiments as it aligns with standard practices for physiological probe of pure cultures. A sample size of four was chosen for transcriptomics, as it is consistent with transcriptomics standards to ensure robust detection of differentially expressed genes while balancing cost and sample availability. |
|---|---|
| Data exclusions | No data were excluded from the analysis |
| Replication | All incubation experiments were performed with three biological replicates for each growth condition (formate+ferrihydrite, FeS+ferrihydrite, sulfide+ferrihydrite, sulfide+nitrate, S(0)). The transcriptomics were performed with four replicates for each of five incubation conditions. Incubation experiments with environmentally relevant sulfide (~50 μM) and ferrihydrite were carried out using six biological replicates as follows: three independent cultures with different cell densities, each with two replicates. The growth experiments and 13C-bicarbonate labelling experiments were conducted with three bioloigcal replicates for each of described treatments. We confirm all attempts at replication were successful. |
| Randomization | The experiments were not randomized, since all analyses concerned a single pure culture |
| Blinding | The investigators were not blinded to allocation during experiments and outcome assessment. Blinding was not performed in this study due to the visible differences between alternative incubation conditions (e.g., ferrihydrite + sulfide, nitrate + sulfide, S(0), formate+ sulfide and etc). |

# Behavioural & social sciences study design

All studies must disclose on these points even when the disclosure is negative.

| | |
|---|---|
| Study description | *Briefly describe the study type including whether data are quantitative, qualitative, or mixed-methods (e.g. qualitative cross-sectional, quantitative experimental, mixed-methods case study).* |
| Research sample | *State the research sample (e.g. Harvard university undergraduates, villagers in rural India) and provide relevant demographic information (e.g. age, sex) and indicate whether the sample is representative. Provide a rationale for the study sample chosen. For studies involving existing datasets, please describe the dataset and source.* |
| Sampling strategy | *Describe the sampling procedure (e.g. random, snowball, stratified, convenience). Describe the statistical methods that were used to predetermine sample size OR if no sample-size calculation was performed, describe how sample sizes were chosen and provide a rationale for why these sample sizes are sufficient. For qualitative data, please indicate whether data saturation was considered, and what criteria were used to decide that no further sampling was needed.* |
| Data collection | *Provide details about the data collection procedure, including the instruments or devices used to record the data (e.g. pen and paper, computer, eye tracker, video or audio equipment) whether anyone was present besides the participant(s) and the researcher, and whether the researcher was blind to experimental condition and/or the study hypothesis during data collection.* |
| Timing | *Indicate the start and stop dates of data collection. If there is a gap between collection periods, state the dates for each sample cohort.* |
| Data exclusions | *If no data were excluded from the analyses, state so OR if data were excluded, provide the exact number of exclusions and the rationale behind them, indicating whether exclusion criteria were pre-established.* |
| Non-participation | *State how many participants dropped out/declined participation and the reason(s) given OR provide response rate OR state that no participants dropped out/declined participation.* |
| Randomization | *If participants were not allocated into experimental groups, state so OR describe how participants were allocated to groups, and if allocation was not random, describe how covariates were controlled.* |

# Ecological, evolutionary & environmental sciences study design

All studies must disclose on these points even when the disclosure is negative.

| | |
|---|---|
| Study description | *Briefly describe the study. For quantitative data include treatment factors and interactions, design structure (e.g. factorial, nested, hierarchical), nature and number of experimental units and replicates.* |
| Research sample | *Describe the research sample (e.g. a group of tagged Passer domesticus, all Stenocereus thurberi within Organ Pipe Cactus National Monument), and provide a rationale for the sample choice. When relevant, describe the organism taxa, source, sex, age range and any manipulations. State what population the sample is meant to represent when applicable. For studies involving existing datasets, describe the data and its source.* |
| Sampling strategy | *Note the sampling procedure. Describe the statistical methods that were used to predetermine sample size OR if no sample-size calculation was performed, describe how sample sizes were chosen and provide a rationale for why these sample sizes are sufficient.* |
| Data collection | *Describe the data collection procedure, including who recorded the data and how.* |
| Timing and spatial scale | *Indicate the start and stop dates of data collection, noting the frequency and periodicity of sampling and providing a rationale for these choices. If there is a gap between collection periods, state the dates for each sample cohort. Specify the spatial scale from which the data are taken* |
| Data exclusions | *If no data were excluded from the analyses, state so OR if data were excluded, describe the exclusions and the rationale behind them, indicating whether exclusion criteria were pre-established.* |
| Reproducibility | *Describe the measures taken to verify the reproducibility of experimental findings. For each experiment, note whether any attempts to repeat the experiment failed OR state that all attempts to repeat the experiment were successful.* |
| Randomization | *Describe how samples/organisms/participants were allocated into groups. If allocation was not random, describe how covariates were controlled. If this is not relevant to your study, explain why.* |
| Blinding | *Describe the extent of blinding used during data acquisition and analysis. If blinding was not possible, describe why OR explain why blinding was not relevant to your study.* |

Did the study involve field work?  ☐ Yes  ☐ No

# Field work, collection and transport

| | |
|---|---|
| Field conditions | *Describe the study conditions for field work, providing relevant parameters (e.g. temperature, rainfall).* |
| Location | *State the location of the sampling or experiment, providing relevant parameters (e.g. latitude and longitude, elevation, water depth).* |
| Access & import/export | *Describe the efforts you have made to access habitats and to collect and import/export your samples in a responsible manner and in compliance with local, national and international laws, noting any permits that were obtained (give the name of the issuing authority, the date of issue, and any identifying information).* |
| Disturbance | *Describe any disturbance caused by the study and how it was minimized.* |

# Reporting for specific materials, systems and methods

We require information from authors about some types of materials, experimental systems and methods used in many studies. Here, indicate whether each material, system or method listed is relevant to your study. If you are not sure if a list item applies to your research, read the appropriate section before selecting a response.

## Materials & experimental systems

| n/a | Involved in the study |
|---|---|
| ☒ | ☐ Antibodies |
| ☒ | ☐ Eukaryotic cell lines |
| ☒ | ☐ Palaeontology and archaeology |
| ☒ | ☐ Animals and other organisms |
| ☒ | ☐ Clinical data |
| ☒ | ☐ Dual use research of concern |
| ☒ | ☐ Plants |

## Methods

| n/a | Involved in the study |
|---|---|
| ☒ | ☐ ChIP-seq |
| ☐ | ☒ Flow cytometry |
| ☒ | ☐ MRI-based neuroimaging |

## Antibodies

| | |
|---|---|
| Antibodies used | *Describe all antibodies used in the study; as applicable, provide supplier name, catalog number, clone name, and lot number.* |
| Validation | *Describe the validation of each primary antibody for the species and application, noting any validation statements on the manufacturer's website, relevant citations, antibody profiles in online databases, or data provided in the manuscript.* |

## Eukaryotic cell lines

Policy information about cell lines and Sex and Gender in Research

| | |
|---|---|
| Cell line source(s) | *State the source of each cell line used and the sex of all primary cell lines and cells derived from human participants or vertebrate models.* |
| Authentication | *Describe the authentication procedures for each cell line used OR declare that none of the cell lines used were authenticated.* |
| Mycoplasma contamination | *Confirm that all cell lines tested negative for mycoplasma contamination OR describe the results of the testing for mycoplasma contamination OR declare that the cell lines were not tested for mycoplasma contamination.* |
| Commonly misidentified lines (See ICLAC register) | *Name any commonly misidentified cell lines used in the study and provide a rationale for their use.* |

## Palaeontology and Archaeology

| | |
|---|---|
| Specimen provenance | *Provide provenance information for specimens and describe permits that were obtained for the work (including the name of the issuing authority, the date of issue, and any identifying information). Permits should encompass collection and, where applicable, export.* |
| Specimen deposition | *Indicate where the specimens have been deposited to permit free access by other researchers.* |

| Dating methods | *If new dates are provided, describe how they were obtained (e.g. collection, storage, sample pretreatment and measurement), where they were obtained (i.e. lab name), the calibration program and the protocol for quality assurance OR state that no new dates are provided.* |

☐ Tick this box to confirm that the raw and calibrated dates are available in the paper or in Supplementary Information.

| Ethics oversight | *Identify the organization(s) that approved or provided guidance on the study protocol, OR state that no ethical approval or guidance was required and explain why not.* |

Note that full information on the approval of the study protocol must also be provided in the manuscript.

# Animals and other research organisms

Policy information about studies involving animals; ARRIVE guidelines recommended for reporting animal research, and Sex and Gender in Research

| Laboratory animals | *For laboratory animals, report species, strain and age OR state that the study did not involve laboratory animals.* |

| Wild animals | *Provide details on animals observed in or captured in the field; report species and age where possible. Describe how animals were caught and transported and what happened to captive animals after the study (if killed, explain why and describe method; if released, say where and when) OR state that the study did not involve wild animals.* |

| Reporting on sex | *Indicate if findings apply to only one sex; describe whether sex was considered in study design, methods used for assigning sex. Provide data disaggregated for sex where this information has been collected in the source data as appropriate; provide overall numbers in this Reporting Summary. Please state if this information has not been collected. Report sex-based analyses where performed, justify reasons for lack of sex-based analysis.* |

| Field-collected samples | *For laboratory work with field-collected samples, describe all relevant parameters such as housing, maintenance, temperature, photoperiod and end-of-experiment protocol OR state that the study did not involve samples collected from the field.* |

| Ethics oversight | *Identify the organization(s) that approved or provided guidance on the study protocol, OR state that no ethical approval or guidance was required and explain why not.* |

Note that full information on the approval of the study protocol must also be provided in the manuscript.

# Clinical data

Policy information about clinical studies
All manuscripts should comply with the ICMJE guidelines for publication of clinical research and a completed CONSORT checklist must be included with all submissions.

| Clinical trial registration | *Provide the trial registration number from ClinicalTrials.gov or an equivalent agency.* |

| Study protocol | *Note where the full trial protocol can be accessed OR if not available, explain why.* |

| Data collection | *Describe the settings and locales of data collection, noting the time periods of recruitment and data collection.* |

| Outcomes | *Describe how you pre-defined primary and secondary outcome measures and how you assessed these measures.* |

# Dual use research of concern

Policy information about dual use research of concern

## Hazards

Could the accidental, deliberate or reckless misuse of agents or technologies generated in the work, or the application of information presented in the manuscript, pose a threat to:

| No | Yes | |
|----|-----|---|
| ☐ | ☐ | Public health |
| ☐ | ☐ | National security |
| ☐ | ☐ | Crops and/or livestock |
| ☐ | ☐ | Ecosystems |
| ☐ | ☐ | Any other significant area |

## Experiments of concern

Does the work involve any of these experiments of concern:

| No | Yes | |
|----|-----|---|
| ☐ | ☐ | Demonstrate how to render a vaccine ineffective |
| ☐ | ☐ | Confer resistance to therapeutically useful antibiotics or antiviral agents |
| ☐ | ☐ | Enhance the virulence of a pathogen or render a nonpathogen virulent |
| ☐ | ☐ | Increase transmissibility of a pathogen |
| ☐ | ☐ | Alter the host range of a pathogen |
| ☐ | ☐ | Enable evasion of diagnostic/detection modalities |
| ☐ | ☐ | Enable the weaponization of a biological agent or toxin |
| ☐ | ☐ | Any other potentially harmful combination of experiments and agents |

# Plants

| | |
|---|---|
| Seed stocks | *Report on the source of all seed stocks or other plant material used. If applicable, state the seed stock centre and catalogue number. If plant specimens were collected from the field, describe the collection location, date and sampling procedures.* |
| Novel plant genotypes | *Describe the methods by which all novel plant genotypes were produced. This includes those generated by transgenic approaches, gene editing, chemical/radiation-based mutagenesis and hybridization. For transgenic lines, describe the transformation method, the number of independent lines analyzed and the generation upon which experiments were performed. For gene-edited lines, describe the editor used, the endogenous sequence targeted for editing, the targeting guide RNA sequence (if applicable) and how the editor was applied.* |
| Authentication | *Describe any authentication procedures for each seed stock used or novel genotype generated. Describe any experiments used to assess the effect of a mutation and, where applicable, how potential secondary effects (e.g. second site T-DNA insertions, mosiacism, off-target gene editing) were examined.* |

# ChIP-seq

## Data deposition

☐ Confirm that both raw and final processed data have been deposited in a public database such as GEO.

☐ Confirm that you have deposited or provided access to graph files (e.g. BED files) for the called peaks.

| | |
|---|---|
| Data access links<br>*May remain private before publication.* | *For "Initial submission" or "Revised version" documents, provide reviewer access links.  For your "Final submission" document, provide a link to the deposited data.* |
| Files in database submission | *Provide a list of all files available in the database submission.* |
| Genome browser session<br>(e.g. UCSC) | *Provide a link to an anonymized genome browser session for "Initial submission" and "Revised version" documents only, to enable peer review.  Write "no longer applicable" for "Final submission" documents.* |

## Methodology

| | |
|---|---|
| Replicates | *Describe the experimental replicates, specifying number, type and replicate agreement.* |
| Sequencing depth | *Describe the sequencing depth for each experiment, providing the total number of reads, uniquely mapped reads, length of reads and whether they were paired- or single-end.* |
| Antibodies | *Describe the antibodies used for the ChIP-seq experiments; as applicable, provide supplier name, catalog number, clone name, and lot number.* |
| Peak calling parameters | *Specify the command line program and parameters used for read mapping and peak calling, including the ChIP, control and index files used.* |
| Data quality | *Describe the methods used to ensure data quality in full detail, including how many peaks are at FDR 5% and above 5-fold enrichment.* |
| Software | *Describe the software used to collect and analyze the ChIP-seq data. For custom code that has been deposited into a community repository, provide accession details.* |

# Flow Cytometry

## Plots

Confirm that:

☐ The axis labels state the marker and fluorochrome used (e.g. CD4-FITC).

☐ The axis scales are clearly visible. Include numbers along axes only for bottom left plot of group (a 'group' is an analysis of identical markers).

☐ All plots are contour plots with outliers or pseudocolor plots.

☒ A numerical value for number of cells or percentage (with statistics) is provided.

## Methodology

| | |
|---|---|
| Sample preparation | The cultures are sampled from incubations of D. alkaliphilus with (1) ferrihydrite and 50 µM sulfide; (2) ferrihydrite and 1 mM sulfide; (3) ferrihydrite and FeS; and (4) sulfide and nitrate. subcultures (450 µl) were sampled periodically and preserved in 2.3% formaldehyde (final concentration). Before counting, 500 µl dithionite solution was added to 50-100 µl of fixed cells to dissolve the FeS and ferrihydrite particles. After dissolution of solid iron phase (within 10–15 minutes), 100 µl of each sample was diluted in 900 µl of 1× phosphate-buffered saline (PBS). The suspension was then sonicated using a SONOPULS ultrasonic homogenizer (Bandelin, Berlin, Germany) at 25% power with a cycle setting of 2 for a total of 30 seconds. Cells were subsequently stained with SYBR Green 1x (ThermoFisher, Waltham, USA) and incubated for 10 minutes at room temperature in the dark. |
| Instrument | CytoFLEX S flow cytometer (Beckman Coulter, Brea, CA, USA) equipped with a blue 488 nm laser |
| Software | CytExpert 2.6 software (Beckman Coulter) |
| Cell population abundance | During flow cytometry experiments D. alkaliphilus population abundance was between approximately 300-5,000 cells/µl in 60-fold dilutions of dithionite-treated cultures. |
| Gating strategy | D. alkaliphilus cells were gated based on forward scatter (FSC) and SYBR Green fluorescence using an excitation at 488 nm and emission detection with a 525/40 nm bandpass filter. Cells were discriminated from background particles by the SYBR Green-induced fluorescence shift relative to unstained (negative) controls. For high throughput analysis a fluorescence threshold was applied on SYBR Green fluorescence to reduce background noise. Total counts were monitored over time, and only data acquired during the period of stable event rates (counts per second) were included in the analysis. Gating was performed individually for each sample to account for sample-specific variation. |

☒ Tick this box to confirm that a figure exemplifying the gating strategy is provided in the Supplementary Information.

# Magnetic resonance imaging

## Experimental design

| | |
|---|---|
| Design type | *Indicate task or resting state; event-related or block design.* |
| Design specifications | *Specify the number of blocks, trials or experimental units per session and/or subject, and specify the length of each trial or block (if trials are blocked) and interval between trials.* |
| Behavioral performance measures | *State number and/or type of variables recorded (e.g. correct button press, response time) and what statistics were used to establish that the subjects were performing the task as expected (e.g. mean, range, and/or standard deviation across subjects).* |

## Acquisition

| | |
|---|---|
| Imaging type(s) | *Specify: functional, structural, diffusion, perfusion.* |
| Field strength | *Specify in Tesla* |
| Sequence & imaging parameters | *Specify the pulse sequence type (gradient echo, spin echo, etc.), imaging type (EPI, spiral, etc.), field of view, matrix size, slice thickness, orientation and TE/TR/flip angle.* |
| Area of acquisition | *State whether a whole brain scan was used OR define the area of acquisition, describing how the region was determined.* |

Diffusion MRI ☐ Used ☐ Not used

## Preprocessing

| | |
|---|---|
| Preprocessing software | *Provide detail on software version and revision number and on specific parameters (model/functions, brain extraction, segmentation, smoothing kernel size, etc.).* |

| Normalization | *If data were normalized/standardized, describe the approach(es): specify linear or non-linear and define image types used for transformation OR indicate that data were not normalized and explain rationale for lack of normalization.* |
|---|---|
| Normalization template | *Describe the template used for normalization/transformation, specifying subject space or group standardized space (e.g. original Talairach, MNI305, ICBM152) OR indicate that the data were not normalized.* |
| Noise and artifact removal | *Describe your procedure(s) for artifact and structured noise removal, specifying motion parameters, tissue signals and physiological signals (heart rate, respiration).* |
| Volume censoring | *Define your software and/or method and criteria for volume censoring, and state the extent of such censoring.* |

## Statistical modeling & inference

| Model type and settings | *Specify type (mass univariate, multivariate, RSA, predictive, etc.) and describe essential details of the model at the first and second levels (e.g. fixed, random or mixed effects; drift or auto-correlation).* |
|---|---|
| Effect(s) tested | *Define precise effect in terms of the task or stimulus conditions instead of psychological concepts and indicate whether ANOVA or factorial designs were used.* |

Specify type of analysis:  ☐ Whole brain   ☐ ROI-based   ☐ Both

| Statistic type for inference | *Specify voxel-wise or cluster-wise and report all relevant parameters for cluster-wise methods.* |
|---|---|

(See Eklund et al. 2016)

| Correction | *Describe the type of correction and how it is obtained for multiple comparisons (e.g. FWE, FDR, permutation or Monte Carlo).* |
|---|---|

## Models & analysis

| n/a | Involved in the study |
|---|---|
| ☐ | ☐ Functional and/or effective connectivity |
| ☐ | ☐ Graph analysis |
| ☐ | ☐ Multivariate modeling or predictive analysis |

| Functional and/or effective connectivity | *Report the measures of dependence used and the model details (e.g. Pearson correlation, partial correlation, mutual information).* |
|---|---|
| Graph analysis | *Report the dependent variable and connectivity measure, specifying weighted graph or binarized graph, subject- or group-level, and the global and/or node summaries used (e.g. clustering coefficient, efficiency, etc.).* |
| Multivariate modeling and predictive analysis | *Specify independent variables, features extraction and dimension reduction, model, training and evaluation metrics.* |

