## [Peer Review File · Nature]

Microbial iron oxide respiration coupled to sulfide oxidation

Corresponding Author: Professor Alexander Loy

Version 0:

Reviewer comments:

Referee #1

(Remarks on code availability)

(Remarks to the Author)

The current manuscript describes the novel and highly interesting finding that the use of iron oxide as a respiratory electron acceptor can be coupled to the oxidation of sulfide as an electron donor. Up to now the process had been thought to occur exclusively by spontaneous chemical reactions, albeit some involvement of biological processes had already been suggested by geochemical observations.

I have some comments that need attention:

The authors first scanned the available archaeal and bacterial genomes for the occurrence of more than a hundred genes related to sulfur cycling. In line 86, I do not understand the rationale for selecting the subset of 40 sulfur cycling enzymes. Why were these particular enzymes chosen? If they are indeed key markers, this needs to be proven by citing appropriate references in the corresponding Supplementary Table.

In the list of sulfur cycling enzymes the sHdr system of sulfur oxidation although it is wide-spread and although it is the only pathway for sulfane sulfur oxidation in the cytoplasm existing in archaea (but not restricted to them!). among the Bacteria, the respective gene set is more common than the genes for the Dsr pathway of sulfur oxidation. Why was sHdr not included?

Taking it into consideration might considerably expand the number and diversity of sulfur-metabolizing bacteria and archaea. If the authors think that sHdr should be disregarded, this would need at least need some explanation. Published comparative and proteomic studies have already indicated its existence more than ten years ago.

Fig. 1 shows phyla with species encoding at least one sulfur-cycling marker protein. This appears very little. The authors should explain how the ability of an organism to play a significant role in sulfur cycling can be inferred from the presence of a single gene. In some cases this might be true, e.g. the possession of sulfide:quinone oxidoreductase is enough to get from sulfide to sulfur, but in many other cases a set of enzymes has to work together (e.g. Dsr). The authors must take better account of this problem.

In a number of sulfur-metabolizing bacteria, the authors detected co-occurring genes related to iron oxidation. The conclusion from the metabolic reconstruction was that these organisms should be able to couple sulfide oxidation and iron oxidation which was then tested for a cultivated representative, *Desulfurivibrio alkaliphilus*, an organisms that is known for its slow growth and very low growth yields. The authors first tested media containing FeS. Only the FeS-sulfur can be oxidized. Sulfate was formed in the presence of living cells of *D. alkaliphilus* and this went along with Fe(III) reduction. Growth was not observed. The authors state that the process facilitated survival.

Very little growth was then observed on sulfide and Fe(III). In this case, growth was monitored by increased 16S rRNA gene copy numbers. However, gene copy numbers rose by only a factor of about 2.5. Is it sure that this really went along with an increase in cell numbers? In Thorup et al 2017 who describe sulfide oxidation with nitrate reduction by the same organism, the increase in copy numbers was also low but certainly more pronounced (about 4 to 5-fold). More explanation in the text, regarding the limited general growth capacities of the studied organism is certainly necessary. On days 2 to 4 sulfate formation continued with the same rate as during the first two days, but there was no increase in 16s RNA gene copies any more. What is the reason for this behavior? What happens when incubation times are increased? In addition, in the control experiments, where the cells were exposed to ferrihydrite alone and to sulfide alone there were no 16S rRNA gene copies detectable. However, we need some proof for the presence of living cells in these experiments.

In line 218, it is stated that *D. alkaliphilus* shows a "high sulfate formation rate". What does "high" mean? In comparison to

what other rate(s) is this rate high?

A protein whose mRNA is 100 times more abundant in the presence of Fe(III) is proposed to produce nanowire-like structures. It would be good to have real evidence for the formation of these nanowires. For example, electron micrographs. In Fig. 2 the authors propose that it should also be possible to couple thiosulfate oxidation to Fe(III) reduction. Organism with the genetic capacity to carry out this process include several cultivated members of alpha- and gammaproteobacterial families of the phylum Pseudomonadota. Have any efforts been made to provide evidence here? The authors should definitely comment on why this was not tested (possibly even faster side reactions?). The conclusions from line 413 onwards appear very far-fetched given the fact that growth with the process is hardly possible or does not even occur (in the case of FeS as the substrate).

Referee #2

(Remarks to the Author)

Chen et al., REVIEW

This manuscript describes nothing less than proof of a novel microbial metabolism, the oxidation of reduced sulfur compounds with solid iron oxides. The discovery that microbes not only perform but outpace this known chemical reaction is an important breakthrough for microbiology and earth & environmental sciences alike. In addition to this key discovery, the study also offers the to date most thorough overview of the distribution of sulfur cycling marker genes in the prokaryotic world, with newly and more strictly defined HMMs (that are made publicly available). Together with marker genes for iron reduction, these sulfur genes served to point the authors to likely candidate bacteria for the predicted metabolism, which then was tested with *Desulfurivibrio alkaliphilus* as model.

Although the paper is very well structured and written, it takes stamina to get through the wealth of physiological and chemical analyses that were necessary to unambiguously separate bacterial from chemical reactions and to exclude that sulfur disproportionation (a known trait of *D. alkaliphilus*) is the actual reason for sulfate production. The authors succeed with both, in a thorough and convincing way. Transcriptomic and bioinformatic analyses complement the study and provide additional, robust support of the novel metabolism by suggesting a plausible mechanism for the pathway. Methods, data, and statistics are throughout valid, of high quality, and appropriate (a few questions about statistical support remain, see detailed comments below). The paper ends with a balanced and careful conclusion on the ecological and geochemical implications, which showcases the authors' extraordinary overview of literature on sulfate formation in anoxic environments. I have only one major and a few minor (mostly technical) comments for improvement and would otherwise like to congratulate the authors to their profound work and exceptional discovery.

Main comment: while the sulfur gene analysis is extremely thorough, the Fe reduction potential seems not equally well supported – apparently only the most common (Gram negative) model organisms were included, newer insights from Gram positives (e.g. Paquette et al 2020, Light et al., 2018, 2019) or Archaea (e.g., ANME, methanogens) were not included in the search. Not surprisingly, the predicted Fe-reducing S oxidizers are restricted to *Desulfurivibrionaceae* and a few Proteobacteria. While this does not diminish the value of the discovery, it may limit its evolutionary, ecological, and environmental implication. If it is not possible to update the Fe reducer marker gene analysis, this limitation should at least be mentioned.

Minor comments:

- l. 28: consider changing “biogeochemical” to “geochemical” and “non-enzymatic” to “abiotic or non-biological”
 - Fig. 1: the “Inorg. S” genes are separated by a gap into an apparent mostly oxidative (left) and mostly reductive (right) part – maybe consider choosing different colors/indicating this separation? Or remove the gap, if too much mixed?
 - Fig. 2, DeltaG for thiosulfate oxidation: do the red and black lines overlap? If so, please mention in the figure legend
 - Fig. 3C, 16S rRNA gene copies: according to methods, incubations were started with 30 ml of a culture with approx. 10^8 cells/ml, into 60 ml, i.e. the starting conc. should be around 5×10^7 cells/ml. The qPCR data show around 5×10^6 : is this a problem with the qPCR assay, the DNA extraction efficiency, or an error?
 - l. 283: “killed cells were served as controls” remove “were”
 - l. 291-292: this is a bit confusing, as the first candidate genes for this metabolism were already identified in the genome searches for sulfur and Fe marker genes, weren't they? Consider rephrasing.
 - l. 324-325: the role of PilA as “conductive nanowire”, although not fully disproven yet, is currently being replaced by cytochrome nanowires (OmcS, OmcZ etc.), while the actual role of PilA is not completely clear. This is also evident from your data, where you see a clear upregulation of omcS (and other MHCs) with iron oxides, but not of PilA (T4P). I suggest to reflect that also in the text, instead of “neutrally” entertaining two alternative roles.
 - Extended data Fig. 4: it would be helpful to add the gene names to the locus tags, so one doesn't have to compare one by one with Extended data Fig. 3 to identify the MHCs
 - l. 354: where is the statistical significance shown? Or does the figure only contain genes that are significantly upregulated? If so, please indicate in the legend, and please be specific which conditions were compared (there's only one obvious pair, sulfide/Fe(III) vs. sulfide/nitrate; or did you average all 3 Fe(III) conditions against both non-Fe(III) conditions?
 - l. 366: a TM-score of 0.56 is only slightly above the accepted threshold of 0.5 for structural homology. Consider additional support by adding rmsd or DALI scores?
 - l. 471: “KoFAM” is listed as “KEGG” in Fig S2, please be consistent
 - l. 574-575: how were the ferrihydrites sterilized when preparing them with mortar and pestle?
- Supplementary Information:
- l. 71: “were” should read “was”
 - l. 83: “oxidize sulfur to elemental sulfur” should read “oxidize sulfide to elemental sulfur”
 - l. 121: “share close homology” NO! they “share close identity/similarity” – there are no close or distant homologs: if they were

not homologs, they should not be in the same phylogenetic tree.

I. 142-143: statistical significance of temporal changes: not indicated in the figure? Please add.

I. 233-247, Relevance of MISO: I can see why the authors think this is a conservative estimate, given the bio and geo chemical recycling of Fe. But on the other hand, not the entire riverine input of Fe oxides will end up in zones without other oxidants, and then oxygen and nitrate (and depending on the minerals even manganese) are more favorable electron acceptors, so that part of the Fe may only undergo chemical transformation if any.

L. 245: "100-300 times of redox cycles" delete "times of"

(Remarks on code availability)

Code available and well described. I did not try to reproduce the results of the paper.

Referee #3

(Remarks to the Author)

Review of Nature manuscript 2024-10-20940 "Microbial iron oxide respiration coupled to sulfide oxidation" by Chen et al.

Overview

In this article, the authors present experimental (and genomic) evidence for the existence of microbial sulfide oxidation coupled to Fe(III) mineral reduction at slightly alkaline pH (9.3), a process that has been suggested (mainly speculated) to exist for a long time, but so far no convincing experimental evidence (with a pure culture of microorganisms) has been provided. This process represents an important link between the sulfur and iron biogeochemical cycles and has been shown so far to be purely abiotic. The study presented here is novel because it provides for the first time convincing evidence that microbes are able to harvest the energy available in the reaction between sulfide and Fe(III). This evidence comes from a thorough genomic analysis and laboratory experiments with a microbial pure culture, i.e. *Desulfurivibrio alkaliphilus*. Since the sulfur and iron cycles are two of the most important environmental biogeochemical cycles, this is very important and certainly deserves publication.

The data provided in the paper are very convincing although I am not an expert in the genomic analysis that has been performed. However, the microbial cultivation work was very carefully done, the setups and in particular the control experiments were very smart and the authors provide convincing data that the observed process is indeed enzymatic and not an abiotic reaction. The data is clearly presented and easy to follow. In my opinion, the conclusions drawn from the data are robust and reliable. Both the abstract and the main manuscript text are well written and easy to read.

Below, I provide some general comments and specific line-by-line suggestions for revising the manuscript and I hope that these are helpful to the authors.

General comments and questions

1. Please be consistent with the terminology: sometimes the authors write "sulfur oxidation" when they mean "sulfide oxidation", sometimes I was not sure what they mean (see also specific comments below). Maybe at one point it would be good to define "sulfur oxidation" as oxidation of reduced sulfur species in general? Please check carefully and revise throughout the whole manuscript.
2. The authors used ferrihydrite for their experiments but in the text use the term "iron oxide". Ferrihydrite is an "iron(III) oxyhydroxide" (maybe for simplification you could say "iron hydroxide"). I recommend either using "iron oxyhydroxide" or at least defining "iron oxide" as a simplified term for ferrihydrite in the beginning of the manuscript.
3. Overall I find the results provided from the *D. alkaliphilus* experiments convincing (demonstrating sulfide oxidation coupled to iron(III) reduction). I am wondering whether S-isotope analyses could provide an additional argument in favor of this microbial process. I don't think it is necessary for the present paper, but maybe for future publications. I also thought about using Fe-isotope analysis, but I assume the Fe-isotope fractionation when comparing the abiotic reduction of ferrihydrite by sulfide to the biotic process will not be very different from each other. Nevertheless, maybe worth a try. Both isotope systems (S and Fe) could help to get some information about the role of microbial (and abiotic) sulfide oxidation coupled to iron(III) reduction on early Earth.
4. Did the authors try to use ¹³C labelled CO₂/bicarbonate to demonstrate CO₂ fixation and growth in cultures incubated with sulfide and ferrihydrite and in cultures incubated with FeS and ferrihydrite?

Specific comments

1. L32, L52, L57: sulfur or sulfide oxidation?
2. L59: this would be a good place to define "iron oxides". Here you mean all iron(III) minerals including oxides (hematite, magnetite) and oxyhydroxides (goethite, lepidocrocite, ferrihydrite). I therefore usually prefer "iron(III) (oxyhydr)oxides" to include all of them. Maybe here you can explain that from now on the term "iron oxides" will be used for simplicity?
3. L62: what does "this process" refer to?
4. L106 and L109: sulfur or sulfide oxidation?
5. L111: I assume you mean "dissolved iron(III)"?
6. L114: "iron(III) reduction".
7. L117 and L123: "iron(III) reduction".
8. L130: "iron(III)-reducer".
9. L135: "...extracellular iron(III)-dependent thiosulfate oxidation"?
10. L138: "iron(III)-dependent".
11. L183: should the cell numbers not increase (instead of staying constant)?
12. L204 and L206: "iron(III) reduction".
13. L214 and L218: I recommend writing "poorly soluble", not "insoluble".
14. L242: is it really "amorphous" or maybe rather "poorly crystalline"?

15. L244: regarding the 1:8 stoichiometry: if the cells grow and use some electrons to also fix CO₂, the ratio should be even a bit higher (slightly more sulfide oxidized than iron(III) reduced), right?
16. L307: "iron(III)-respiring".
17. L316: "iron(III) oxides".
18. L379: "dissolved iron(III)".
19. L511: "iron(III)-dependent".
20. L538: see comment above regarding the terminology for the FeS: I personally feel that "poorly ordered" or "poorly crystalline" are more appropriate than "amorphous".
21. L572: I think following sulfide and/or sulfate provides evidence for microbial activity, but not for growth. This requires biomass quantification, cell counts, protein content quantification, ¹³CO₂ fixation, etc.
22. L625-626: during the acidification step, remaining sulfide will react with remaining Fe(III) (especially when the ferrihydrite gets dissolved), right? Does this lead to an overestimation of the amount of formed Fe(II)?

(Remarks on code availability)

Version 1:

Reviewer comments:

Referee #1

(Remarks to the Author)

All my concerns were clarified by additional experiments and methods.

(Remarks on code availability)

Referee #2

(Remarks to the Author)

This is an extensive revision of a manuscript I had for review in its original version, and I can only congratulate the authors to an excellent and thorough job that has fully addressed all my concerns and doubts, and significantly improved both the data by adding new experiments and analyses, and the presentation of the manuscript: this is how peer review should work, at its best.

Again, congratulations to a groundbreaking discovery and a now even better presentation.

(Remarks on code availability)

Referee #3

(Remarks to the Author)

The authors did an excellent job considering my suggestions and comments in their revised manuscript. No further changes required from my side.

(Remarks on code availability)

Response to reviewers' comments

We are very happy that our study was generally well received by the reviewers and are grateful for their valuable comments, which helped us to improve the manuscript. We have addressed all comments in our point-by-point response and revised the manuscript accordingly. Changes in the manuscript and Supplementary Information have been marked in **red** font with yellow background.

Our revision is based on additional computational analyses and extensive growth experiments that, for example, clearly show that ***Desulfurivibrio alkaliphilus* grows autotrophically by coupling sulfide oxidation to ferrihydrite reduction.**

Please find below a brief summary of the key experiments and analyses and the main outcomes.

Inclusion of the sHdr sulfur oxidation system for establishing the genomic census of sulfur-metabolizing bacteria and archaea. Phylogeny-informed Hidden Markov models were developed for all six essential subunits of the sulfur-oxidizing heterodisulfide reductase (sHdr) enzymatic complex. **The sHdr system was detected in 138 genomes from the GTDB database**, including well-characterized sulfur-oxidizing bacteria (e.g., *Sulfobacillus* and *Thioalkalivibrio*) and archaea (e.g., *Acidianus*). Including the sHDR system in the screening increased the number of species (from 16814 to 16839) but not the number of phyla that include members with a putative sulfur metabolism.

Re-analysis of co-occurrence of dissimilatory sulfur and iron(III) respiration across microbial genomes. We have now included additional genetic markers for the sHdr system and extracellular electron transfer (EET) pathways, such as flavin-based EET in Gram-positive bacteria and outer-membrane multiheme cytochromes (OMCs) in various taxa, in our genome screening. This considerably **expanded the taxonomic diversity of microorganisms predicted to couple sulfur oxidation with iron(III) reduction, identifying members from 37 bacterial and archaeal phyla.** These findings now highlight an even broader diversity and environmental relevance of potential Fe(III) oxide-reducing sulfur oxidizers.

Growth experiments demonstrated that *Desulfurivibrio alkaliphilus* utilizes energy generated from coupled sulfide oxidation and iron(III) oxide reduction for growth. Using an improved cell counting method, we observed a 2.5–3-fold increase in cell density in cultures with ferrihydrite and sulfide or with ferrihydrite and FeS. Specific growth rates were estimated as $0.436 \pm 0.074 \text{ day}^{-1}$, $0.288 \pm 0.015 \text{ day}^{-1}$, and $0.089 \pm 0.005 \text{ day}^{-1}$ for cultures amended periodically with ca. 50 μM sulfide, 1 mM sulfide, and FeS, respectively. These results show that the **MISO process enables growth under environmentally relevant sulfide concentrations**, albeit with limited yields, possibly due to high maintenance energy demands under alkaline conditions.

Bulk biomass and single cell ^{13}C -bicarbonate labeling directly linked MISO to autotrophic carbon fixation. Experiments with ^{13}C -labeled bicarbonate (10 atom%) demonstrated carbon fixation in *D. alkaliphilus* cultures incubated with ferrihydrite and sulfide or ferrihydrite and FeS. The ^{13}C atom% in the total biomass and single cells increased

significantly in living cultures over six days, demonstrating that **the MISO process supports autotrophic carbon fixation.**

Referees' comments:

Referee #1 (Remarks to the Author):

The current manuscript describes the novel and highly interesting finding that the use of iron oxide as a respiratory electron acceptor can be coupled to the oxidation of sulfide as an electron donor. Up to now the process had been thought to occur exclusively by spontaneous chemical reactions, albeit some involvement of biological processes had already been suggested by geochemical observations.

I have some comments that need attention:

The authors first scanned the available archaeal and bacterial genomes for the occurrence of more than a hundred genes related to sulfur cycling. In line 86, I do not understand the rationale for selecting the subset of 40 sulfur cycling enzymes. Why were these particular enzymes chosen? If they are indeed key markers, this needs to be proven by citing appropriate references in the corresponding Supplementary Table.

RESPONSE: Thank you for the comment. We now provide the rationale for choosing a subset of the sulfur cycling genes/proteins for the initial screening. It is based on at least one of the following three criteria:

(1) **Genes that are widely recognized as markers for sulfur metabolism.** Examples include the selection of established marker genes *dsrAB* among *dsr*-related genes *dsrABMKJOPEFHCL*¹⁻³, *aprA* among *aprABM-qmoABC*⁴, *soxB* among *soxABXYZCD*⁵, and the gateway gene *yihQ* among a broad list of over 30 genes involved in different sulfoglycolytic pathways⁶.

(2) **Genes encoding catalytic subunits essential for enzymatic activities.** Examples include selection of *fccB* encoding the large sulfide-binding subunit of the sulfide dehydrogenase FccBA^{7,8}, *phsA/psrA* encoding the catalytic subunit of the thiosulfate reductase PhsABC / polysulfide reductase PsrABC^{9,10}, *soeA* encoding the catalytic subunit of the sulfite dehydrogenase SoeABC¹¹.

(3) **Genes that alone confer a specific sulfur redox transformation.** Examples include *sqr* for a sulfide:quinone oxidoreductase, *otr* for an octaheme tetrathionate reductase, and many genes involved in DMS/DMSP metabolism (e.g., *dddP*, *dddL*, *dddD*, *dddQ*, and *dddY*).

These selection criteria kept the core genes that are indicative of sulfur metabolism among a list of now 116 sulfur-cycling genes. While reviewing the literature on these genes, we noticed *mmtN* also fulfilled criterion (1), as it has been recognized as a marker for bacterial synthesis of dimethylsulfoniopropionate¹². We thus also included it in the revised subset of sulfur-cycling marker genes; its taxonomic distribution has been updated in the revised Fig.

1. Supporting literature and detailed justifications for selection of each gene are now included in Table S1. The primary criteria for selecting the subset of sulfur-cycling genes as markers have been added to the Method section (line 542-545).

In the list of sulfur cycling enzymes the sHdr system of sulfur oxidation although it is wide-spread and although it is the only pathway for sulfane sulfur oxidation in the cytoplasm existing in archaea (but not restricted to them!). among the Bacteria, the respective gene set is more common than the genes for the Dsr pathway of sulfur oxidation. Why was sHdr not included? Taking it into consideration might considerably expand the number and diversity of sulfur-metabolizing bacteria and archaea. If the authors think that sHdr should be disregarded, this would need at least need some explanation. Published comparative and proteomic studies have already indicated its existence more than ten years ago.

RESPONSE: We thank the reviewer for the hint. We have now accounted for all essential components ($n = 6$) of the sHdr enzymatic complex in the revised manuscript. These include sHdrA, B1, C1, B2, C2, and B3 subunits from two types of the sHdr system¹³. To ensure consistency with other sulfur-cycling enzymes, we performed phylogenetic analysis for all six subunits and developed hidden markov models of the corresponding proteins, following the same pipeline described in the Method section. Using the proposed catalytic subunit sHdrB1 as a representative marker, we detected the sHdr system in 138 genomes from the GTDB database (release 95). These include well-characterized sulfur-oxidizing Bacteria (e.g., *Sulfobacillus* sp., *Aquifex* sp., and *Thioalkalivibrio* sp.) and Archaea (e.g., *Acidianus* sp.).

We revised Fig. 1b to show the phylum-level distribution of the sHdr system. We also updated Fig. 1a and the main text (line 86, 93, 100) to reflect the revised counts of sulfur-cycling genes and genomes showing sulfur-cycling potential.

Fig. 1 shows phyla with species encoding at least one sulfur-cycling marker protein. This appears very little. The authors should explain how the ability of an organism to play a significant role in sulfur cycling can be inferred from the presence of a single gene. In some cases this might be true, e.g. the possession of sulfide:quinone oxidoreductase is enough to get from sulfide to sulfur, but in many other cases a set of enzymes has to work together (e.g. Dsr). The authors must take better account of this problem.

RESPONSE: As explained above, we selected individual genes that encode (1) marker proteins of distinct sulfur metabolisms, (2) catalytic subunits of sulfur-transforming enzymatic complexes or (3) single-component enzymes conferring the ability of particular sulfur cycling alone (e.g., sulfide:quinone oxidoreductase). In the former two cases, most of the retrieved genomes also encode auxiliary proteins required for the sulfur-cycling capacity. For example, more than 95% of genomes extracted based on the presence of *dsrA* harbor at least four additional *dsr* genes (e.g., *dsrBCEFHLMKJOP*) for the coordinated operation in sulfur redox chemistry (new Fig. S3). The varied counts of *dsr* genes in *dsrA*-harboring genomes could either result from the incomplete nature of GTDB genomes recovered from metagenomes or it may reflect variants of Dsr-dependent pathways employing distinct sets of *dsr*¹⁴. Similar patterns were also observed for other selected markers or catalytic subunit encoding genes (new Fig. S3), with the majority (86-100%) of the retrieved genomes containing >50% of genes necessary for the full biochemical activities of the sulfur-cycling enzymes. These

results support the selected genes could serve as a proxy for the corresponding sulfur-cycling pathway.

Nevertheless, we emphasize that the results based on the analysis of a single gene provides only an initial insight, calling for more detailed genome-based metabolic reconstruction, as exemplified by the search for sulfur oxidation metabolism coupled with dissimilatory iron reduction. We have added a note on this in the main text at lines 97-99 “Most genomes that encode a marker protein (e.g., DsrA) also encode additional proteins (e.g., DsrBCEFHLMKJOP) associated with the respective marker protein for coordinated enzymatic function (Fig. S3).” and lines 104-105 “Our overview of putative sulfur microorganisms provides a foundation for more detailed metabolic predictions and experimental validation.”.

New Figure S3 | Majority of genomes that encode a sulfur-cycling marker protein also encode additional enzymatic components for coordinated function. For each marker (e.g., DsrA), the number of additional sulfur metabolism proteins associated with it (e.g., DsrBCEFHLMKJOP) was counted in each genome that contains the marker (e.g., DsrA-encoding genomes). The bar plot shows the frequency of the additional protein counts in the retrieved genomes. The horizontal line indicates the genomes with >50% of proteins (e.g., five in the case of the Dsr system) for the full pathway or the complete multienzyme complex. The accumulated fraction (e.g., 95.9% for DsrA) of genomes harboring >50% of proteins involved in the pathway or the enzymatic complex is shown above the line. Sulfur-cycling marker proteins that alone catalyze the specific sulfur redox transformation (e.g., sulfide:quinone oxidoreductase) were not analyzed.

In a number of sulfur-metabolizing bacteria, the authors detected co-occurring genes related to iron oxidation. The conclusion from the metabolic reconstruction was that these organisms should be able to couple sulfide oxidation and iron oxidation which was then tested for a cultivated representative, *Desulfurivibrio alkaliphilus*, an organism that is known for its slow growth and very low growth yields. The authors first tested media containing FeS. Only the FeS-sulfur can be oxidized. Sulfate was formed in the presence of living cells of *D. alkaliphilus* and this went along with Fe(III) reduction. Growth was not observed. The authors state that the process facilitated survival.

Very little growth was then observed on sulfide and Fe(III). In this case, growth was monitored by increased 16S rRNA gene copy numbers. However, gene copy numbers rose by only a factor of about 2.5. Is it sure that this really went along with an increase in cell numbers? In Thorup et al 2017 who describe sulfide oxidation with nitrate reduction by the same organism, the increase in copy numbers was also low but certainly more pronounced (about 4 to 5-fold). More explanation in the text, regarding the limited general growth capacities of the studied organism is certainly necessary.

RESPONSE: We thank the reviewer for raising the discussion on growth of *D. alkaliphilus*. In response, we performed a more comprehensive investigation of cell growth and confirmed the connection between MISO metabolism and cell proliferation. We achieved this by establishing an improved cell counting protocol that couples sonication with semi-automated flow cytometry. This dramatically improved the speed and reproducibility compared to manual cell counting. As a benchmark of the method, we monitored the growth of cells on sulfide (3 mM) and nitrate (4 mM), and observed 5- to 6-fold increases in cell density (new Fig. 4d), consistent with the results reported previously¹⁵.

With the refined flow cytometry protocol, we repeated our physiological experiments by counting the cells during incubation with ferrihydrite and sulfide or FeS. Given the generally limited growth of *D. alkaliphilus*, we reduced the inoculum concentration to $3\text{-}5 \times 10^7$ cells ml⁻¹, one-third to one-fifth of that used initially ($1\text{-}1.5 \times 10^8$ cells ml⁻¹). The new results showed that *D. alkaliphilus* was capable of growth in incubations with ferrihydrite and sulfide or FeS. We observed a 2.5-3 fold increase in cell density for cultures amended with 1 mM sulfide (new Fig. 4a), a 2-2.5 fold increase for cultures periodically amended with low amounts (ca. 50 μM) of sulfide (new Fig. 4c), and a 2-2.5 fold increase for FeS-amended cultures (new Fig. 4b). The estimated specific growth rate was 0.288 ± 0.015 day⁻¹, 0.089 ± 0.005 day⁻¹, and 0.436 ± 0.074 day⁻¹ for cultures amended periodically with 1 mM sulfide, FeS, and ca. 50 μM sulfide, respectively (new Fig. S11). We reasoned the weaker growth on FeS is likely due to limited accessibility to the substrate considering its low solubility. In contrast, we detected no or only minor increases in cell number in the sulfide-only, FeS-only, or ferrihydrite-only controls. In summary, these results demonstrate that MISO metabolism supports growth of *D. alkaliphilus*, despite its general slow growth and low growth yields.

We propose the limited growth capacity reflects an intrinsic property of *D. alkaliphilus* driven by high energy demands for maintenance metabolism. Under high pH conditions (pH > 9), *D. alkaliphilus* experiences stressful conditions known to increase the maintenance energy requirement, primarily due to the need for cytoplasmic pH homeostasis and the physiological adaptation to environmental stress^{16,17}. This elevated maintenance cost likely reduces the fraction of energy available for anabolic reactions, limiting the general growth capacity. The even more constrained growth with ferrihydrite vs. nitrate can be attributed to overall lower

energy yield associated with ferrihydrite reduction. Under the alkaline conditions tested (pH = 9.3), the ferrihydrite/Fe(II)CO₃ couple ($E^0 = -67.4$ mV) has a lower standard redox potential compared to the nitrate/ammonium redox couple ($E^0 = +194$ mV). This implies ferrihydrite respiration yields less catabolic energy that can be utilized for anabolism, and thereby results in weaker growth compared to nitrate respiration.

In the revised manuscript, we have now included these new cell growth results and explanation of the limited growth capacity in a new paragraph at line 253-268 (including new Fig. 4) and in Supplementary Text at line 237-251 (including new Fig. S11).

Selected panels from the new Figure 4 | Growth of *D. alkaliphilus* with different combinations of electron donors (i.e., sulfide or FeS) and acceptors (ferrihydrite or nitrate). **a**, Mean cell density ($n = 3$ replicate cultures) increased significantly ($P = 2.18 \times 10^{-8}$; ANOVA) over 4-day incubation with ferrihydrite (ca. 62 mM Fe equivalents) and daily addition of sulfide (1 mM). Parallel incubations devoid of either Fh or sulfide showed no increase in cell numbers. The arrows indicate the daily spike of sulfide. **b**, Mean cell density ($n = 3$ replicate cultures) increased significantly ($P = 3.79 \times 10^{-10}$; ANOVA) over 13-day incubation with ferrihydrite (ca. 62 mM Fe equivalents) and periodic spikes of FeS (ca. 1 mM). Parallel incubations devoid of either Fh or FeS showed no or only minor increases in cell numbers. The slight increase (10-25%) in Fh-only controls is likely due to the residue sulfide carried over from the inoculum. The arrows indicate the FeS additions every three to four days. **c**, Mean cell density ($n = 3$ replicate cultures) increased significantly ($P = 1.84 \times 10^{-8}$; ANOVA) over 5-day incubation with ferrihydrite (ca. 62 mM Fe equivalents) and periodic addition of environmentally relevant sulfide (ca. 50 μM). The arrows indicate the spike of sulfide. **d**, Mean cell density ($n = 3$ replicate cultures) increased significantly ($P = 5.53 \times 10^{-11}$; ANOVA) coinciding with sulfide consumption over 3-day incubation with sulfide and nitrate (4 mM). Sulfide was added at 0 (1.5 mM) and 1.8 days (1 mM) of incubation.

New Figure S11 | Specific growth rate (μ ; mean \pm standard deviation; unit: day^{-1}) of *D. alkaliphilus* with different combinations of electron donors (i.e., sulfide or FeS) and acceptors (ferrihydrite or nitrate). The specific growth rate under each growth condition was estimated by performing linear regression of $\ln(\text{Cell}_t/\text{Cell}_0)$ on time during the apparent exponential growth phase. Cell_t is the cell concentration (cells ml^{-1}) at sampling time t (in days). The 95% confidence interval (CI) of μ is shown for each growth condition (panel a-d)

On days 2 to 4 sulfate formation continued with the same rate as during the first two days, but there was no increase in 16s RNA gene copies any more. What is the reason for this behavior? What happens when incubation times are increased?

RESPONSE: Quantification of growth by 16S rRNA gene-targeted real-time PCR is prone to some limitations in our system, as DNA may strongly adsorb to iron oxides or FeS particles during DNA extraction¹⁸, which may cause low and variable DNA extraction efficiency. Therefore, we re-quantified cell density during periodic sulfide addition experiments using flow cytometry for direct cell counting as mentioned above. This approach is based on dissolving the solid iron phases with dithionite in an acetate buffer prior to staining and counting of the cells.

Our flow cytometry results showed a consistent increase in cell density from day 2 to day 4 (new Fig. 4a), coinciding with sulfate formation (Fig. 3c). By the end of incubation (day 4), cell density had roughly tripled compared to initial populations. In contrast, no cell growth was observed in the sulfide-only or ferrihydrite-only controls. These findings demonstrate a connection between cell growth and sulfate formation.

In addition, in the control experiments, where the cells were exposed to ferrihydrite alone and to sulfide alone there were no 16S rRNA gene copies detectable. However, we need some proof for the presence of living cells in these experiments.

RESPONSE: Actually, we previously have not analyzed 16S rRNA gene copy dynamics in the ferrihydrite-only or sulfide-only controls. We have now assessed cell viability in these control incubations by adding formate (10 mM) to the ferrihydrite-only cultures and nitrate (4 mM) to the sulfide-only cultures at the end of the 4-day incubation. Continued incubation revealed formate-dependent Fe(II) formation in ferrihydrite-only cultures (new Fig. S10a) and nitrate-dependent sulfide consumption in sulfide-only cultures (new Fig. S10b). These results confirmed the presence of viable cells capable of responding to the addition of an electron donor or acceptor.

In the revised manuscript, we have replaced the qPCR results from original Fig. 3 with the new flow cytometry-based cell counting data (new Fig. 4a-d) and added the results of the viability test in the Supplementary Text at lines 220-227, including new Fig. S10.

New Figure S10 | Viability of *D. alkaliphilus* cells after 4-day incubation with ferrihydrite (ca. 62 mM; Fh) or sulfide alone. **a**, Formation of HCl-extractable Fe(II) in Fh-only controls upon the addition of 10 mM formate at day 5 confirmed cell viability (red symbols). The arrow indicates the addition of formate. No significant Fe(II) increase was detected in Fh-only controls without formate addition over 15-day incubation. **b**, Consumption of sulfide in sulfide-only controls upon the addition of 4 mM nitrate at day 5 confirmed cell viability (red symbols). The red arrow indicates the addition of nitrate. Sulfide concentrations did not change in sulfide-only controls without nitrate addition after day 5.

In line 218, it is stated that *D. alkaliphilus* shows a “high sulfate formation rate”. What does “high” mean? In comparison to what other rate(s) is this rate high?

RESPONSE: We refer to the 10-fold higher sulfate formation rate in phase I compared to phase II. We have revised the text for clarification at lines 232-234, “We thus attribute the sulfide-dependent, high sulfate formation rate of *D. alkaliphilus* in phase I to direct biological sulfide oxidation with ferrihydrite.”

A protein whose mRNA is 100 times more abundant in the presence of Fe(III) is proposed to produce nanowire-like structures. It would be good to have real evidence for the formation of these nanowires. For example, electron micrographs.

RESPONSE: As suggested by the reviewer, we have now performed scanning electron microscopy (SEM) and transmission electron microscopy (TEM) imaging of *D. alkaliphilus* cells grown with ferrihydrite and sulfide. We did not obtain any clear evidence for appendages such as nanowires on the cell outer surfaces by either method (new Fig. S13a-e). We suspect that the lack of detection of nanowire-like structures could be due to (1) insufficient spatial resolution of SEM imaging to consistently reveal such fine structures, (2) the harsh sample preparation procedure for TEM, e.g., dithionite treatment, which may have destroyed fragile fine structures like nanowires^{19,20} or (3) the possibility that the MHC mediates Fe(III) reduction through direct contact, as demonstrated for other outer membrane cytochromes²¹. The latter is supported by fluorescence microscopy, i.e., the observation that most MISO-grown cells were attached to the iron particles (new Fig. S13f). At this stage, we can not exclude the presence of any potentially conductive appendages such as nanowires, but this would require substantial method optimization. We have included these results in the new section “Microscopy of *D. alkaliphilus* incubated with ferrihydrite and sulfide” in the Supplementary Information at lines 323-332, including new Fig. S13. In light of the microscopic observations and lack of conclusive evidence of nanowire structures, we have revised the respective parts in the main text (line 366-368 and 372-373) and Figure 5.

New Figure S13 | Microscopic images of *Desulfurivibrio alkaliphilus* grown with ferrihydrite and sulfide. Cells treated with dithionite for removal of solid-phase iron particles were imaged by scanning electron microscopy (a, b) and transmission electron microscopy (c-e). The scale bar in panels a-e shows 0.5 μm. Cells without dithionite treatment were stained with SYBR Green and imaged by fluorescence microscopy (f). The arrows point to cells and iron particles.

In Fig. 2 the authors propose that it should also be possible to couple thiosulfate oxidation to Fe(III) reduction. Organism with the genetic capacity to carry out this process include several cultivated members of alpha- and gammaproteobacterial families of the phylum Pseudomonadota. Have any efforts been made to provide evidence here? The authors should definitely comment on why this was not tested (possibly even faster side reactions?).

RESPONSE: In this study, we prioritized our efforts to validate the predicted sulfur metabolism associated with sulfide over thiosulfate primarily based on the consideration of environmental significance. Sulfate and iron(III) oxides are the two largest pools of oxidants in the Earth's exosphere^{22,23}. Dissimilatory reduction of sulfate consistently produces sulfide, leading to common co-occurrence of sulfide with abundant iron(III) oxides in diverse ecosystems, including anoxic marine/freshwater sediments and aquifers^{24–28}. While thiosulfate is also relevant as a reductant, we reasoned that validating the microbially-mediated reaction between sulfide and iron(III) oxides has the utmost implications to the understanding of coupled biogeochemical cycles of sulfur and iron.

Furthermore, testing the predicted thiosulfate metabolism in the cultivated members is potentially more constrained for the following reasons: (1) the lack of commercially available strains (e.g., Burkholderiaceae JOSHI), (2) obligatory dependence on light (e.g., *Halorhodospira halochloris*), and (3) halophilic lifestyle (*Ectothiorhodospira haloalkaliphila*), where high chloride levels interfere with chemical analysis of sulfur species. In our opinion, overcoming these challenges would require laborious experimental investigations and method development, representing a complex study on its own and being beyond the scope of this work.

The conclusions from line 413 onwards appear very far-fetched given the fact that growth with the process is hardly possible or does not even occur (in the case of FeS as the substrate).

RESPONSE: We would like to emphasize that microorganisms that are physiologically active but show limited or even no net growth capacity can nevertheless play a substantial role in global biogeochemical processes. Prominent examples include anaerobic methanotrophs (ANME), which are estimated to mediate up to 90% of global methane production in marine sediments^{29–32}; anammox bacteria, which account for 10–50% N loss in marine and terrestrial environments^{33–39}; and low-abundance *Desulfosporosinus*, which contribute to 1–10% microbial sulfate reduction in peatland while remaining in a zero-growth state⁴⁰. These microorganisms typically show limited or minimal growth capacity, with doubling times ranging from weeks to months^{37,40–42}. Yet, the relatively high metabolic activity at low growth rates enables them to drive key biogeochemical processes at ecosystem scales.

In the case of MISO, we now demonstrate that the process supports the growth of *D. alkaliphilus*, allowing doubling to tripling cell density starting from $3\text{--}5 \times 10^7$ cells ml⁻¹ within 4–13 days (new Fig. 4a-c). Although overall growth remains modest—likely constrained by increased maintenance energy requirements under stressful pH conditions—we observed a high metabolic rate powered by the MISO process. Specifically, cell-specific sulfate production rates of MISO reach around 1 fmol SO₄²⁻ cell⁻¹ d⁻¹ with FeS (Fig. 3b), 4 fmol SO₄²⁻ cell⁻¹ d⁻¹ with environmentally relevant dissolved sulfide concentrations (ca. 50 μM; Extended Data Fig. 2), and 16 fmol SO₄²⁻ cell⁻¹ d⁻¹ with elevated dissolved sulfide levels (1 mM; Fig. 3g). For context, these values exceed the cell-specific sulfate reduction rate (csSRR) commonly measured in marine sediments ($0.1\text{--}5 \times 10^{-4}$ fmol SO₄²⁻ cell⁻¹ d⁻¹) by one to four orders of magnitude⁴³, and align with csSRR observed in pure cultures of sulfate-reducing

bacteria ($2\text{-}47 \text{ fmol SO}_4^{2-} \text{ cell}^{-1} \text{ d}^{-1}$)⁴³. Thus, we argue that the MISO process has the potential to influence the biogeochemical sulfur cycle, by producing sulfate at a rate comparable with the sulfate reduction observed in natural environments. We have added these arguments in the Supplementary Information as the new section “Cell-specific metabolic rates of MISO” at lines 365-376,

Referee #2 (Remarks to the Author):

Chen et al., REVIEW

This manuscript describes nothing less than proof of a novel microbial metabolism, the oxidation of reduced sulfur compounds with solid iron oxides. The discovery that microbes not only perform but outpace this known chemical reaction is an important breakthrough for microbiology and earth & environmental sciences alike. In addition to this key discovery, the study also offers the to date most thorough overview of the distribution of sulfur cycling marker genes in the prokaryotic world, with newly and more strictly defined HMMs (that are made publicly available). Together with marker genes for iron reduction, these sulfur genes served to point the authors to likely candidate bacteria for the predicted metabolism, which then was tested with *Desulfurivibrio alkaliphilus* as model.

Although the paper is very well structured and written, it takes stamina to get through the wealth of physiological and chemical analyses that were necessary to unambiguously separate bacterial from chemical reactions and to exclude that sulfur disproportionation (a known trait of *D. alkaliphilus*) is the actual reason for sulfate production. The authors succeed with both, in a thorough and convincing way. Transcriptomic and bioinformatic analyses complement the study and provide additional, robust support of the novel metabolism by suggesting a plausible mechanism for the pathway. Methods, data, and statistics are throughout valid, of high quality, and appropriate (a few questions about statistical support remain, see detailed comments below). The paper ends with a balanced and careful conclusion on the ecological and geochemical implications, which showcases the authors' extraordinary overview of literature on sulfate formation in anoxic environments. I have only one major and a few minor (mostly technical) comments for improvement and would otherwise like to congratulate the authors to their profound work and exceptional discovery.

Main comment: while the sulfur gene analysis is extremely thorough, the Fe reduction potential seems not equally well supported – apparently only the most common (Gram negative) model organisms were included, newer insights from Gram positives (e.g. Paquete et al 2020, Light et al., 2018, 2019) or Archaea (e.g., ANME, methanogens) were not included in the search. Not surprisingly, the predicted Fe-reducing S oxidizers are restricted to *Desulfurivibrionaceae* and a few Proteobacteria. While this does not diminish the value of the discovery, it may limit its evolutionary, ecological, and environmental implication. If it is not possible to update the Fe reducer marker gene analysis, this limitation should at least be mentioned.

RESPONSE: We thank the reviewer for the comment. We have now updated our analysis by accounting for additional genes involved in dissimilatory iron metabolism. These include genes for (1) the flavin-based extracellular electron transfer (EET) pathway

(FmnAB-DmkAB-PplA-Ndh2-EetAB) in Gram-positive bacteria (e.g., *Listeria monocytogenes*⁴⁴); (2) outer-membrane multiheme cytochromes (OMCs; DFE_0450 and DFE_0464) that enabled EET between cells and insoluble minerals in *Desulfovibrio ferrophilus* IS5⁴⁵; (3) key determinants (GACE_1845 and GACE_1847) of insoluble Fe(III) reduction in *Geoglobus* spp.^{46,47}; (4) MmcA that facilitates iron reduction in methanogens (e.g., *Methanosarcina acetivorans*⁴⁸); and (5) predicted OMCs that have been proposed to mediate EET with metal oxides in ANME (e.g., *Ca. Methanoperedens*^{49,50}) and in putative electroactive bacteria⁵¹. The updated analysis recovered co-occurrence of genes for dissimilatory iron(III) reduction and sulfur oxidation in microorganisms from 37 bacterial and archaeal phyla (new Fig. S4 and S5), dramatically expanding the taxonomic diversity predicted by our original analysis. For example, beyond *Desulfurivibrionaceae* and *Proteobacteria*, we observed that (1) members from *Chloroflexota*, *Bacteroidota*, and *Marinisomatota* encode the sHdr complex and predicted OMCs, which would facilitate reaction 1; (2) members from *Aquificota*, *Campylobacterota*, *Methylomirabilis*, and *Nitrospirota* encode the Sox complex, as well as *D. ferrophilus*-related and/or predicted OMCs, which would facilitate reaction 2; and (3) Sqr-carrying members from *Halobacteriota*, *Firmicutes*, *Bdellovibrionota*, *Bacteroidota*, and *Acidobacteriota* encode the flavin-based EET pathway, *Geoglobus*-related OMCs or predicted OMCs, which have the potential to catalyze reaction 3. To incorporate these results, we have revised our text at lines 37-38 “Metabolic reconstructions predicted co-occurrence of sulfur compound oxidation and iron(III) oxide respiration in diverse members of 37 prokaryotic phyla”, lines 124-126 “We identified co-occurring genetic features for dissimilatory sulfur oxidation and extracellular iron(III) reduction metabolisms across 37 bacterial and archaeal phyla (Supplementary Text, Fig. S4-5)”, and lines 147-148 “Beyond *Desulfobacterota* and *Proteobacteria*, also members from 35 other microbial phyla have the potential to catalyze reactions 1-3 via different gene combinations (Supplementary Text, Fig. S4-5).” to indicate broader taxonomic diversity of potential Fe-reducing S oxidizers.

New Figure S4 | Phylogeny of microbes with the genomic potential for extracellular iron(III)-dependent oxidation of reduced sulfur compounds. The phylogenomic tree is pruned from the GTDB r95 bacterial and archaeal species tree. The presence/absence of metabolic pathways and/or key enzymes involved in dissimilatory sulfur oxidation (green) and dissimilatory iron(III) reduction (blue) is shown in the outer circles. DsrAB, dissimilatory sulfite reductase; DsrEFH, sulfur-relay system; SoxB, S-sulfosulfanyl-L-cysteine sulfohydrolase; Sqr, sulfide quinone reductase; FccB, sulfide dehydrogenase flavocytochrome subunit; sHdr, sulfur-oxidizing heterodisulfide reductase-like complex; CymA/napC, tetraheme quinol dehydrogenase; MacA, diheme cytochrome c involved in iron(III) reduction; PCC, porin cytochrome complex; MtrCAB, extracellular iron oxide respiratory system; MmcA, multiheme c-type cytochromes from *Methanosarcina acetivorans*; DFE_OMCs, outer-membrane multiheme c-type cytochromes (OMCs; DFE_0450 and DFE_0464) that enabled extracellular electron transfer (EET) between cells and insoluble minerals in *Desulfovibrio ferrophilus* IS5; Flavin_EET, the flavin-based EET pathway involved in iron reduction in Gram-positive bacteria (e.g., *Listeria monocytogenes*); GACE_OMCs, OMCs (GACE_1845 and 1847) involved in insoluble Fe(III) reduction in *Geoglobus* species; predicted OMCs, extracellular cytochromes that have ≥ 4 heme-binding sites; OmcS/F/Z, outer membrane cytochromes in *Geobacter* species.

New Figure S5 | Phylogeny of representative microbes with the metabolic potential for extracellular iron(III)-dependent oxidation of reduced sulfur compounds. Representative taxa are selected from Fig. S4 for visualization purposes. Pure cultures are highlighted with red circles and uncultured taxa depicted in blue. The presence/absence of metabolic pathways and/or key enzymes involved in dissimilatory sulfur oxidation (green), dissimilatory iron(III) reduction (blue), and carbon fixation (orange) is shown in the right panel. DsrAB, dissimilatory sulfite reductase; DsrL, NAD(P)H oxidoreductase; DsrEFH, sulfur-relay system; DsrMKJOP, [DsrC]-trisulfide reductase; AprBA, adenylylsulfate reductase; QmoABC, quinone-modifying oxidoreductase; SoxAX, L-cysteine S-thiosulfotransferase; SoxB, S-sulfosulfanyl-L-cysteine sulfohydrolase; SoxY, sulfur-oxidizing protein SoxY; SoxZ, sulfur-oxidizing protein SoxZ; SoxCD, sulfane dehydrogenase; Sqr, sulfide quinone reductase; FccB, sulfide dehydrogenase flavocytochrome subunit; sHdrAB1B2B3C1C2, sulfur-oxidizing heterodisulfide reductase-like complex; CymA/NapC, tetraheme quinol dehydrogenase; MacA, diheme cytochrome c involved in iron(III) reduction; PCC, porin cytochrome complex; MtrCAB, extracellular iron oxide respiratory system; MmcA, multiheme c-type cytochromes from *Methanosarcina acetivorans*; DFE_OMCs, the outer-membrane multiheme c-type cytochromes (OMCs; DFE_0450 and DFE_0464) that enabled extracellular electron transfer (EET) between cells and insoluble minerals in *Desulfovibrio ferrophilus* IS5; Flavin_EET, the flavin-based EET pathway involved in iron reduction in Gram-positive bacteria (e.g., *Listeria monocytogenes*); GACE_OMCs, OMCs (GACE_1845 and 1847) involved in insoluble Fe(III) reduction in *Geoglobus* species; predicted OMCs, extracellular cytochromes that have ≥ 4 heme-binding sites; OmcS/F/Z, outer membrane cytochromes in *Geobacter* species. RuBisCO, ribulose-bisphosphate carboxylase; CBB, Calvin-Benson-Bassham cycle; WL, Wood Ljungdahl pathway; rTCA, reverse tricarboxylic acid cycle.

Minor comments:

I. 28: consider changing “biogeochemical” to “geochemical” and “non-enzymatic” to “abiotic or non-biological”

RESPONSE: Revised as suggested but we changed “biogeochemical” to “environmental”, which we feel fits slightly better.

Lines 31-33. “One undiscovered metabolism is the coupling of sulfide oxidation with iron oxide reduction, a ubiquitous environmental process hitherto considered to be strictly abiotic.”

Fig. 1: the “Inorg. S” genes are separated by a gap into an apparent mostly oxidative (left) and mostly reductive (right) part – maybe consider choosing different colors/indicating this separation? Or remove the gap, if too much mixed?

RESPONSE: We have removed the gap between the Inorg. S genes in Fig. 1 as suggested.

Fig. 2, DeltaG for thiosulfate oxidation: do the red and black lines overlap? If so, please mention in the figure legend

RESPONSE: Yes. We have now mentioned the overlap between red and black lines in the legend of Fig. 2.

Fig. 3C, 16S rRNA gene copies: according to methods, incubations were started with 30 ml of a culture with approx. 10^8 cells/ml, into 60 ml, i.e. the starting conc. should be around 5×10^7 cells/ml. The qPCR data show around 5×10^6 : is this a problem with the qPCR assay, the DNA extraction efficiency, or an error?

RESPONSE: Please see also our responses to reviewer #1. We attribute underestimation of cell number by qPCR to low DNA extraction efficiency, especially for samples rich in iron oxides and FeS. We have now replaced qPCR results with direct flow cytometry-based cell counts (new Fig. 4a-d). The starting cell density in our new growth experiments ranges from $3\text{-}5 \times 10^7$ cells ml^{-1} (new Fig. 4a-d), prepared by 3-fold dilution of the master culture ($\sim 1\text{-}1.5 \times 10^8$ cells ml^{-1}) with fresh medium. We would like to point out that all our incubation experiments were performed in 60 ml serum bottles containing 30 ml culture and 30 ml headspace.

I. 283: “killed cells were served as controls” remove “were”

RESPONSE: Many thanks for the hint. Done as suggested.

I. 291-292: this is a bit confusing, as the first candidate genes for this metabolism were already identified in the genome searches for sulfur and Fe marker genes, weren't they? Consider rephrasing.

RESPONSE: We agree and have rephrased the sentence.

Lines 344-346. “After providing physiological evidence for microbial reduction of extracellular iron(III) being coupled to sulfide oxidation (MISO) in *D. alkaliphilus*, we revealed the differential activity of candidate genes of this metabolism by comparative transcriptomics under various ferrihydrite-amended and ferrihydrite-free growth conditions.”

I. 324-325: the role of PilA as “conductive nanowire”, although not fully disproven yet, is currently being replaced by cytochrome nanowires (OmcS, OmcZ etc.), while the actual role of PilA is not completely clear. This is also evident from your data, where you see a clear upregulation of omcS (and other MHCs) with iron oxides, but not of PilA (T4P). I suggest to reflect that also in the text, instead of “neutrally” entertaining two alternative roles.

RESPONSE: Done as suggested. We have rephrased the text and coloring scheme of Fig. 5 to more clearly outline the differences in the expression pattern between MHCs, Tad pili and PilA-based pili and their potential involvement in electron transfer and attachment.

Lines 378-379. “Transcription levels of *pilA*, encoding the main component of another type IV pilus, was high across all growth conditions but not increased during iron reduction.”

Extended data Fig. 4: it would be helpful to add the gene names to the locus tags, so one doesn't have to compare one by one with Extended data Fig. 3 to identify the MHCs

RESPONSE: Many thanks for the hint. Done as suggested.

I. 354: where is the statistical significance shown? Or does the figure only contain genes that are significantly upregulated? If so, please indicate in the legend, and please be specific which conditions were compared (there's only one obvious pair, sulfide/Fe(III) vs. sulfide/nitrate; or did you average all 3 Fe(III) conditions against both non-Fe(III) conditions?)

RESPONSE: Yes, the figure only contains MHCs that are significantly upregulated. To minimize false positive rate, we showed the MHCs that are significantly upregulated (adjusted $P < 0.01$) in all pairwise comparisons ($n = 6$) between Fe(III)-conditions ($n = 3$) vs. non-Fe(III) conditions ($n = 2$). Specifically, the six pairwise comparisons are sulfide/Fe(III) vs. sulfide/nitrate, sulfide/Fe(III) vs. S(0), FeS/Fe(III) vs. sulfide/nitrate, FeS/Fe(III) vs. S(0), formate/Fe(III) vs. sulfide/nitrate, and formate/Fe(III) vs. S(0), as shown in Extended Data Table 2. We reasoned that the genuine MHCs involved in dissimilatory Fe(III) reduction should be actively transcribed under all Fe(III)-conditions. We have revised the figure legend to clarify this.

I. 366: a TM-score of 0.56 is only slightly above the accepted threshold of 0.5 for structural homology. Consider additional support by adding rmsd or DALI scores?

RESPONSE: We have now added the DALI-score in Extended Data Fig. 5. We calculated the DALI-score between DA_402 and OmcS of 14.3, which is above the accepted threshold of 2 for significant structural similarity. We refrained from using rmsd as it is dominated by the largest error among different scores, and was shown to be the least representative of the degree of structural similarity⁵².

I. 471: “KoFAM” is listed as “KEGG” in Fig S2, please be consistent

RESPONSE: Thanks for the hint. We have now changed “KEGG” to “KoFAM” in Fig. S2.

I. 574-575: how were the ferrihydrites sterilized when preparing them with mortar and pestle?

RESPONSE: The synthesized ferrihydrite was not sterilized to avoid any potential changes in crystallinity. We minimized potential contamination during preparation by wiping the agate mortar and pestle with 70% ethanol before grinding. We did not have any indications that potential contaminants contributed to the activity and growth in ferrihydrite-incubated cultures as (i) periodic testing of cultures by direct Sanger sequencing of the 16S rRNA gene PCR product revealed >99% sequence identity to *D. alkaliphilus* AHT2 and (ii) >98% of transcriptomic reads mapped to the genome of *D. alkaliphilus* AHT2.

Supplementary Information:

I. 71: “were” should read “was”

RESPONSE: Thanks for the hint. Done as suggested.

I. 83: “oxidize sulfur to elemental sulfur” should read “oxidize sulfide to elemental sulfur”

RESPONSE: Thanks for the hint. Done as suggested.

I. 121: “share close homology” NO! they “share close identity/similarity” – there are no close or distant homologs: if they were not homologs, they should not be in the same phylogenetic tree.

RESPONSE: Thanks for the hint. Changed to “share close similarity” as suggested.

I. 142-143: statistical significance of temporal changes: not indicated in the figure? Please add.

RESPONSE: We have replaced the section with new results of growth experiments. The statistical significance of temporal changes under each growth condition have been added to the legend of the new figure (Fig. 4).

I. 233-247, Relevance of MISO: I can see why the authors think this is a conservative estimate, given the bio and geo chemical recycling of Fe. But on the other hand, not the entire riverine input of Fe oxides will end up in zones without other oxidants, and then oxygen and nitrate (and depending on the minerals even manganese) are more favorable electron acceptors, so that part of the Fe may only undergo chemical transformation if any.

RESPONSE: We agree that it is difficult to assess if our first, rough estimate is an over- or underestimation. We have thus removed the sentence “The inferred contribution of MISO to the global sulfide-to-sulfate flux in marine sediments is thus likely an underestimation of its true contribution.”

L. 245: “100-300 times of redox cycles” delete “times of”

RESPONSE: Thanks for the hint. Done as suggested.

Referee #2 (Remarks on code availability):

Code available and well described. I did not try to reproduce the results of the paper.

Referee #3 (Remarks to the Author):

Review of Nature manuscript 2024-10-20940 “Microbial iron oxide respiration coupled to sulfide oxidation” by Chen et al.

Overview

In this article, the authors present experimental (and genomic) evidence for the existence of microbial sulfide oxidation coupled to Fe(III) mineral reduction at slightly alkaline pH (9.3), a process that has been suggested (mainly speculated) to exist for a long time, but so far no convincing experimental evidence (with a pure culture of microorganisms) has been provided. This process represents an important link between the sulfur and iron biogeochemical cycles and has been shown so far to be purely abiotic. The study presented here is novel because it provides for the first time convincing evidence that microbes are able to harvest the energy available in the reaction between sulfide and Fe(III). This evidence comes from a thorough genomic analysis and laboratory experiments with a microbial pure culture, i.e. *Desulfurivibrio alkaliphilus*. Since the sulfur and iron cycles are two of the most important environmental biogeochemical cycles, this is very important and certainly deserves publication.

The data provided in the paper are very convincing although I am not an expert in the genomic analysis that has been performed. However, the microbial cultivation work was very carefully done, the setups and in particular the control experiments were very smart and the authors provide convincing data that the observed process is indeed enzymatic and not an abiotic reaction. The data is clearly presented and easy to follow. In my opinion, the conclusions drawn from the data are robust and reliable. Both the abstract and the main manuscript text are well written and easy to read.

Below, I provide some general comments and specific line-by-line suggestions for revising the manuscript and I hope that these are helpful to the authors.

General comments and questions

1. Please be consistent with the terminology: sometimes the authors write “sulfur oxidation” when they mean “sulfide oxidation”, sometimes I was not sure what they mean (see also specific comments below). Maybe at one point it would be good to define “sulfur oxidation” as oxidation of reduced sulfur species in general? Please check carefully and revise throughout the whole manuscript.

RESPONSE: Done as suggested. To avoid confusion, we now introduce the term ‘sulfur oxidation’ at the beginning of the manuscript.

Lines 57-59. “Yet, the microbial genomic signatures for the full spectrum of dissimilatory sulfur metabolisms, including oxidation of reduced sulfur compounds (hereafter called ‘sulfur oxidation’), has not been explored.”

2. The authors used ferrihydrite for their experiments but in the text use the term “iron oxide”. Ferrihydrite is an “iron(III) oxyhydroxide” (maybe for simplification you could say “iron hydroxide”). I recommend either using “iron oxyhydroxide” or at least defining “iron oxide” as a simplified term for ferrihydrite in the beginning of the manuscript.

RESPONSE: Many thanks for the hint. We now introduce the term ‘iron(III) oxides’ at the start of the manuscript as suggested.

Lines 66-67. “Ferric oxyhydroxides and oxides (hereafter called ‘iron(III) oxides’) represent one of the largest pools of oxidants on Earth’s surface”

3. Overall I find the results provided from the *D. alkaliphilus* experiments convincing (demonstrating sulfide oxidation coupled to iron(III) reduction). I am wondering whether S-isotope analyses could provide an additional argument in favor of this microbial process. I don’t think it is necessary for the present paper, but maybe for future publications. I also thought about using Fe-isotope analysis, but I assume the Fe-isotope fractionation when comparing the abiotic reduction of ferrihydrite by sulfide to the biotic process will not be very different from each other. Nevertheless, maybe worth a try. Both isotope systems (S and Fe) could help to get some information about the role of microbial (and abiotic) sulfide oxidation coupled to iron(III) reduction on early Earth.

RESPONSE: We have considered sulfur isotope analysis for distinguishing sulfide oxidation and sulfur disproportionation. However, the paper by Pellerin et al. 2019⁵³ showed that *D. alkaliphilus* produced unusually large sulfur isotopic enrichments in sulfate of +12 to +13.1 ‰ during sulfide oxidation with nitrate, which is in the same high range as generally measured during sulfur disproportionation (+11.0 to +35.3‰). This indicates that distinguishing sulfide oxidation from sulfur disproportionation in *D. alkaliphilus* based on sulfur isotope fractionation may be challenging. Similarly, Fe isotope fractionations between aqueous Fe²⁺ and iron oxide minerals show comparable magnitudes for biotic (-0.5 to -2.3‰) and abiotic (0 to -1‰) iron reduction⁵⁴. This highlights the inherent difficulties in distinguishing alternative iron reduction pathways based on Fe isotope fractionations. We agree that it is relevant to analyze S- and Fe-isotope signatures generated by different metabolisms of *D. alkaliphilus*, and thus to provide new data for interpretation of the isotopic records. Yet, we anticipate that the isotope analysis would not change our main conclusions, and thus the assessment of sulfur and iron isotope fractionation by MISO warrants a separate study.

4. Did the authors try to use ¹³C labelled CO₂/bicarbonate to demonstrate CO₂ fixation and growth in cultures incubated with sulfide and ferrihydrite and in cultures incubated with FeS and ferrihydrite?

RESPONSE: Thank you for your suggestion. We have now performed additional experiments by adding ¹³C-labelled bicarbonate (10 atom%) to the ferrihydrite-incubated cultures fed with sulfide or FeS. Bulk carbon isotope analysis revealed gradual enrichment of ¹³C in the living cells over six days of incubation (p < 0.01; ANOVA) (new Fig. 4e-f). To further show ¹³C incorporation at the single-cell level, we analyzed cells from the sulfide incubation with nano-scale secondary ion mass spectrometry (NanoSIMS). We observed that the ¹³C abundance of cells after a 5-day incubation period reached a median of 4 atom%, whereas cells at day 0 were close to natural ¹³C abundance (1.08 atom%) (new Fig.

4g-i). The significant enrichment of ^{13}C in the bulk biomass and in the single cells clearly demonstrates fixation of ^{13}C -bicarbonate by the culture for growth. Consistent with this, we observed that most genes involved in the Wood Ljungdahl pathway ranked among the top 30% most highly transcribed genes in cultures incubated with sulfide or FeS (new Fig. S14), which suggests their involvement in autotrophic carbon fixation.

We have included these results in the main text at lines 264-268 “To further explore the association between growth and carbon fixation, we supplemented parallel cultures with ^{13}C -bicarbonate (10 atom%), ferrihydrite, and sulfide or FeS. Carbon isotope composition analysis of bulk biomass and single cells revealed enrichment of ^{13}C in living cells after 5-day incubation (Fig. 4e-i), reflecting a chemoautotrophic lifestyle of *D. alkaliphilus* when growing on ferrihydrite and dissolved sulfide or FeS.” and in new section “Active transcription of genes involved in Wood Ljungdahl pathway under different growth conditions” in the Supplementary Information at lines 340-349, including new Fig. S14.

Selected panels from the new Figure 4 | *Desulfurivibrio alkaliphilus* cells incorporate bicarbonate-derived ^{13}C during incubation with ferrihydrite and sulfide (1 mM) or FeS. **e**, Bulk ^{13}C abundance (atom %) increased significantly ($p = 0.0002$; ANOVA) over six days in living cultures ($n = 3$ replicates; orange circles) incubated with ferrihydrite (ca. 62 mM) and 10% ^{13}C -bicarbonate, and with daily addition of sulfide (1 mM). Bulk ^{13}C abundance did not increase significantly ($p = 0.05$; ANOVA) in killed controls (gray circles). The arrows indicate sulfide additions. **f**, Bulk ^{13}C abundance (atom %) increased significantly ($p = 0.003$; ANOVA) over six days in living cultures ($n = 3$ replicates; orange circles) incubated with ferrihydrite (ca. 62 mM) and 10 atom% ^{13}C -bicarbonate, and with two spikes of FeS (~1 mM). Bulk ^{13}C abundance did not increase significantly ($p = 0.520$; ANOVA) in killed controls (gray circles). The arrows indicate FeS additions. The statistical significance of comparison between living and killed treatments on day 6 is indicated by * ($p < 0.05$) based on the t-test. The natural level of ^{13}C (~1.12%) is shown as a horizontal dashed line. Two points at day 6 of FeS living cultures (panel **b**) were jittered for visualization purposes.

Selected panels from the new Figure 4 | Nano-scale secondary ion mass spectrometry (NanoSIMS) analysis reveals incorporation of ^{13}C into individual cells after 5-day incubation with 10% ^{13}C bicarbonate and ferrihydrite, and with daily addition of dissolved sulfide (1 mM). g, NanoSIMS $^{12}\text{C}^{14}\text{N}^-$ secondary ion images showing cellular biomass obtained at day 0 and day 5. h, NanoSIMS images showing ^{13}C content distribution for cells harvested at day 0 and 5. Colour scale refers to ^{13}C atom%. Scale bars correspond to 5 μm . i, Dot plots displaying the ^{13}C content of individual cells at day 0 and 5. The adjacent box plots show summary statistics indicating the median, the 25th, and 75th percentiles, respectively. The whiskers extend 1.5 times the interquartile ranging from the first and third quartiles. The dashed lines refer to the mean ^{13}C content of the cells at the beginning of the incubation (day 0).

New Figure S14 | Ranked transcription abundance plots of *Desulfurivibrio alkaliphilus* genes showing the relative transcription level of Wood Ljungdahl (WL) pathway genes (orange circles) under different growth conditions. a, Each point (black) is the medium transcription level of a gene and error bars (green) correspond to the interquartile range of replicate cultures ($n = 4$). The two vertical, dashed red lines indicate top 10% and 30% transcribed genes. **b,** Boxplot displaying the ranked transcription abundance of WL genes in three ferrihydrite-amended conditions (red) and two ferrihydrite-free growth conditions (cyan). The center lines and box limits of the boxplot denote the median, and the 25% and 75% percentile of the ranked transcription abundance. The whiskers extend

1.5 times the interquartile range from the 25th and 75th percentiles. Genes involved in WL pathway: *acsA*, acetyl-CoA synthase / CO dehydrogenase (ACS/CODH), complex subunit alpha; *acsB*, ACS/CODH complex subunit beta; *acsC*, ACS/CODH complex subunit gamma; *acsD*, ACS/CODH complex subunit delta; *acsE*, 5-methyltetrahydrofolate corrinoid/iron sulfur protein methyltransferase; *fhs*, formate-tetrahydrofolate ligase; *fo/D*, methenyltetrahydrofolate cyclohydrolase; *metF*, methylenetetrahydrofolate reductase. Fh, ferrihydrite; RPKM, reads per kilobase per million mapped reads.

Specific comments

1. L32, L52, L57: sulfur or sulfide oxidation?

RESPONSE: See also response to comment above. To avoid confusion, we now introduce the term 'sulfur oxidation' at the beginning of the manuscript.

Lines 57-59. "Yet, the microbial genomic signatures for the full spectrum of dissimilatory sulfur metabolisms, including oxidation of reduced sulfur compounds (hereafter called 'sulfur oxidation'), has not been explored."

2. L59: this would be a good place to define "iron oxides". Here you mean all iron(III) minerals including oxides (hematite, magnetite) and oxyhydroxides (goethite, lepidocrocite, ferrihydrite). I therefore usually prefer "iron(III) (oxyhydr)oxides" to include all of them. Maybe here you can explain that from now on the term "iron oxides" will be used for simplicity?

RESPONSE: Done as suggested (see also response to comment above).

Lines 66-67. "Ferric oxyhydroxides and oxides (hereafter called 'iron(III) oxides') represent one of the largest pools of oxidants on Earth's surface"

3. L62: what does "this process" refer to?

RESPONSE: We have revised the sentence for clarification.

Lines 69-70. "However, current biogeochemical models consider the reaction of sulfide and iron(III) oxides as purely abiotic, primarily producing elemental sulfur and poorly crystalline FeS..."

4. L106 and L109: sulfur or sulfide oxidation?

RESPONSE: See also response to comments above. To avoid confusion, we now introduce the term 'sulfur oxidation' at the beginning of the manuscript.

5. L111: I assume you mean "dissolved iron(III)"?

RESPONSE: Thanks for the hint. Changed as suggested.

6. L114: "iron(III) reduction".

RESPONSE: Changed as suggested.

7. L117 and L123: “iron(III) reduction”.

RESPONSE: Changed as suggested.

8. L130: “iron(III)-reducer”.

RESPONSE: Changed as suggested.

9. L135: “...extracellular iron(III)-dependent thiosulfate oxidation”?

RESPONSE: Changed as suggested.

10. L138: “iron(III)-dependent”.

RESPONSE: Changed as suggested.

11. L183: should the cell numbers not increase (instead of staying constant)?

RESPONSE: Our new growth experiments (Fig. 4a-d) revealed that FeS and ferrihydrite could support cell growth. Thus, we have now replaced the sentence with a full paragraph discussing the growth of *D. alkaliphilus* with different combinations of electron donors and acceptors.

12. L204 and L206: “iron(III) reduction”.

RESPONSE: Changed as suggested.

13. L214 and L218: I recommend writing “poorly soluble”, not “insoluble”.

RESPONSE: Changed as suggested.

14. L242: is it really “amorphous” or maybe rather “poorly crystalline”?

RESPONSE: Thanks for the hint. We have now used poorly crystalline FeS through the manuscript.

15. L244: regarding the 1:8 stoichiometry: if the cells grow and use some electrons to also fix CO₂, the ratio should be even a bit higher (slightly more sulfide oxidized than iron(III) reduced), right?

RESPONSE: We fully agree. The growing cells divert part of the electrons from FeS-sulfide oxidation for carbon fixation, resulting in the formation of relatively less Fe(II) (or more sulfate) as the expected 1:8 stoichiometry. This is consistent with the slightly higher sulfate-to-Fe(II) formation ratio that we observed (close to 1:6), as well as the growth of cells incubated with FeS and ferrihydrite (Fig. 4b). We have added this explanation in the main text at lines 195-197 “The slight deviation from the expected stoichiometry most likely results from reverse electron flow for carbon fixation (Supplementary Text, Fig. S8-9)”.

16. L307: “iron(III)-respiring”.

RESPONSE: Changed as suggested.

17. L316: “iron(III) oxides”.

RESPONSE: Changed as suggested.

18. L379: “dissolved iron(III)”.

RESPONSE: Changed as suggested.

19. L511: “iron(III)-dependent”.

RESPONSE: Changed as suggested.

20. L538: see comment above regarding the terminology for the FeS: I personally feel that “poorly ordered” or “poorly crystalline” are more appropriate than “amorphous”.

RESPONSE: Yes, we agree that poorly crystalline FeS is more appropriate, as true amorphous FeS does not exist⁵⁵. We have now changed ‘amorphous FeS’ into ‘poorly crystalline FeS’ throughout the whole manuscript.

21. L572: I think following sulfide and/or sulfate provides evidence for microbial activity, but not for growth. This requires biomass quantification, cell counts, protein content quantification, ¹³CO₂ fixation, etc.

RESPONSE: Thanks for the hint. We now use “microbial activity” instead of “growth” at line 638.

22. L625-626: during the acidification step, remaining sulfide will react with remaining Fe(III) (especially when the ferrihydrite gets dissolved), right? Does this lead to an overestimation of the amount of formed Fe(II)?

RESPONSE: Yes, sulfide liberated from, e.g., FeS, during the acidification step can react with ferrihydrite and may thus lead to overestimation of HCl-extractable Fe(II) as previously reported⁵⁶. To estimate the effect of this reaction in our incubation experiment, we correlated the HCl-extractable Fe(II) in two treatments from our FeS/ferrihydrite incubation experiment. These include (1) the abiotic control fed with ferrihydrite and FeS and (2) the FeS-only control, both of which are free of biologically produced Fe(II). We reasoned that the FeS in abiotic controls gets dissolved during the acidification step, with the released sulfide reacting with ferrihydrite that produces additional Fe(II) on top of FeS-Fe(II). Yet, the same reaction between ferrihydrite and acid-liberated sulfide does not occur in the FeS-only control. Correlation analysis of HCl-extractable Fe(II) in abiotic control vs. FeS-only control revealed a slope of 1.23 ± 0.07 (new Fig. S9), indicating a $23 \pm 7\%$ overestimation of Fe(II). Nevertheless, we argue this overestimation does not affect our conclusion because we interpreted most of Fe(II) data qualitatively. The only quantitative case was the stoichiometry

during the FeS/ferrhydrite incubation. Correcting the observed stoichiometry (~1:6) by accounting for the overestimation of Fe(II) yielded ~1:4.9, which still supports the reverse electron flow for carbon fixation as discussed above.

We have added these results in the Supplementary Information at lines 189-203, including new Fig. S9.

New Figure S9 | Overestimation of HCl-extractable Fe(II) in the incubations containing both ferrhydrite and sulfide. The Fe(II) concentrations shown here are from two treatments in the FeS experiment, including (1) the abiotic control fed with both ferrhydrite and FeS and (2) FeS-only control fed with the same amount of FeS. The measured Fe(II) in the FeS-control derived purely from FeS, and in the abiotic control from both FeS and ferrhydrite/sulfide reaction during the acidification step. The slope (mean \pm standard deviation) is estimated via linear regression analysis.

References

1. Müller, A. L., Kjeldsen, K. U., Rattei, T., Pester, M. & Loy, A. Phylogenetic and environmental diversity of DsrAB-type dissimilatory (bi)sulfite reductases. *ISME J.* **9**, 1152–1165 (2015).
2. Loy, A. *et al.* Reverse dissimilatory sulfite reductase as phylogenetic marker for a subgroup of sulfur-oxidizing prokaryotes. *Environ. Microbiol.* **11**, 289–299 (2009).
3. Anantharaman, K. *et al.* Expanded diversity of microbial groups that shape the dissimilatory sulfur cycle. *ISME J.* **12**, 1715–1728 (2018).
4. Meyer, B. & Kuever, J. Molecular analysis of the diversity of sulfate-reducing and sulfur-oxidizing prokaryotes in the environment, using *aprA* as functional marker gene. *Appl. Environ. Microbiol.* **73**, 7664–7679 (2007).
5. Meyer, B., Imhoff, J. F. & Kuever, J. Molecular analysis of the distribution and phylogeny of the *soxB* gene among sulfur-oxidizing bacteria - evolution of the Sox sulfur oxidation enzyme system. *Environ. Microbiol.* **9**, 2957–2977 (2007).
6. Speciale, G., Jin, Y., Davies, G. J., Williams, S. J. & Goddard-Borger, E. D. YihQ is a sulfoquinovosidase that cleaves sulfoquinovosyl diacylglyceride sulfolipids. *Nat. Chem. Biol.* **12**, 215–217 (2016).
7. Kostanjevecki, V. *et al.* A membrane-bound flavocytochrome c-sulfide dehydrogenase from the purple phototrophic sulfur bacterium *Ectothiorhodospira vacuolata*. *J. Bacteriol.* **182**, 3097–3103 (2000).
8. Gregersen, L. H., Bryant, D. A. & Frigaard, N.-U. Mechanisms and evolution of oxidative sulfur metabolism in green sulfur bacteria. *Front. Microbiol.* **2**, 116 (2011).
9. Heinzinger, N. K., Fujimoto, S. Y., Clark, M. A., Moreno, M. S. & Barrett, E. L. Sequence analysis of the *phs* operon in *Salmonella typhimurium* and the contribution of thiosulfate reduction to anaerobic energy metabolism. *J. Bacteriol.* **177**, 2813–2820 (1995).
10. Krafft, T. *et al.* Cloning and nucleotide sequence of the *psrA* gene of *Wolinella succinogenes* polysulphide reductase. *Eur. J. Biochem.* **206**, 503–510 (1992).
11. Dahl, C., Franz, B., Hensen, D., Kesselheim, A. & Zigann, R. Sulfite oxidation in the purple sulfur bacterium *Allochromatium vinosum*: identification of SoeABC as a major player and relevance of SoxYZ in the process. *Microbiology* **159**, 2626–2638 (2013).
12. Williams, B. T. *et al.* Bacteria are important dimethylsulfoniopropionate producers in coastal sediments. *Nat. Microbiol.* **4**, 1815–1825 (2019).
13. Tanabe, T. S. *et al.* A cascade of sulfur transferases delivers sulfur to the sulfur-oxidizing heterodisulfide reductase-like complex. *Protein Sci.* **33**, e5014 (2024).
14. Pereira, I. A. C. *et al.* A comparative genomic analysis of energy metabolism in sulfate reducing bacteria and archaea. *Front. Microbiol.* **2**, 69 (2011).
15. Thorup, C., Schramm, A., Findlay, A. J., Finster, K. W. & Schreiber, L. Disguised as a sulfate reducer: Growth of the deltaproteobacterium *Desulfurivibrio alkaliphilus* by sulfide oxidation with nitrate. *MBio* **8**, (2017).
16. Krulwich, T. A. Alkaliphiles: ‘basic’ molecular problems of pH tolerance and bioenergetics. *Mol. Microbiol.* **15**, 403–410 (1995).
17. Padan, E., Bibi, E., Ito, M. & Krulwich, T. A. Alkaline pH homeostasis in bacteria: new insights. *Biochim. Biophys. Acta* **1717**, 67–88 (2005).
18. Hettiarachchi, E. & Grassian, V. H. Impact of surface adsorption on DNA structure and stability: Implications for environmental DNA interactions with iron oxide surfaces. *Langmuir* **40**, 27194–27205 (2024).
19. Ray, R., Lizewski, S., Fitzgerald, L. A., Little, B. & Ringeisen, B. R. Methods for imaging *Shewanella oneidensis* MR-1 nanofilaments. *J. Microbiol. Methods* **82**, 187–191 (2010).
20. Gorby, Y. A. *et al.* Electrically conductive bacterial nanowires produced by *Shewanella oneidensis* strain MR-1 and other microorganisms. *Proc. Natl. Acad. Sci. U. S. A.* **103**, 11358–11363 (2006).
21. Shi, L. *et al.* The roles of outer membrane cytochromes of *Shewanella* and *Geobacter* in extracellular electron transfer. *Environ. Microbiol. Rep.* **1**, 220–227 (2009).
22. Fike, D. A., Bradley, A. S. & Rose, C. V. Rethinking the ancient sulfur cycle. *Annu. Rev.*

- Earth Planet. Sci.* **43**, 593–622 (2015).
23. Hayes, J. M. & Waldbauer, J. R. The carbon cycle and associated redox processes through time. *Philos. Trans. R. Soc. Lond. B Biol. Sci.* **361**, 931–950 (2006).
 24. Flynn, T. M., O'Loughlin, E. J., Mishra, B., DiChristina, T. J. & Kemner, K. M. Sulfur-mediated electron shuttling during bacterial iron reduction. *Science* **344**, 1039–1042 (2014).
 25. Hansel, C. M. *et al.* Dominance of sulfur-fueled iron oxide reduction in low-sulfate freshwater sediments. *ISME J.* **9**, 2400–2412 (2015).
 26. Poulton, S. W. Sulfide oxidation and iron dissolution kinetics during the reaction of dissolved sulfide with ferrihydrite. *Chem. Geol.* **202**, 79–94 (2003).
 27. Poulton, S. W., Krom, M. D. & Raiswell, R. A revised scheme for the reactivity of iron (oxyhydr)oxide minerals towards dissolved sulfide. *Geochim. Cosmochim. Acta* **68**, 3703–3715 (2004).
 28. Jakobsen, R. & Postma, D. Redox zoning, rates of sulfate reduction and interactions with Fe-reduction and methanogenesis in a shallow sandy aquifer, Rømø, Denmark. *Geochim. Cosmochim. Acta* **63**, 137–151 (1999).
 29. Miyajima, Y. *et al.* Impact of concurrent aerobic-anaerobic methanotrophy on methane emission from marine sediments in gas hydrate area. *Environ. Sci. Technol.* **58**, 4979–4988 (2024).
 30. Stranne, C. *et al.* Anaerobic oxidation has a minor effect on mitigating seafloor methane emissions from gas hydrate dissociation. *Commun. Earth Environ.* **3**, 1–10 (2022).
 31. Knittel, K. & Boetius, A. Anaerobic oxidation of methane: progress with an unknown process. *Annu. Rev. Microbiol.* **63**, 311–334 (2009).
 32. Girguis, P. R., Cozen, A. E. & DeLong, E. F. Growth and population dynamics of anaerobic methane-oxidizing archaea and sulfate-reducing bacteria in a continuous-flow bioreactor. *Appl. Environ. Microbiol.* **71**, 3725–3733 (2005).
 33. Yao, Y. *et al.* Global variations and controlling factors of anammox rates. *Glob. Chang. Biol.* **29**, 3622–3633 (2023).
 34. Xu, X. *et al.* Global patterns and drivers of coupling between anammox and denitrification processes across inland aquatic ecosystems. *Commun. Earth Environ.* **6**, 1–11 (2025).
 35. Kuypers, M. M. M., Marchant, H. K. & Kartal, B. The microbial nitrogen-cycling network. *Nat. Rev. Microbiol.* **16**, 263–276 (2018).
 36. Yang, X.-R. *et al.* Potential contribution of anammox to nitrogen loss from paddy soils in Southern China. *Appl. Environ. Microbiol.* **81**, 938–947 (2015).
 37. Zhang, L. *et al.* Maximum specific growth rate of anammox bacteria revisited. *Water Res.* **116**, 296–303 (2017).
 38. Strous, M. *et al.* Missing lithotroph identified as new planctomycete. *Nature* **400**, 446–449 (1999).
 39. Sun, J. *et al.* Potential growth of anammox bacteria under aerobic conditions. *Environ. Sci. Technol.* **58**, 18244–18254 (2024).
 40. Hausmann, B., Pelikan, C., Rattei, T., Loy, A. & Pester, M. Long-term transcriptional activity at zero growth of a cosmopolitan rare biosphere member. *MBio* **10**, (2019).
 41. Strous, M., Heijnen, J. J., Kuenen, J. G. & Jetten, M. S. M. The sequencing batch reactor as a powerful tool for the study of slowly growing anaerobic ammonium-oxidizing microorganisms. *Appl. Microbiol. Biotechnol.* **50**, 589–596 (1998).
 42. Bhattarai, S., Cassarini, C. & Lens, P. N. L. Physiology and distribution of Archaeal methanotrophs that couple anaerobic oxidation of methane with sulfate reduction. *Microbiol. Mol. Biol. Rev.* **83**, (2019).
 43. Hoehler, T. M. & Jørgensen, B. B. Microbial life under extreme energy limitation. *Nat. Rev. Microbiol.* **11**, 83–94 (2013).
 44. Light, S. H. *et al.* A flavin-based extracellular electron transfer mechanism in diverse Gram-positive bacteria. *Nature* **562**, 140–144 (2018).
 45. Deng, X., Dohmae, N., Nealson, K. H., Hashimoto, K. & Okamoto, A. Multi-heme cytochromes provide a pathway for survival in energy-limited environments. *Sci. Adv.* **4**,

- eaao5682 (2018).
46. Mardanov, A. V. *et al.* The *Geoglobus acetivorans* genome: Fe(III) reduction, acetate utilization, autotrophic growth, and degradation of aromatic compounds in a hyperthermophilic archaeon. *Appl. Environ. Microbiol.* **81**, 1003–1012 (2015).
 47. Manzella, M. P., Reguera, G. & Kashefi, K. Extracellular electron transfer to Fe(III) oxides by the hyperthermophilic archaeon *Geoglobus ahangari* via a direct contact mechanism. *Appl. Environ. Microbiol.* **79**, 4694–4700 (2013).
 48. Gupta, D., Chen, K., Elliott, S. J. & Nayak, D. D. MmcA is an electron conduit that facilitates both intracellular and extracellular electron transport in *Methanosarcina acetivorans*. *Nat. Commun.* **15**, 3300 (2024).
 49. Leu, A. O. *et al.* Lateral gene transfer drives metabolic flexibility in the anaerobic methane-oxidizing Archaeal family Methanoperedenaceae. *MBio* **11**, (2020).
 50. Zhang, X. *et al.* Multi-heme cytochrome-mediated extracellular electron transfer by the anaerobic methanotroph ‘*Candidatus Methanoperedens nitroreducens*’. *Nat. Commun.* **14**, 6118 (2023).
 51. Garber, A. I., Nealson, K. H. & Merino, N. Large-scale prediction of outer-membrane multiheme cytochromes uncovers hidden diversity of electroactive bacteria and underlying pathways. *Front. Microbiol.* **15**, 1448685 (2024).
 52. Kufareva, I. & Abagyan, R. Methods of protein structure comparison. *Methods Mol. Biol.* **857**, 231–257 (2012).
 53. Pellerin, A. *et al.* Large sulfur isotope fractionation by bacterial sulfide oxidation. *Sci Adv* **5**, eaaw1480 (2019).
 54. Alison McAnena, Silke Severmann, Romain Guilbaud, Simon W. Poul. Iron isotope fractionation during sulfide-promoted reductive dissolution of iron (oxyhydr)oxide minerals. *Geochimica et Cosmochimica Acta* **369**, 17–34 (2024).
 55. Rickard, D. & Morse, J. W. Acid volatile sulfide (AVS). *Mar. Chem.* **97**, 141–197 (2005).
 56. Peiffer, S. *et al.* Pyrite formation and mineral transformation pathways upon sulfidation of ferric hydroxides depend on mineral type and sulfide concentration. *Chem. Geol.* **400**, 44–55 (2015).

Response to reviewers' comments

We sincerely thank the reviewers for their constructive feedback and insightful suggestions. We believe the manuscript has greatly benefited from this review process and are grateful for the reviewers' time and expertise.

Referees' comments:

Referee #1 (Remarks to the Author):

All my concerns were clarified by additional experiments and methods.

Response: We appreciate your acknowledgment of the new data and methods.

Referee #2 (Remarks to the Author):

This is an extensive revision of a manuscript I had for review in its original version, and I can only congratulate the authors to an excellent and thorough job that has fully addressed all my concerns and doubts, and significantly improved both the data by adding new experiments and analyses, and the presentation of the manuscript: this is how peer review should work, at its best.

Again, congratulations to a groundbreaking discovery and a now even better presentation.

Response: We are grateful for your encouraging assessment and are delighted that the revised manuscript meets your approval.

Referee #3 (Remarks to the Author):

The authors did an excellent job considering my suggestions and comments in their revised manuscript. No further changes required from my side.

Response: We appreciate your positive feedback and confirmation that our revisions adequately addressed your comments.